# What Survives Privatization? A Guide to Structure and Utility in Differentially Private Genome-Wide Association Studies

## Abstract

Single nucleotide polymorphisms (SNPs) are among the most common and informative forms of genetic variation in the human genome and constitute the primary data representation used in genome-wide association studies (GWAS). Due to their extreme dimensionality, strong correlation structure, and the presence of both population-level and familial dependencies, SNP datasets exhibit structural properties that fundamentally distinguish them from standard tabular data. At the same time, genomic data is uniquely sensitive; it is immutable, identifying, and shared across relatives, and has been shown to be vulnerable to a wide range of attacks, including membership inference, reconstruction, and kinship inference. As a result, protecting SNP data has become a critical and practically unavoidable requirement.

Differential privacy (DP) provides a rigorous mathematical framework for protecting sensitive data under strong adversarial assumptions. However, in the context of GWAS, the design and evaluation of meaningful DP mechanisms crucially depend on understanding the biological, statistical, and structural properties of SNP data and the downstream analysis pipelines. For a typical privacy researcher, acquiring even the minimal domain knowledge required to reason correctly about the structure of genomic data and the associated analysis pipelines represents a substantial and time-consuming barrier. Yet, without this understanding, progress in private genomic data analysis risks being misguided or misleading.

This survey explicitly bridges this gap. We provide a structured, self-contained primer on the structural properties of SNP data and the core analytical workflows of GWAS, focusing on the aspects most consequential for privacy definitions, mechanism design, and utility. Building on this foundation, we present a comprehensive and systematic overview of differentially private methods for SNP datasets. We organize the literature through a release-oriented taxonomy that reframes existing approaches in terms of what survives privatization, revealing the design choices and trade-offs that shape their scientific and practical utility. Finally, we identify key open challenges arising from mismatches between existing differential privacy methodologies and the scientific, statistical, and operational realities of genomic data analysis, and outline future research directions toward principled and deployable privacy-preserving GWAS.

## Contents

# 1 Introduction

The issues of fairness, consent, general expectations of personal privacy, and data governance become particularly complex, and sometimes even contradictory, in the context of genomic data (Dedrickson, 2017; Zeevi, 2019; Kaye, 2012; Guo et al., 2023; Belani et al., 2021). Genomes constitute a compact yet still only partially understood representation of living organisms (ENCODE Project Consortium, 2012). They are highly identifying, inherently sensitive, and exhibit strong correlations both within families and across populations, creating unique and persistent privacy risks Gymrek et al. (2013); Erlich & Narayanan (2014); Humbert et al. (2013); Deznabi et al. (2017); Humbert et al. (2017).

Differential privacy (DP) Dwork (2006); Dwork et al. (2014) provides an attack-agnostic and theoretically grounded framework for offering individuals plausible deniability with respect to their data. However, applying DP to genomic datasets presents challenges that extend well beyond standard tabular settings. Although genomic data can be represented in matrix form, it exhibits distinctive structural properties, including extremely high dimensionality (Visscher et al., 2017), complex and only partially understood correlation patterns across loci and individuals (Slatkin, 2008; Phillips, 2008), and highly specialized downstream analysis pipelines. Moreover, genomic datasets are rarely used for a single task; instead, they are repeatedly analyzed across diverse studies (Consortium et al., 2015; Bycroft et al., 2018). Consequently, preserving utility in this domain requires maintaining not only performance on specific downstream analyses, but also fundamental structural and biological properties of the data.

These characteristics create a substantial barrier to entry for privacy researchers. Effective mechanism design requires not only expertise in DP, but also sufficient familiarity with genomics to understand analysis workflows, identify appropriate locations for the *privacy barrier*, and determine meaningful notions of *utility*. Acquiring this knowledge demands significant time and effort, as relevant concepts are scattered across interdisciplinary literature spanning genetics, statistics, and biomedical data analysis. This knowledge gap has limited broader engagement of the privacy community with genomic data.

At the same time, existing DP research in genomics remains fragmented. Most proposed methods target narrowly defined tasks and are evaluated using heterogeneous metrics and assumptions, making results difficult to compare and hindering the development of standardized practices. Several prior surveys have examined privacy in genomics from broader perspectives, including genome privacy research and privacy-preserving genomic computation more generally (e.g., Aherrahrou et al., 2024; Bonomi et al., 2020; Aziz et al., 2019; Mittos et al., 2017). These works cover heterogeneous privacy paradigms, such as cryptographic methods, secure computation, federated approaches, and DP, often in the context of genomic data sharing, testing, or collaborative genomics. However, they do not provide a systematic framework that simultaneously consolidates the genomic background needed for DP mechanism design and organizes existing DP methods according to their methodological choices, structural assumptions, released quantities, and evaluation perspectives.

This work addresses these gaps by providing a comprehensive and self-contained guide to DP for genomic datasets, with a particular focus on single nucleotide polymorphism (SNP) data. SNPs represent the most common form of genetic variation in the human genome and the primary data modality used in genome-wide association studies (GWAS) Visscher et al. (2017). Our goal is to substantially reduce the effort required for privacy researchers to enter this area by consolidating essential background knowledge on genomic structure, terminology, and analytical workflows, together with curated references. We also provide a systematic overview of existing DP methods for SNP datasets to clarify assumptions, unify evaluation perspectives, and support more comparable future research.

Building on this foundation, we organize existing methods through a release-oriented taxonomy that focuses on *what is released* and under *which assumptions*. For each category, we analyze the modeling choices and limitations that shape algorithm design. Beyond systematizing the literature, this perspective exposes recurring design patterns, implicit trade-offs, and structural mismatches between privacy mechanisms and genomic data properties. As a result, the taxonomy not only clarifies what aspects of genomic data are ultimately protected and enables transparent comparisons across methods, but also serves as a conceptual framework for guiding future research. In particular, it highlights unexplored design spaces, identifies where current approaches fail to account for key biological structures, and suggests principled directions for de-

Table 1: Terms Frequently Used in Genome-Wide Association Studies

**Locus (pl. loci)**
A specific position on a chromosome where a genetic variant occurs.

**Allele**
An alternative form of a genetic variant at a given locus.

**Major allele**
The more frequent allele at a genetic locus within a population.

**Minor allele**
The less frequent allele at a genetic locus within a population.

**Single nucleotide polymorphism (SNP)**
A single-nucleotide variation at a specific genomic locus among individuals.

**Diploid cell**
A cell containing two copies of each chromosome, one from each parent.

**Homologous chromosomes**
A pair of chromosomes (one maternal and one paternal) with the same genes at corresponding loci.

**Haploid cell**
A cell containing one copy of each chromosome.

**Gamete**
A haploid reproductive cell (sperm or egg).

**Homozygous**
Having two identical alleles at a genetic locus.

**Heterozygous**
Having two different alleles at a genetic locus.

**Recombination**
A process in which segments of DNA are exchanged between homologous chromosomes during the formation of reproductive cells, creating new combinations of alleles.

**Haplotype**
A set of genetic variants at multiple loci inherited together from one parent.

**Haplotype block**
Region of strongly correlated variants.

**Linkage disequilibrium (LD)**
The non-random association of alleles across loci.

**Minor allele frequency (MAF)**
The proportion of chromosomes carrying the minor allele at a given locus.

**Genotype**
The set of alleles an individual carries at a given locus.

**Phenotype**
An observable trait or disease outcome.

**Hardy–Weinberg equilibrium (HWE)**
A state in which genotype frequencies are determined by allele frequencies under random mating and absence of evolutionary forces.

**Genetic drift**
Random changes in allele frequencies within a population over time.

**Kinship**
A measure of genetic relatedness between individuals, often defined as the probability of sharing alleles identical by descent.

**Mendelian disease**
A disease caused primarily by variation in a single gene.

**Polygenic disease**
A disease influenced by many genetic variants.

**Association**
A statistical relationship between a genetic variant and a phenotype.

**Effect size**
The estimated magnitude of a variant's influence on a phenotype.

**Polygenic Risk Score (PRS)**
A score estimating genetic predisposition by aggregating effects across multiple variants.

**Genotyping**
The process of determining which variants an individual carries at specific loci.

**Sequencing**
The process of determining the nucleotide sequence of DNA.

**Quality control (QC)**
Procedures used to detect errors and filter unreliable samples or variants prior to analysis.

**Statistical query**
A query that returns an aggregate statistic (e.g., MAF or counts) evaluated at a specific genetic locus in the dataset.

**Beacon query**
A query that returns a binary response indicating whether at least one individual in the dataset carries a specified allele at a given genetic locus.

veloping more realistic and effective privacy-preserving mechanisms. Finally, drawing on these insights, we discuss major open challenges and promising avenues for future work.

## 1.1 Preview of Sections

This paper is organized as follows:

1. **Differential privacy background:** Section 2 introduces the key concepts of DP, focusing on handling correlations in SNP datasets. Section 2.1 presents local DP and discusses challenges arising from correlated loci, while Section 2.2 covers centralized DP mechanisms and methods for handling dependencies among individuals.

2. **Genomics background:** Section 3 provides essential terminology and concepts in genomics. We focus on two major sources of correlation in genomic data: dependencies among loci (Section 3.2.1) and relatedness among individuals (Section 3.2.2), both of which are central to privacy risks and DP design.

3. **GWAS and core tasks:** Section 4 introduces GWAS, the objectives, and standard setups. We review common association testing approaches (Sections 4.3 and 4.4), which form the primary downstream tasks for evaluating DP methods on SNP data.

4. **Structural notions of utility:** Section 5 examines intrinsic structural properties of SNP datasets beyond downstream analyses. This is particularly relevant for DP mechanisms that release sanitized or synthetic data, where preserving biological structures is as important as maintaining task-level utility.

5. **Privacy attacks:** Section 6 provides a high-level overview of major privacy threats in genomic data. We discuss membership inference (MI) attacks (Section 6.1), correlation-based reconstruction attacks (Section 6.2), and kinship-aware inference attacks exploiting both locus and familial dependencies (Section 6.3).

6. **Taxonomy:** Section 7 presents the central contribution of this work: a structured overview of DP methods for SNP datasets organized by their released outputs. Specifically, we classify approaches according to whether they release association statistics (Section 7.1), associated loci (Section 7.2), sanitized datasets (Section 7.3), or preprocessing information used during quality control (Section 7.4). This output-oriented perspective provides a transparent and practical view of what is ultimately protected and how each method can be used in downstream workflows. It also enables a systematic comparison of underlying assumptions, modeling choices, and privacy–utility trade-offs, which is essential for designing realistic DP mechanisms in genomics. Finally, we highlight methods that explicitly address dependencies arising from related individuals in the dataset (Section 7.5).

7. **Future directions and challenges:** Building on the taxonomy developed in Section 7, Section 8 summarizes open challenges revealed throughout this survey. We organize future research directions according to key mismatches between current differential privacy methodologies and the scientific, methodological, and deployment characteristics of genomic data analysis.

Table 1 provides a concise overview of commonly used genomics terminology to support readers unfamiliar with the domain.

## 2 Differential Privacy

Differential privacy (DP) is a rigorous mathematical framework for limiting the information that an adversary can infer about individuals from released data, statistics, or trained models. Rather than preventing data release altogether, DP introduces carefully calibrated randomness to ensure that sensitive information about any individual cannot be inferred with high confidence. The precise interpretation of this guarantee and the form of deniability it provides depend on the underlying trust and data-sharing model.

A central design choice in DP concerns the *trust model*, that is, which entities are assumed to be trusted with access to raw data. This choice determines the level at which privacy protection is enforced and leads to different DP paradigms.

If individuals do not trust any external entity with their raw data, *local differential privacy* (LDP) can be employed, where data are privatized at the individual level before being shared. If a data holder maintains a centralized dataset collected from multiple individuals and aims to release statistics or trained models

without trusting external parties, the setting corresponds to *central differential privacy* (CDP). Finally, when multiple data holders collaborate—most commonly in a federated learning setup—by sharing only intermediate statistics or model updates with a trusted central server, the setting is referred to as *federated differential privacy* (FDP).

In the following subsections, we briefly review these three privacy settings and clarify how plausible deniability is achieved in each case. Since we assume the audience is already familiar with the fundamentals of DP, we omit detailed introductions and instead refer the reader to existing surveys and foundational works. For comprehensive overviews, we recommend Xiong et al. (2020) for LDP, Dwork (2008) for classical CDP, and El Ouadrhiri & Abdelhadi (2022) for deep learning and FDP.

## 2.1 Local Differential Privacy

LDP is one of the earliest models under the umbrella of DP for protecting individual-level data (Evfimievski et al., 2003; Kasiviswanathan et al., 2011). In this setting, privacy is enforced *at the data source*: each individual perturbs their data locally before it is shared with any external party. As a result, no trusted data curator is required.

Formally, an individual's data point $x \in \mathcal{X}$ is locally randomized by a mechanism $\mathcal{A}$ to produce a privatized output $x' = \mathcal{A}(x)$ before any communication occurs. In the LDP setting, deniability stems from uncertainty over the *true value* of the data, that is, an adversary observing $x'$ cannot reliably infer the original input $x$, even when the individual's participation is known.

**Definition 1** ($\varepsilon$-Local Differential Privacy (Evfimievski et al., 2003; Kasiviswanathan et al., 2011))**.** *A randomized mechanism $\mathcal{A}$ satisfies $\varepsilon$-LDP if, for any two input values $x, y$ and any measurable subset $T \subseteq \text{Range}(\mathcal{A})$, the following holds:*

$$\Pr[\mathcal{A}(x) \in T] \leq e^{\varepsilon} \Pr[\mathcal{A}(y) \in T],$$

*where $\varepsilon \geq 0$ is the privacy parameter.*

Smaller values of $\varepsilon$ generally indicate stronger privacy guarantees but introduce higher distortion in the reported data.

One of the most widely used mechanisms for achieving LDP is the *generalized randomized response* (GRR). When the domain of possible values is discrete and relatively small, GRR can be implemented via the *direct encoding* method, which is known to be optimal in this regime (Wang et al., 2017b).

**Definition 2** (Direct Encoding GRR (Kairouz et al., 2016))**.** *Given a domain of possible values $\mathcal{V} = \{v_1, v_2, ..., v_k\}$ and an input $v \in \mathcal{V}$, GRR perturbs $v$ into another value $v' \in \mathcal{V}$ such that:*

$$\Pr[\mathcal{A}(v) = v'] = \begin{cases} p = \frac{e^{\varepsilon}}{e^{\varepsilon}+k-1} & \text{if } v = v' \\ q = \frac{1}{e^{\varepsilon}+k-1} & \text{if } v \neq v' \end{cases}$$

When the goal is to estimate aggregate statistics, such as value frequencies, from data released under LDP, unbiased estimators can be applied to correct for the noise introduced by the mechanism (Wang et al., 2017b). These unbiasing techniques are essential for achieving meaningful utility at scale and are commonly used in practical LDP deployments.

### 2.1.1 Privacy Budget Scaling in Local Differential Privacy

So far, we have treated a user record as a single attribute. We now consider the more general setting in which an individual releases a vector of $L$ attributes, $\mathbf{x} = (x_1, x_2, \ldots, x_L)$. A common approach is to apply an $\varepsilon$-LDP mechanism independently to each coordinate and release the full vector. However, how the overall privacy budget $\varepsilon_{\text{total}}$ scales depends not only on the number of released attributes, but also on how the joint mechanism is constructed and on the dependence structure that can be exploited at the mechanism level.

**Case 1: Coordinate-wise release.** If each coordinate is perturbed independently using an $\varepsilon$-LDP mechanism and all $L$ privatized values are released, the joint mechanism $\mathcal{A}(\mathbf{x}) = (\mathcal{A}(x_1), \ldots, \mathcal{A}(x_L))$ satisfies, for

any two input records $\mathbf{x}, \mathbf{y}$ and any output vector $\mathbf{z}$,

$$\frac{\Pr[\mathcal{A}(\mathbf{x}) = \mathbf{z}]}{\Pr[\mathcal{A}(\mathbf{y}) = \mathbf{z}]} = \prod_{i=1}^{L} \frac{\Pr[\mathcal{A}(x_i) = z_i]}{\Pr[\mathcal{A}(y_i) = z_i]} \leq e^{L\varepsilon}.$$

Thus, the overall privacy budget scales linearly with the dimensionality of the record, $\varepsilon_{\text{total}} = L\varepsilon$ by standard sequential composition (Dwork et al., 2014). This worst-case composition bound holds irrespective of whether the underlying attributes are statistically independent or correlated, and is therefore the default assumption in standard LDP deployments (Arcolezi & Gambs, 2023).

**Case 2: Joint release with fully dependent entries.** At the opposite extreme, suppose the attributes are fully dependent (e.g., $x_1 = x_2 = \cdots = x_L$), and the mechanism is designed to exploit this structure by releasing the entire record through a single randomized transformation. In this case, the joint mechanism can be written as

$$\mathcal{A}(\mathbf{x}) = f(x_1, Z), \quad Z \sim \text{Noise},$$

where all $L$ outputs are deterministically derived from one noisy draw. Although all coordinates change simultaneously when the input changes, the output distribution depends on only one randomized variable. Consequently, for any two possible inputs $\mathbf{x}, \mathbf{y}$ and any output $\mathbf{z}$,

$$\frac{\Pr[\mathcal{A}(\mathbf{x}) = \mathbf{z}]}{\Pr[\mathcal{A}(\mathbf{y}) = \mathbf{z}]} \leq e^{\varepsilon},$$

and the total privacy budget remains $\varepsilon_{\text{total}} = \varepsilon$. This scenario can be interpreted as an effective dimensionality reduction; since the features are perfectly correlated, observing one reveals all others. Consequently, a single randomized perturbation is sufficient to protect the entire record, as additional noise applied to identical entries would not further increase indistinguishability.

**Case 3: Intermediate correlation.** From a mechanism-design perspective, exploiting correlations can reduce the privacy budget required relative to naive coordinate-wise composition. That is, the effective privacy loss may lie between the two extremes discussed above, ranging from the fully dependent case with $\varepsilon_{\text{total}} = \varepsilon$ to the coordinate-wise composition bound $\varepsilon_{\text{total}} = L\varepsilon$. From a privacy–utility perspective, treating all attributes as independent can therefore lead to excessive noise injection. Designing joint mechanisms that exploit correlations intrinsic to the data (rather than instance-specific coincidences) can substantially reduce privacy budget consumption when applying LDP to high-dimensional records (Seeam et al., 2025; Wang et al., 2019; Du et al., 2021). This observation is particularly relevant for structured domains such as genomic data, where strong correlations across features are common (Slatkin, 2008).

## 2.2 Central Differential Privacy (Record-Level)

CDP, often referred to simply as DP, is the original and most widely studied privacy model, under which the foundational mathematical definitions of DP were first introduced (Dwork et al., 2006b). In this setting, a trusted data curator collects a dataset containing records from multiple individuals and aims to release summary statistics or trained models without revealing the underlying raw data.

CDP provides privacy guarantees at the *record level*, ensuring that an external adversary observing the released output cannot confidently infer whether a specific individual's data was included in the dataset. This guarantee holds even if the adversary has full knowledge of all other records in the dataset.

**Definition 3** ($\varepsilon$-Differential Privacy (Dwork et al., 2014))**.** *Let $\mathcal{D}$ denote a domain of datasets and $\mathcal{R}$ denote an output space. A randomized algorithm $\mathcal{A} : \mathcal{D} \to \mathcal{R}$ is $\varepsilon$-DP if, for all neighboring datasets $D, D' \in \mathcal{D}$ and all measurable subsets $S \subseteq \mathcal{R}$,*

$$\Pr[\mathcal{A}(D) \in S] \leq e^{\varepsilon} \Pr[\mathcal{A}(D') \in S], \tag{1}$$

*where $\varepsilon \geq 0$ is the privacy parameter.*

Intuitively, this definition guarantees that the outputs of $\mathcal{A}$ on neighboring datasets $D$ and $D'$ are statistically similar, limiting the extent to which an adversary can infer whether a particular individual's data influenced the result.

The choice of neighboring datasets is central to CDP and directly affects mechanism design and sensitivity analysis. Two common adjacency definitions are used. In the *bounded* model, neighboring datasets differ in the value of exactly one record (*replace-one DP*). In the *unbounded* model, neighboring datasets differ by the addition or removal of a single record (*add-or-remove-one DP*). The choice between these models depends on the application and the associated sensitivity analysis (Ponomareva et al., 2023; Vadhan, 2017; Kairouz et al., 2021).

Many CDP mechanisms rely on bounding the influence of individual records through the concept of *global sensitivity*.

**Definition 4** (Global $L_p$-Sensitivity (Dwork et al., 2006b)). *Let $f : \mathcal{D} \to \mathbb{R}^k$. The global $L_p$-sensitivity of $f$ is defined as*

$$\Delta_p f = \max_{\substack{D,D' \in \mathcal{D} \\ D,D' \text{ neighboring}}} \|f(D) - f(D')\|_p,$$

*where the maximum is taken over all neighboring datasets $D$ and $D'$, and $\|\cdot\|_p$ denotes the $L_p$ norm.*

Bounding the global sensitivity limits how much the output of $f$ can change under the chosen neighboring relation, enabling calibrated noise addition to achieve DP.

A fundamental mechanism for achieving CDP for real-valued queries is the *Laplace mechanism*.

**Definition 5** (Laplace Mechanism (Dwork et al., 2014)). *Let $f : \mathcal{D} \to \mathbb{R}^k$. The Laplace mechanism is defined as*

$$\mathcal{A}(D) = f(D) + (\mathcal{L}_1, \dots, \mathcal{L}_k),$$

*where each $\mathcal{L}_i$ is drawn independently from the Laplace distribution $\mathcal{L}(0, \ell)$, with scale parameter $\ell = \Delta_1 f / \varepsilon$ and density*

$$\mathsf{pdf}_{\mathcal{L}(0,\ell)}(x) = \frac{1}{2\ell} \exp\left(-\frac{|x|}{\ell}\right).$$

*This mechanism satisfies $\varepsilon$-DP.*

In many applications, including GWAS and statistical hypothesis testing, the goal is not to release a numerical statistic directly, but rather to *select* an element from a discrete set of candidates, such as the most significant variant, the best-scoring model, or the most plausible hypothesis. In such settings, adding noise to a numeric output is often inappropriate or even meaningless. Instead, one must define a *utility function* that scores the quality of each candidate output with respect to the dataset, and then select an output in a DP manner based on these scores. A fundamental mechanism for this purpose is the *exponential mechanism*.

**Definition 6** (Utility function and its sensitivity (McSherry & Talwar, 2007)). *Let $\mathcal{D}$ denote the data domain and $\mathcal{R}$ a (possibly discrete) set of candidate outputs. A utility function is a mapping*

$$u : \mathcal{D} \times \mathcal{R} \to \mathbb{R},$$

*which assigns a real-valued score to each pair $(D, r)$ measuring the quality of output $r$ when applied to dataset $D$. The global sensitivity of $u$ is defined as*

$$\Delta u = \max_{r \in \mathcal{R}} \max_{D,D'} \left| u(D, r) - u(D', r) \right|,$$

*where $D$ and $D'$ are neighboring datasets under the chosen adjacency notion.*

**Definition 7** (Exponential Mechanism (McSherry & Talwar, 2007)). *Let $\mathcal{R}$ be a set of possible outputs and let $u : \mathcal{D} \times \mathcal{R} \to \mathbb{R}$ be a utility function with sensitivity $\Delta u$. The exponential mechanism selects an output $r \in \mathcal{R}$ with probability proportional to*

$$\exp\left(\frac{\varepsilon \, u(D, r)}{2\Delta u}\right).$$

*This mechanism satisfies $\varepsilon$-DP.*

Intuitively, the exponential mechanism favors outputs with higher utility while ensuring that the contribution of any single individual to the selection probability is appropriately bounded.

**Relaxations.** Several relaxations of pure DP have also been proposed. The most common is $(\varepsilon, \delta)$-DP (Dwork et al., 2006a), which allows the guarantee in Equation 1 to be violated with probability at most $\delta$. This relaxation is most commonly enforced using the *Gaussian mechanism*, which adds Gaussian noise calibrated to the $L_2$-sensitivity of the output function.

In some settings, computing global sensitivity is difficult or overly conservative. For example, attributes such as age or height may have unbounded or unrealistic worst-case ranges. To address this, alternative notions such as *local sensitivity* and *smooth sensitivity* have been introduced (Nissim et al., 2007). Local sensitivity measures the maximum change in the output for neighboring datasets relative to a fixed dataset, while smooth sensitivity provides a stable upper bound that preserves DP while yielding tighter noise calibration.

**Post-processing.** A fundamental property shared by DP and its standard relaxations is *closure under post-processing* (Dwork et al., 2014). If a randomized mechanism $\mathcal{A}$ satisfies a given privacy guarantee (e.g., $\varepsilon$-DP or $(\varepsilon, \delta)$-DP), then for any (possibly randomized) function $g$ that does not depend on the underlying dataset, the composed mechanism $g \circ \mathcal{A}$ satisfies the same privacy guarantee. Therefore, arbitrary downstream transformations of DP outputs incur no additional privacy loss.

### 2.2.1 Group Differential Privacy

While the previous section discussed correlations among features within a single record in the context of LDP, an analogous issue arises in central DP when multiple records in a dataset are correlated. This setting is particularly relevant for genomic datasets, where records corresponding to related individuals (e.g., family members) exhibit strong genetic similarity.

Group differential privacy (Dwork et al., 2014) extends the standard record-level DP guarantee to protect *groups of up to $R$ records* simultaneously. It captures the worst-case scenario in which multiple correlated records may change together, such as when several entries correspond to the same individual, household, or family.

Let $\mathcal{A}$ be a randomized mechanism satisfying $\varepsilon$-DP under single-record adjacency, and let $\mathcal{D}$ denote the data domain. Then, for any datasets $D, D' \in \mathcal{D}$ differing in at most $R$ records and any measurable output $S \subseteq \text{Range}(\mathcal{A})$,

$$\Pr[\mathcal{A}(D) \in S] \leq e^{R\varepsilon} \Pr[\mathcal{A}(D') \in S].$$

This bound provides a conservative privacy guarantee that remains valid even under worst-case correlations among records within a group. Importantly, group DP is a direct consequence of standard DP through repeated application of the adjacency relation and does not require a separate mechanism design. In practice, however, this worst-case assumption can be overly pessimistic for genomic datasets, where records may exhibit varying degrees of genetic similarity due to shared ancestry or familial relationships. Consequently, group DP may substantially overestimate privacy loss. The role of such correlations in genomic privacy will be revisited in Sections 7.5.

### 2.3 Federated Differential Privacy

Federated learning (FL) (McMahan et al., 2017a; Kairouz & McMahan, 2021) enables multiple data holders to jointly train models without sharing raw data, by exchanging intermediate quantities such as model updates that are aggregated by a central coordinator. This paradigm naturally extends to *federated analytics* (Google AI Blog, 2020), where participants collaboratively compute aggregate statistics rather than train predictive models.

FDP is best viewed as the application of differential privacy in federated settings rather than as a distinct privacy definition. The primary distinction lies in where privacy is enforced within the federated pipeline and which entities are trusted. While numerous algorithmic variants and refinements exist, most approaches can be understood through two primary trust models.

In the first setting, participants do not trust the central server or other parties. Each data holder applies a DP mechanism locally to the updates computed from its dataset before transmission, enforcing privacy prior to communication. In this regime, protection is achieved independently by each participant, eliminating the need for a trusted aggregator but typically requiring stronger noise injection and leading to reduced utility (Wei et al., 2020; Zhu & Ling, 2022).

In the second setting, participants trust the central server to enforce privacy during aggregation. Data holders transmit intermediate updates derived from their local datasets, and DP is applied centrally to the aggregated result. Privacy guarantees are commonly formulated at the participant (user) level, allowing noise to be calibrated to the collective signal and therefore improving utility at the cost of relying on a trusted server (Geyer et al., 2017; McMahan et al., 2017b; Gu et al., 2025).

These settings correspond to different placements of the privacy barrier and represent the dominant trust configurations underlying federated DP systems, with practical deployments often lying between or combining elements of both.

## 3 Genomics and SNPs: Background, Definitions and Properties

In this section, we provide a brief overview of genomics and introduce the necessary terminology. We then focus on the correlation structures observed in SNP datasets.

### 3.1 Genomic Foundations and Core Concepts

A genome comprises the complete genetic information of an organism. This information is encoded in chromosomes as DNA sequences over a four-letter alphabet of nucleotides: Adenine (A), Guanine (G), Cytosine (C), and Thymine (T). Different segments of these sequences serve distinct biological functions. Among the most important are *genes*, which encode instructions required for cellular processes and for the development, function, and maintenance of the organism (Griffiths, 2005).

Humans, like most mammals, are *diploid*, meaning they carry two copies of each chromosome, one inherited from each parent. These paired chromosomes, known as *homologous chromosomes*, contain corresponding genomic regions arranged in largely the same order. Humans typically possess 22 pairs of homologous chromosomes (*autosomes*) and one pair of sex chromosomes. Corresponding genomic positions or regions are referred to as *loci* (singular: *locus*). A locus is generally defined relative to a reference genome coordinate system and may correspond to a single nucleotide position, a sequence interval, or a larger genomic region. Although homologous chromosomes share the same overall structure, their DNA sequences are not necessarily identical. Variation exists both between individuals in a population and between the two chromosome copies carried by a single individual, reflecting the combined effects of mutation, ancestry, and evolutionary history (Griffiths, 2005).

During the formation of reproductive cells (*gametes*), homologous chromosomes undergo *recombination*, a process in which segments from the two parental chromosome copies are exchanged, producing mosaic chromosomes (Low, 2012). Each parent contributes one such recombined chromosome copy to the offspring. Because recombination operates on contiguous chromosomal segments rather than mixing the genome uniformly, neighboring loci are often inherited together across generations (Slatkin, 2008).

### 3.1.1 Genetic Variations

Genomes of different individuals are highly similar, sharing the vast majority of their sequences, but they are not identical. A *genetic variation* refers to any difference in DNA sequence between individuals. Genetic variation occurs at multiple genomic scales (Sharp et al., 2006).

At the smallest scale, *single nucleotide variants* (SNVs) correspond to changes in a single nucleotide at a specific genomic position (International SNP Map Working Group, 2001; Cargill et al., 1999). Slightly larger modifications include *insertions and deletions* (indels), which add or remove short DNA segments within a genomic region, typically ranging from a few to several dozen nucleotide bases (Mills et al., 2006). At an intermediate scale, *short tandem repeats* (STRs) consist of short DNA motifs repeated consecutively, where the

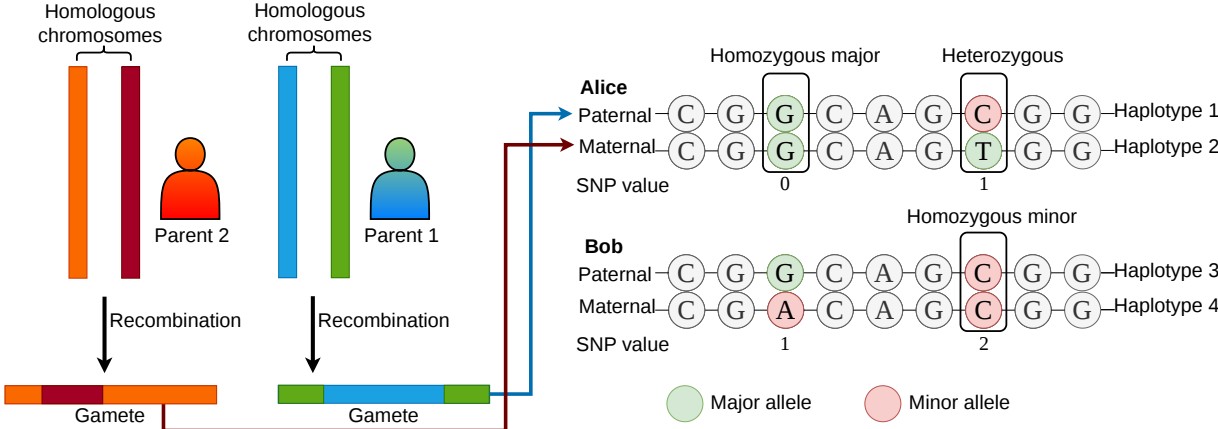

Figure 1: Schematic illustration of the formation of a diploid offspring (Alice) from two diploid parents. Recombination during gamete formation produces mosaic parental chromosomes, of which one gamete from each parent is transmitted to the child.

number of repeat units varies among individuals (Fan & Chu, 2007; Willems et al., 2014). Finally, *structural variants* (SVs) comprise large-scale genomic rearrangements, such as deletions, duplications, inversions, or translocations of DNA segments (Feuk et al., 2006). An important subclass of SVs are *copy number variants* (CNVs), which alter the number of copies of a genomic region in the genome (Redon et al., 2006). Despite their diversity, these variant types are described within a common reference genome coordinate framework, which provides a consistent notion of locus even for variants that span genomic intervals or modify sequence length.

**Single Nucleotide Polymorphisms.** SNVs that occur in a population above a conventional frequency threshold (typically, $1 - 5\%$ (Gibson, 2012)) are referred to as *single nucleotide polymorphisms* (SNPs, pronounced "snips"). For SNPs, a locus corresponds to a specific genomic position at which nucleotide variation is observed in the population. In this work, we focus specifically on SNP variation and, by extension, on SNP datasets, which we use as a general term for representations of SNP data.

The nucleotides observed at a SNP locus across a population are called *alleles*. In principle, any of the four nucleotide types may occur at a genomic position. In practice, however, SNP datasets are typically represented in a simplified biallelic form by retaining only two allelic states at each locus. Among these, the more frequent nucleotide is called the *major allele*, while the less frequent one is called the *minor allele*. Because humans carry two homologous copies of each chromosome, each individual possesses two alleles at every SNP locus. If the two alleles are identical, the individual is said to be *homozygous*; if they differ, the individual is *heterozygous*.

A *haplotype* is the ordered sequence of alleles carried on a single chromosome across multiple loci. In principle, this sequence reflects the underlying DNA sequence of that chromosomal region; in SNP datasets, however, haplotypes are represented only by the allelic states at the selected SNP loci. We provide a visual summary of these concepts in Figure 1.

## 3.2 Correlations and Structures in SNP Data: Linkage and Kinship

We could generally consider two types of correlations in a genomic dataset. Correlations could be either column-wise between loci (features) or row-wise between individuals.

Understanding correlations is crucial for designing privacy-preserving methods. On the one hand, privacy researchers can exploit correlation structures to reduce privacy costs and improve utility (Section 7.5). On the other hand, adversaries can leverage the same structures to mount more effective attacks when such dependencies are not adequately accounted for by the privacy designer (Sections 6.2 and 6.3).

### 3.2.1 Correlation of Loci

SNPs across the genome exhibit complex correlation structures, a phenomenon known as *linkage disequilibrium* (LD) (Lewontin & Kojima, 1960; Slatkin, 2008). As discussed in the previous section, recombination produces mosaic chromosomes during gamete formation, but this mixing is not uniform along the genome. As a result, loci that are physically close on a chromosome are more likely to be inherited together across generations, leading to statistical dependencies between nearby variants. Beyond physical proximity, population-level evolutionary forces further shape LD patterns. *Natural selection* can preserve favorable combinations of alleles, while *genetic drift* can amplify or reduce the frequencies of specific variant combinations in finite populations (Lewontin, 1964). These processes give rise to regions of the genome in which sets of neighboring SNPs exhibit strong mutual correlations. Such regions are commonly referred to as *haplotype blocks*, which denote contiguous genomic segments characterized by high internal LD (Daly et al., 2001; Gabriel et al., 2002). Unlike a *haplotype*, which refers to the specific sequence of alleles carried on a single chromosome, a haplotype block describes a population-level correlation structure indicating that multiple loci within the region tend to be co-inherited.

**Major and minor allele frequencies.** Let us first define the concept of allele frequency within a population. In the SNP datasets considered in this survey, loci are represented using the standard *biallelic* encoding, in which exactly two allelic states are considered at each locus $i$. These correspond to the *major* and *minor* alleles, denoted by uppercase and lowercase letters, respectively. Although a genomic locus may in principle admit more than two allelic variants, SNP datasets are commonly represented in a biallelic form, with multi-allelic sites often decomposed or filtered during preprocessing. This representation is also reflected in widely used formats such as the Variant Call Format (VCF) (Danecek et al., 2011). Under this representation, the two encoded allelic states exhaust the possible alleles at locus $i$. We denote by $f_i(A)$ the population frequency of allele $A$ at locus $i$. Based on these definitions, the frequency of the major allele $A$ at position $i$ in a population of $N$ diploid individuals is given by:

$$f_i(A) = \frac{1 \times n_i^{Aa} + 2 \times n_i^{AA}}{2N},$$

where $n_i^{Aa}$ denotes the number of individuals with one major allele at locus $i$ (heterozygous), $n_i^{AA}$ the number of individuals with two major alleles (homozygous major), and $2N$ represents the total number of alleles in the population. Similarly, the frequency of the minor allele $a$ is defined as:

$$f_i(a) = \frac{1 \times n_i^{Aa} + 2 \times n_i^{aa}}{2N}, \tag{2}$$

where $n_i^{aa}$ is the number of individuals with two minor alleles at locus $i$ (homozygous minor). For each locus $i$, the frequencies satisfy the relationships $f_i(a) + f_i(A) = 1$ and $0 \leq f_i(a), f_i(A) \leq 1$. Minor allele frequency is commonly abbreviated as **MAF** in the genomic literature.

**Linkage disequilibrium.** Consider allele $A$ at locus $i$ with frequency $f_i(A)$ and allele $B$ at locus $j$ with frequency $f_j(B)$. If these alleles are statistically independent within the population, then by definition

$$f_{i,j}(AB) = f_i(A)f_j(B),$$

where $f_{i,j}(AB)$ denotes the joint probability of observing both alleles $A$ and $B$ simultaneously at loci $i$ and $j$, respectively.

The *coefficient of linkage disequilibrium* (Lewontin & Kojima, 1960) between loci $i$ and $j$ is defined as

$$D_{i,j}^{A,B} = f_{i,j}(AB) - f_i(A)f_j(B),$$

which measures the deviation from statistical independence. Equivalently, if the presence of alleles $A$ and $B$ is encoded by Bernoulli random variables, $D_{i,j}^{A,B}$ corresponds to their covariance. A nonzero $D$ indicates the presence of LD.

Because $D$ depends on allele frequencies, it is common to normalize it to obtain a scale-free measure. One widely used normalization expresses LD as a Pearson correlation coefficient by dividing this covariance by the product of the corresponding standard deviations (Wright, 1933; Hill & Robertson, 1968):

$$r_{i,j}^{A,B} \;=\; \frac{D_{i,j}^{A,B}}{\sqrt{f_i(A)\big(1 - f_i(A)\big)f_j(B)\big(1 - f_j(B)\big)}},$$

$$\text{or} \quad \left(r_{i,j}^{A,B}\right)^2 \;=\; \frac{\left(D_{i,j}^{A,B}\right)^2}{f_i(A)\big(1 - f_i(A)\big)f_j(B)\big(1 - f_j(B)\big)}. \tag{3}$$

The statistic $r^2$, known as the *squared correlation coefficient*, is one of the most commonly used measures of LD between pairs of loci (Hill & Robertson, 1968). Its values range from $0 \leq r^2 \leq 1$, where $r^2 = 0$ denotes no statistical association, and $r^2 = 1$ indicates perfect positive or negative correlation between the two loci. It is important to note that multiple normalization schemes for $D$ exist, each with distinct advantages and limitations (Kang & Rosenberg, 2020). Pairwise LD values are commonly visualized from the binary major/minor allele perspective, using heatmaps that summarize the strength of association across all locus pairs under consideration.

**Hardy–Weinberg equilibrium (HWE).** In addition to statistical dependencies between loci, it is also important to consider statistical structure at the level of a single locus. In particular, one may ask whether the two alleles carried by an individual at the same genomic position behave as independent samples from the population. HWE (Hardy, 1908; Weinberg, 1908) formalizes the idealized situation in which this is the case, and serves as a fundamental baseline model in population genetics.

Consider a biallelic locus $i$ with major allele $A$ and minor allele $a$, with corresponding population frequencies $f_i(A)$ and $1 - f_i(A)$. Under HWE, the two homologous chromosome copies carried by an individual are assumed to be sampled independently from the population allele distribution. As a consequence, the probabilities of the possible allelic configurations at locus $i$ factorize as

$$f_i(AA) = f_i(A)^2, \qquad f_i(Aa) = 2f_i(A)\big(1 - f_i(A)\big), \qquad f_i(aa) = \big(1 - f_i(A)\big)^2.$$

Thus, HWE expresses the absence of statistical dependence between the two alleles at the same locus. Conceptually, this is directly analogous to linkage equilibrium, which expresses the absence of statistical dependence between alleles at different loci. Together, these two notions characterize independence both *within* loci and *across* loci.

### 3.2.2 Correlations of Data Points

As explained in Section 3.1, each diploid individual inherits one chromosome from each parent, forming a homologous chromosome pair. Given the allelic composition carried by the parents at a given locus, classical Mendelian inheritance specifies the probability distribution over the allelic configurations of the child at that locus (Mendel, 1866). These inheritance rules are inherently probabilistic and describe allele transmission at the level of individual loci.

**Child from parents.** Consider a specific locus $i$. A diploid parent carries two alleles at this position, which together form one of the possible allelic configurations $\{AA, Aa, aa\}$. Given the allelic configurations of both parents (P1 and P2), Mendelian inheritance determines the probabilities of observing each possible allelic configuration in the child, as illustrated in Figure 2. Note that these inheritance events are not independent across loci. As discussed in the previous section, LD induces correlations between alleles at different genomic positions. For example, if allele $A$ at locus $i$ is strongly correlated with allele $B$ at locus $j$ in the population, then observing allele $A$ in the child at locus $i$ increases the probability of also observing allele $B$ at locus $j$, provided that this combination is supported by the parental haplotypes, namely, that at least one parent carries the haplotype $AB$.

| P1 / P2 | AA | Aa | aa |
|---|---|---|---|
| **AA** | AA 1.0 | (AA, Aa) (0.5, 0.5) | Aa 1.0 |
| **Aa** | (AA, Aa) (0.5, 0.5) | (AA, Aa, aa) (0.25, 0.5, 0.25) | (Aa, aa) (0.5, 0.5) |
| **aa** | Aa 1.0 | (Aa, aa) (0.5, 0.5) | aa 1.0 |

Figure 2: Mendelian probabilities for the allelic configurations $\{AA, Aa, aa\}$ carried by a child, given the allelic configurations of parent 1 (P1) and parent 2 (P2) at the same locus. For each possible configuration in the child, the corresponding probability is indicated.

**Parents from child.** In a similar manner, information can also flow in the opposite direction, from child to parents. Given the allelic configuration carried by a child at a locus and the allelic configuration of one parent, Bayes' rule can be used to infer the posterior distribution over the possible allelic configurations of the other parent. Consider a case in which the child carries configuration $aa$ at locus $i$, and parent 1 (P1) carries configuration $Aa$ at the same locus. From Figure 2, only two configurations for parent 2 (P2) remain possible. Specifically, we have

$$\Pr(\text{child} = aa \mid \text{P1} = Aa, \text{P2} = AA) = 0,$$
$$\Pr(\text{child} = aa \mid \text{P1} = Aa, \text{P2} = Aa) = 0.25,$$
$$\Pr(\text{child} = aa \mid \text{P1} = Aa, \text{P2} = aa) = 0.5.$$

To compute the posterior probabilities for the allelic configuration of parent 2, we apply Bayes' theorem:

$$\Pr(\text{P2} \mid \text{child} = aa, \text{P1} = Aa) = \frac{\Pr(\text{child} = aa \mid \text{P1} = Aa, \text{P2}) \Pr(\text{P2})}{\sum_{C \in \{AA, Aa, aa\}} \Pr(\text{child} = aa \mid \text{P1} = Aa, C) \Pr(C)}.$$

This computation requires prior probabilities over the allelic configurations $\Pr(AA)$, $\Pr(Aa)$, and $\Pr(aa)$, which may be taken to reflect their population frequencies. For illustration, suppose a uniform prior is assumed, i.e., $\Pr(AA) = \Pr(Aa) = \Pr(aa) = 1/3 = 0.\overline{3}$. Then:

$$\Pr(\text{P2} = AA \mid \text{child} = aa, \text{P1} = Aa) = \frac{0}{0.25 \times 0.\overline{3} + 0.5 \times 0.\overline{3}} = 0,$$
$$\Pr(\text{P2} = Aa \mid \text{child} = aa, \text{P1} = Aa) = \frac{0.25 \times 0.\overline{3}}{0.25 \times 0.\overline{3} + 0.5 \times 0.\overline{3}} = 0.\overline{3},$$
$$\Pr(\text{P2} = aa \mid \text{child} = aa, \text{P1} = Aa) = \frac{0.5 \times 0.\overline{3}}{0.25 \times 0.\overline{3} + 0.5 \times 0.\overline{3}} = 0.\overline{6}.$$

In this example, under the assumed priors, it is therefore twice as likely that parent 2 carries configuration $aa$ rather than $Aa$.

Such correlations among related individuals are strongest for first-degree relatives, such as parent–offspring pairs and full siblings, and they extend to more distant relationships, with their strength decreasing as individuals become less closely related (Hartl et al., 1997). Interestingly, when mating is not random and partners are selected based on specific traits, related individuals may exhibit genetic correlations that deviate from those predicted by simple Mendelian inheritance alone (Nagylaki, 1978; Sunde et al., 2024).

**Extreme familial correlations.** Certain parts of the genome exhibit extreme forms of correlation, amounting to near-exact copying from parent to offspring. Two prominent cases are mitochondrial DNA (mtDNA) and the Y chromosome. mtDNA is inherited almost exclusively from the mother, meaning that children typically receive an essentially identical mitochondrial genome (Giles et al., 1980; Wei & Chinnery, 2020). Likewise, the Y chromosome is transmitted from father to son with only minimal changes across generations, resulting in sons carrying nearly the same Y-chromosomal sequence as their fathers Lippold et al. (2014); Jobling & Tyler-Smith (2003). These inheritance patterns create almost deterministic parent–offspring correlations and represent extreme cases in which genetic material is passed on largely intact rather than through the more variable processes observed elsewhere in the genome.

## 4 Genome-wide Association Studies and Their Core Analytical Tasks

This section reviews the core study designs and association testing paradigms that underpin GWAS. In the context of DP, these methods constitute the primary downstream tasks whose fidelity must be assessed after privatization. Because privacy mechanisms may perturb genetic measurements, traits, or intermediate statistical quantities, understanding how association analyses operate, including their assumptions, dependence on study design, and handling of potential confounding factors, is essential for interpreting utility.

We therefore summarize the principal GWAS study designs together with commonly used approaches for detecting and testing statistical associations between genetic variants and traits, ranging from simple counting-based methods to more structured statistical models. Together, these frameworks form the analytical foundation against which privacy-preserving mechanisms are evaluated. An overview of the core GWAS analysis workflow is provided in Figure 3.

### 4.1 Genome-Wide Association Studies

GWAS (Uffelmann et al., 2021) aim to identify statistical associations between genetic variation and observable traits or diseases, collectively referred to as *phenotypes*. The central idea of GWAS is to analyze genetic data from large cohorts of individuals and test whether variation at specific genomic locations is correlated with variation in phenotype. Because phenotypes arise from the combined influence of genetic, environmental, and stochastic factors, GWAS do not seek deterministic rules but rather quantify average statistical associations between genetic variation and phenotypes across populations.

Modern GWAS primarily follow an *indirect* strategy (Kruglyak, 2008). Instead of exhaustively identifying all functional variants, GWAS profile a dense set of common SNPs across the genome and test each SNP for association with the phenotype. Owing to LD, associated SNPs can serve as proxies for nearby functional or candidate causal variants, making it possible to localize genomic regions associated with the trait without directly identifying the underlying functional mutation. This reliance on common variation motivates the use of SNPs, rather than rare single-nucleotide variants, as GWAS requires variants with sufficient population frequency to enable reliable statistical testing.

At each SNP locus, an individual carries two alleles, one inherited from each parent. The ordered pair of alleles at a locus is referred to as the *genotype*. In GWAS, genotypes are typically encoded numerically by counting the number of minor alleles present, that is, $g \in \{0, 1, 2\}$; where $g = 0$ corresponds to two major alleles, $g = 1$ to one major and one minor allele, and $g = 2$ to two minor alleles. This encoding yields a simple representation that integrates naturally into statistical models relating genetic variation to phenotypic outcomes.

**Mendelian vs polygenic disease.** The interpretability of GWAS findings depends strongly on the genetic architecture of the phenotype under study. *Mendelian* diseases are typically associated with rare variants of large phenotypic effect, and exhibit near-deterministic inheritance, although carrying a causal variant does not always result in the corresponding phenotype (Strome et al., 2024). In contrast, most traits investigated in GWAS are *polygenic*, meaning that they are influenced by many genetic variants, each accounting for a small proportion of phenotypic variation (Boyle et al., 2017). In such settings, GWAS

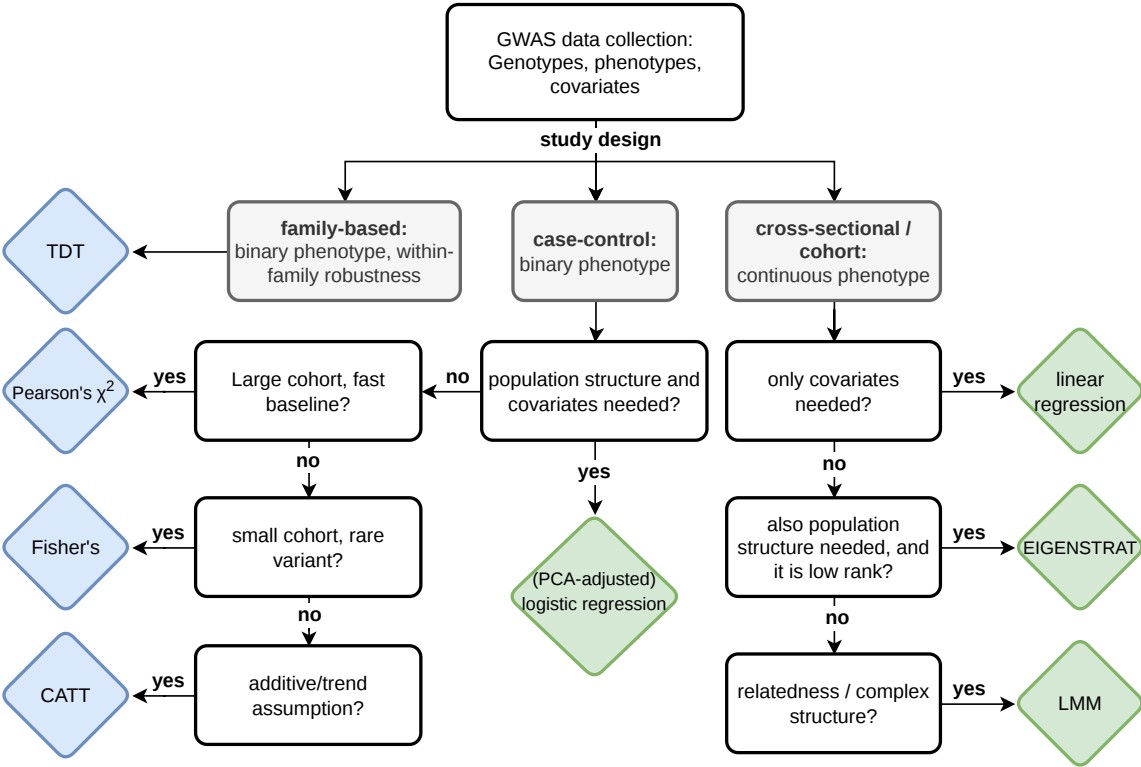

Figure 3: Overview of GWAS association tests considered in this survey. The flowchart illustrates how method selection depends on study design (family-based vs. unrelated individuals), phenotype type (binary or continuous), and the need to account for covariates, population structure, or relatedness. Classical table-based association tests (Section 4.3) are shown in blue and model-based association tests (Section 4.4) in green. Abbreviations appearing in the figure include TDT (Transmission Disequilibrium Test), CATT (Cochran–Armitage Trend Test), and LMM (Linear Mixed Model).

identify loci with statistically detectable contributions rather than single causal determinants, reflecting the inherently probabilistic relationship between genetic variation and complex traits (Visscher et al., 2017).

## 4.2 Study Design

Before delving into the core tasks, we first outline the major study designs commonly used in GWAS. In this context, study design refers to how participants are selected, grouped, and observed. The choice of design influences statistical power, potential biases, and the interpretability of results.

Below, we briefly review the principal GWAS study designs; interested readers are referred to Pearson & Manolio (2008) for a more detailed discussion.

**Case–control study.** In a case–control study, two groups of individuals are compared: the case group, which exhibits a particular trait (e.g., a disease or phenotype), and the control group, which does not. Both groups are genotyped, and association analyses aim to detect statistical differences in SNP frequencies between them. This design is particularly vulnerable to confounding from population substructure, where systematic genetic differences across subpopulations can lead to spurious associations unrelated to the phenotype (Thomas & Witte, 2002). As a result, appropriate population stratification is a critical preprocessing step. Another potential source of bias arises from case ascertainment, where the selected cases may not

represent the full disease spectrum, potentially excluding fatal, mild, or silent presentations (Manolio et al., 2006).

**Family-based study.** Family-based designs, most notably the trio design, include affected offspring along with both biological parents. In contrast to case–control studies, phenotypic assessment is performed exclusively in the offspring. The primary objective is to test whether specific alleles are transmitted to affected offspring more frequently than expected under Mendelian inheritance, thereby identifying SNPs associated with the trait of interest. This design is inherently robust to population stratification bias that can affect case–control studies (Spielman et al., 1993; Laird & Lange, 2006). However, it is highly sensitive to genotyping errors, and assembling sufficiently large datasets can be challenging, particularly for diseases with a later age of onset (Mitchell et al., 2003).

**Cohort study.** Cohort studies recruit a large group of individuals, collect genotype data, and follow participants over time to assess the incidence of disease or changes in other traits. Although these studies are more costly and time-consuming to conduct, they provide a more representative sample of the population and capture a wide range of health-related characteristics (Wijmenga & Zhernakova, 2018).

**Cross-sectional study.** Cross-sectional studies collect genotype and phenotype data from a large number of individuals at a single point in time, providing a snapshot of trait or disease prevalence within the population. Similar to cohort studies, they aim to include a broad and representative sample; however, unlike cohort designs, they do not involve longitudinal follow-up (Wijmenga & Zhernakova, 2018). Compared to case–control studies, which typically sample based on phenotype status, cross-sectional studies recruit participants without regard to outcome, enabling analysis of both binary and quantitative traits in a single dataset.

### 4.3 Classical (Table-Based) Association Tests

Classical association tests constitute some of the earliest and most widely used statistical methods in GWAS. These approaches perform *SNP-wise* hypothesis testing using simple contingency tables that compare observed genotype or allele counts across phenotype categories. They are primarily applicable to *binary phenotypes* and rely on counting-based statistics rather than explicit phenotype models.

These methods typically do not incorporate population structure, relatedness, or additional covariates beyond the variant under consideration. As a result, they are computationally efficient, conceptually transparent, historically important, and they continue to serve as standard baselines for GWAS methodology. While originally developed for case–control settings, analogous table-based formulations exist for family-based designs through within-family transmission comparisons.

In this section, we review representative classical association tests, organized by whether association is assessed across unrelated individuals or within families.

#### 4.3.1 Case–Control Table-Based Association Tests

Case–control table-based association tests evaluate association by comparing genotype or allele distributions between affected and unaffected individuals at a given variant. Test statistics are derived from contingency tables whose structure depends on how genotype information is encoded and on the assumed genetic model.

Different tests within this class arise from alternative representations of the genotype–phenotype relationship. In particular, association may be assessed using full genotypic tables that allow for arbitrary differences between cases and controls, or using reduced representations that impose additive or trend-based constraints across genotype categories. These modeling choices lead to tests with different degrees of freedom, power characteristics, and robustness properties.

**Pearson's $\chi^2$ test.** Pearson's $\chi^2$ test provides the most general table-based framework for testing association between a single genetic variant and a binary phenotype (Stigler, 2002). At each SNP, the null hypothesis

states that genotype frequencies are identical between cases and controls, or equivalently, that genetic variation at the locus is statistically independent of phenotype status in the population. No assumptions are imposed on the underlying genetic model beyond this independence assumption.

For a biallelic SNP, the most direct representation of this hypothesis is given by the *genotypic contingency table* shown in Table 2. The table records the number of cases and controls carrying each genotype, where $r_g$ and $s_g$ denote the counts of genotype $g \in \{0, 1, 2\}$ among cases and controls, respectively. The row totals $n_g = r_g + s_g$ correspond to marginal genotype frequencies, while the column totals $R$ and $S$ denote the total numbers of cases and controls, with $N = R + S$.

Table 2: Genotypic contingency table for a biallelic SNP in a case–control study.

| Genotype | Cases | Controls | Row total |
|---|---|---|---|
| 0 | $r_0$ | $s_0$ | $n_0$ |
| 1 | $r_1$ | $s_1$ | $n_1$ |
| 2 | $r_2$ | $s_2$ | $n_2$ |
| **Column total** | $R$ | $S$ | $N$ |

Let $c \in \{\text{Cases}, \text{Controls}\}$ denote the phenotype class; under the null hypothesis of independence, the expected counts $E_{gc}$ are given by the product of the corresponding row and column marginals divided by the total sample size $N$. Letting $O_{gc}$ denote the observed cell counts, the Pearson test statistic is defined as

$$\chi^2 = \sum_g \sum_c \frac{(O_{gc} - E_{gc})^2}{E_{gc}},$$

which asymptotically follows a chi-squared distribution with two degrees of freedom. This formulation allows for arbitrary differences in genotype distributions between cases and controls and therefore serves as the most general association test in the absence of further genetic assumptions.

An equivalent but lower-dimensional representation is obtained by collapsing genotypes into allele counts, yielding the *allelic contingency table* shown in Table 3. Here, the same data are viewed from an allele-frequency perspective; that is, each genotype contributes two alleles, and the table compares the total number of minor and major alleles observed among cases and controls. The resulting $2 \times 2$ table leads to a one-degree-of-freedom $\chi^2$ statistic.

Table 3: Allelic contingency table for a biallelic SNP derived from the genotypic representation.

| Allele type | Cases | Controls | Row total |
|---|---|---|---|
| Minor | $r_1 + 2r_2$ | $s_1 + 2s_2$ | $n_1 + 2n_2$ |
| Major | $2r_0 + r_1$ | $2s_0 + s_1$ | $2n_0 + n_1$ |
| **Column total** | $2R$ | $2S$ | $2N$ |

The genotypic and allelic formulations test the same underlying null hypothesis but differ in perspective and degrees of freedom. The genotypic table treats genotypes as distinct categories and remains fully general, whereas the allelic table implicitly imposes an additive genetic model by aggregating genotype information. In both cases, the resulting test statistic is compared to the appropriate chi-squared distribution to obtain a $p-$value.

For completeness, when the case–control design is balanced, the $\chi^2$ statistics admit simple closed-form expressions in terms of the table entries. These expressions are algebraically convenient but do not alter the underlying hypothesis or inferential interpretation.

**Fisher's exact test.** Fisher's exact test (Fisher, 1922) provides an inferential alternative to Pearson's $\chi^2$ test for assessing association in $2 \times 2$ contingency tables. While Pearson's test relies on a large-sample approximation to the null distribution of the test statistic, Fisher's exact test computes inference directly from the exact sampling distribution of the table under the null hypothesis. It is therefore particularly useful

in settings where expected cell counts are small or highly unbalanced, conditions under which the chi-squared approximation may be inaccurate.

The null hypothesis tested by Fisher's exact test is the same as in the allelic formulation of Pearson's $\chi^2$ test, namely, that allele counts are independent of phenotype status or, equivalently, that the probability of observing a given allele is identical in cases and controls. Unlike Pearson's test, however, Fisher's approach conditions on the observed row and column totals of the contingency table and evaluates the probability of the observed configuration under this conditioning.

Consider the $2 \times 2$ allelic contingency table

|  | **Cases** | **Controls** |
|---|---|---|
| Minor allele | $a$ | $b$ |
| Major allele | $c$ | $d$ |

where the margins $a+b$, $c+d$, $a+c$, and $b+d$ are treated as *fixed*. Under the null hypothesis of independence, the probability of observing the table with entry $a$ is given by the hypergeometric distribution

$$P(a \mid a+b, a+c, 2N) = \frac{\binom{a+b}{a}\binom{c+d}{c}}{\binom{2N}{a+c}},$$

where $2N = a + b + c + d$ denotes the total number of alleles.

The $p-$value is obtained by enumerating all contingency tables with the same *fixed row and column totals* as the observed table and summing the probabilities, under the null hypothesis, of those tables that exhibit at least as strong a deviation from independence as the observed configuration. The need to enumerate an increasingly large number of admissible tables as the sample size grows leads to a rapidly increasing computational cost, and in large-scale GWAS, the test is therefore typically applied selectively, for example, in the analysis of rare variants or as a robustness check alongside Pearson's $\chi^2$ test (Wang et al., 2011; Ludbrook, 2008).

**Cochran–Armitage trend test.** The Cochran–Armitage trend test (CATT) (Cochran, 1954; Armitage, 1955) introduces additional genetic structure into the general Pearson $\chi^2$ framework by explicitly assuming an additive genetic model for the genotype–phenotype association. Where Pearson's genotypic test allows for arbitrary differences in genotype distributions between cases and controls, the CATT is designed for situations in which the probability of case status is expected to vary monotonically with the number of minor alleles carried by an individual. This means that, for example, genotype value 2 is assumed to have a stronger association with the phenotype than genotype value 1. When this additive assumption is appropriate, the trend test offers increased statistical power relative to the unrestricted Pearson test.

The null hypothesis tested by the CATT states that there is no linear trend in the probability of case status across ordered genotype categories or equivalently, that the probability of being a case does not depend on the number of minor alleles. Under the alternative hypothesis, genotype categories are ordered according to allele count, and association is reflected by a systematic shift in genotype frequencies between cases and controls along this ordering.

Consider the genotypic contingency table in Table 2 with $R$ cases and $S$ controls. To encode the additive structure, numerical scores $w_i \in \{0, 1, 2\}$ are assigned to genotypes according to minor allele count. The test statistic is constructed by comparing the observed weighted genotype counts in cases to those expected under the null hypothesis,

$$T = \sum_{i=0}^{2} w_i \left( r_i - \frac{R}{N} n_i \right) = \left( r_1 - \frac{R}{N} n_1 \right) + 2 \left( r_2 - \frac{R}{N} n_2 \right),$$

where $r_i$ and $n_i$ denote the case counts and marginal counts for genotype $i$, respectively.

The CATT is reported using the standardized statistic

$$\chi^2_{\text{trend}} = \frac{T^2}{\text{Var}(T)},$$

which asymptotically follows a chi-squared distribution with one degree of freedom under the null hypothesis, reflecting the single additive trend parameter being tested.

Under HWE (see Section 3.2.1), the CATT is asymptotically equivalent to the allelic Pearson $\chi^2$ test introduced above, as both reduce to testing for differences in allele frequencies between cases and controls using the allelic contingency table in Table 3. In the absence of this equilibrium assumption, however, the trend test remains well-defined at the genotype level and provides a principled one-degree-of-freedom alternative to the more general genotypic Pearson test.

### 4.3.2 Family-Based Table-Based Association Tests

Family-based table-based association tests assess association by examining transmission patterns within related individuals, most commonly nuclear families. Instead of comparing genotype frequencies across unrelated cases and controls, these methods construct contingency tables based on transmitted and non-transmitted alleles conditional on parental genotypes.

A defining feature of family-based tests is that inference is based entirely on within-family comparisons. As a consequence, these methods are robust to confounding due to population stratification or other forms of between-family structure (Laird & Lange, 2006; Spielman et al., 1993). Under the null hypothesis of no association, alleles are transmitted from parents to offspring according to Mendelian expectations. Systematic deviations from these expectations provide evidence for association. In this subsection, we focus on the transmission disequilibrium test (TDT) as a representative example.

**Transmission disequilibrium test.** The TDT (Spielman et al., 1993) is a family-based association test originally developed for nuclear families consisting of two parents and one *affected offspring*. The test evaluates whether a particular allele is transmitted from heterozygous parents to affected offspring more frequently than expected under Mendelian inheritance.

The null hypothesis of the TDT states that, conditional on parental genotypes, each allele carried by a heterozygous parent is transmitted to the affected offspring with probability 0.5. Equivalently, there is no association between genotype and phenotype, and no LD between the marker and disease locus. The alternative hypothesis corresponds to preferential transmission of one allele, indicating association in the presence of linkage.

At a given SNP, let $A$ denote the major allele and $a$ the minor allele. For each trio family, transmitted and non-transmitted alleles can be identified by comparing parental and offspring genotypes. Only transmissions from heterozygous parents are informative, as homozygous parents contribute no information about transmission bias. Table 4 enumerates the possible transmission configurations for a single trio at one SNP.

When data from $N$ independent trio families are aggregated, the resulting transmission counts can be summarized in the $2 \times 2$ contingency table shown in Table 5. Let $b$ denote the number of times allele $A$ is transmitted while allele $a$ is not transmitted, and let $c$ denote the number of times allele $a$ is transmitted while allele $A$ is not transmitted. The TDT statistic is defined as

$$T(b, c) = \frac{(b - c)^2}{b + c},\tag{4}$$

with the convention that $T = 0$ when $b + c = 0$.

Under the null hypothesis and assuming independence across families, the count $b$ follows a binomial distribution with $b + c$ trials and success probability of 0.5. Consequently, the test statistic in Equation 4 asymptotically follows a chi-squared distribution with one degree of freedom. The test therefore provides evidence for association when systematic deviations from equal transmission are observed within families.

### 4.4 Model-Based (Explicit Phenotype) Association Tests

Model-based association tests form the foundation of modern genome-wide association analyses by explicitly modeling the relationship between genetic variants, phenotypes, and covariates within a unified statistical

Table 4: All possible TDT contingency tables for one trio-family at one SNP

| Trio Genotype parent 1 × parent 2 → child | | Non-transmitted $A$ | Non-transmitted $a$ |
|---|---|---|---|
| $AA \times AA \to AA$ | Transmitted $A$ | 2 | 0 |
| | Transmitted $a$ | 0 | 0 |
| $aa \times aa \to aa$ | Transmitted $A$ | 0 | 0 |
| | Transmitted $a$ | 0 | 2 |
| $AA \times aa \to Aa$ | Transmitted $A$ | 1 | 0 |
| | Transmitted $a$ | 0 | 1 |
| $AA \times Aa \to AA$ | Transmitted $A$ | 1 | 1 |
| | Transmitted $a$ | 0 | 0 |
| $AA \times Aa \to Aa$ | Transmitted $A$ | 1 | 0 |
| | Transmitted $a$ | 1 | 0 |
| $Aa \times aa \to Aa$ | Transmitted $A$ | 0 | 1 |
| | Transmitted $a$ | 0 | 1 |
| $Aa \times aa \to aa$ | Transmitted $A$ | 0 | 0 |
| | Transmitted $a$ | 1 | 1 |
| $Aa \times Aa \to AA$ | Transmitted $A$ | 0 | 2 |
| | Transmitted $a$ | 0 | 0 |
| $Aa \times Aa \to Aa$ | Transmitted $A$ | 0 | 1 |
| | Transmitted $a$ | 1 | 0 |
| $Aa \times Aa \to aa$ | Transmitted $A$ | 0 | 0 |
| | Transmitted $a$ | 2 | 0 |

Table 5: TDT contingency table from $N$ trio families at one SNP

| | Non-transmitted $A$ | Non-transmitted $a$ | **Row total** |
|---|---|---|---|
| Transmitted $A$ | $e$ | $b$ | $e + b$ |
| Transmitted $a$ | $c$ | $d$ | $c + d$ |
| **Column total** | $e + c$ | $b + d$ | $2N$ |

framework. These approaches replace contingency tables with regression-based models, enabling flexible modeling of genotype–phenotype associations and principled adjustment for confounding factors such as population structure, relatedness, and environmental covariates.

Owing to their ability to accommodate complex covariate structures while maintaining strong statistical power and false positive control, model-based approaches are widely used in contemporary GWAS (Uffelmann et al., 2021). Subsequent subsections review commonly used model-based formulations and their extensions.

### 4.4.1 Marginal Regression Models

Regression-based association tests extend beyond classical contingency-table methods by explicitly modeling genotype–phenotype associations while incorporating covariates. In this context, *marginal* refers to modeling the association of a single SNP with the phenotype while treating all other genetic contributions as unmodeled noise, apart from explicitly included covariates such as age, sex, or population structure.

Although multivariate regression may appear natural, it is generally avoided in GWAS for two reasons. First, LD induces strong multicollinearity among nearby SNPs, rendering joint regression unstable and difficult to interpret. Second, the number of SNPs ($m \sim 10^{6-7}$) typically far exceeds the number of individuals ($n \sim 10^{5-6}$), making full multivariate regression statistically ill-posed. As a result, genome-wide association analyses predominantly adopt a marginal testing paradigm, fitting a separate regression model for each

SNP (Uffelmann et al., 2021; Bush & Moore, 2012). This approach preserves interpretability, computational tractability, and well-behaved inference.

**Linear regression for continuous phenotypes.** For continuous traits, a standard linear regression model is used. Consider a study with $n$ individuals and a single SNP $j$ coded as $\mathbf{g}_j \in \{0, 1, 2\}^n$. Let $\mathbf{C} \in \mathbb{R}^{n \times K}$ denote a matrix of $K$ covariates such as age or sex. The linear model for phenotype vector $\mathbf{y} \in \mathbb{R}^n$ is

$$\mathbf{y} = \mathbf{1}\beta_0 + \mathbf{g}_j\beta_j + \mathbf{C}\boldsymbol{\gamma} + \boldsymbol{\epsilon},$$

where $\mathbf{1} \in \mathbb{R}^n$ denotes the vector of ones, $\beta_0$ is the intercept, $\boldsymbol{\gamma} \in \mathbb{R}^K$ are covariate coefficients, and $\boldsymbol{\epsilon}$ represents random noise. Here, $\beta_j$ denotes the SNP regression coefficient, commonly referred to as the SNP's *effect size*, which quantifies the strength and direction of association between the SNP and phenotype. In the absence of covariates, we can drop the third term and set $\boldsymbol{\gamma} = 0$. Parameter estimates are obtained by minimizing the least-squares objective.

To determine whether SNP $j$ is associated with the phenotype, hypothesis testing is applied to the estimated effect size:

$$H_0 : \beta_j = 0 \qquad \text{vs.} \qquad H_1 : \beta_j \neq 0.$$

In practice, significance is assessed using a $t$-test for the coefficient or an $F$-test comparing nested models. A significant non-zero estimate $\hat{\beta}_j$ indicates evidence of association.

**Logistic regression for binary phenotypes.** For binary traits, $\mathbf{y} \in \{0, 1\}^n$, logistic regression models the phenotype probability using a sigmoid function applied to a linear predictor. Let $\boldsymbol{\eta} = \mathbf{1}\beta_0 + \mathbf{g}_j\beta_j + \mathbf{C}\boldsymbol{\gamma}$ denote the linear predictor for SNP $j$. The phenotype is modeled as

$$\Pr(\mathbf{y} = 1 \mid \mathbf{g}_j, \mathbf{C}) = \sigma(\boldsymbol{\eta}),$$

where $\sigma(z) = 1/(1 + e^{-z})$ is applied componentwise, yielding one success probability per individual. Parameters are estimated by maximum likelihood under a Bernoulli model; unlike linear regression, stochasticity is modeled through the likelihood rather than an additive noise term. Association testing proceeds via

$$H_0 : \beta_j = 0 \qquad \text{vs.} \qquad H_1 : \beta_j \neq 0,$$

with significance typically assessed using the Wald test (Bewick et al., 2005), though likelihood-ratio and score tests are also commonly used.

### 4.4.2 PCA-Adjusted Regression (EIGENSTRAT)

Population stratification is a major confounding factor in GWAS, arising when ancestry induces systematic variation in allele frequencies across individuals. In the presence of population structure, ancestry-associated SNPs may appear correlated with the phenotype solely due to shared ancestry rather than genuine genotype–phenotype association, leading to spurious association findings. EIGENSTRAT (Patterson et al., 2006; Price et al., 2006) was proposed to address this limitation of standard regression-based tests by removing ancestry-driven variation prior to association testing. Using principal component analysis (PCA), it provides an effective and computationally tractable way to correct for population structure while retaining marginal SNP-wise inference.

Assume a genotype matrix of $n$ individuals measured at $m$ SNPs, $\mathbf{G} \in \{0, 1, 2\}^{n \times m}$, and a phenotype vector $\mathbf{y} \in \mathbb{R}^n$, such that

$$\mathbf{G} = \begin{pmatrix} g_{11} & g_{12} & \cdots & g_{1m} \\ g_{21} & g_{22} & \cdots & g_{2m} \\ \vdots & \vdots & \ddots & \vdots \\ g_{n1} & g_{n2} & \cdots & g_{nm} \end{pmatrix}, \qquad \mathbf{y} = \begin{pmatrix} y_1 \\ y_2 \\ \vdots \\ y_n \end{pmatrix}.$$

The goal of EIGENSTRAT is to separate ancestry-driven variation from genuine genotype–phenotype association signals. To this end, the genotype matrix is first column-wise standardized to obtain $\mathbf{G}^s$, typically

by centering and scaling each SNP:

$$g_{ij}^s = \frac{g_{ij} - 2f_j}{\sqrt{2f_j(1-f_j)}},$$

where $f_j = \frac{1}{2n}\sum_{i=1}^n g_{ij}$ denotes the MAF at SNP $j$. EIGENSTRAT then constructs the individual-level *genetic relationship matrix* (GRM)

$$\mathbf{\Psi} = \frac{1}{m}\mathbf{G}^s\mathbf{G}^{s\top},$$

which captures genome-wide genetic similarity between individuals (Yang et al., 2011). This matrix reflects population relatedness and clustering induced by shared ancestry.

PCA is performed by computing the eigenvectors of $\mathbf{\Psi}$. The leading eigenvectors correspond to dominant axes of population structure, capturing systematic allele-frequency differences driven by ancestry. Let

$$\mathbf{V} = [\mathbf{v}_1, \ldots, \mathbf{v}_k] \in \mathbb{R}^{n \times k}$$

denote the top $k$ eigenvectors of $\mathbf{\Psi}$, where $k$ specifies the number of retained ancestry components. In practice, a small number of leading principal components is typically used, guided by inspection of the leading principal components and by evaluating standard GWAS diagnostics after correction. Practical considerations regarding PCA-based ancestry inference and component selection are discussed further in Section 5.4.

Since these eigenvectors live in the space of individuals, they can be used to adjust both genotypes and phenotypes. Rather than projecting onto these components, EIGENSTRAT removes their contribution by projecting onto the orthogonal complement of their span. Specifically, the *ancestry-adjusted* genotype matrix and phenotype vector are defined as

$$\mathbf{G}^* = (\mathbf{I} - \mathbf{V}\mathbf{V}^\top)\mathbf{G}^s, \qquad \mathbf{y}^* = (\mathbf{I} - \mathbf{V}\mathbf{V}^\top)\mathbf{y}^s,$$

where $\mathbf{I}$ denotes the $n \times n$ identity matrix, $\mathbf{y}^s$ denotes the standardized phenotype vector and the columns of $\mathbf{V}$ are assumed to be orthonormal. This operation removes from each SNP and from the phenotype any component explained by population structure as captured by the top $k$ principal components. This procedure is equivalent to including these components as covariates in a linear regression model. Notably, in case–control studies the adjusted phenotype vector $\mathbf{y}^*$ is no longer binary; instead, it represents the residual phenotype after removing ancestry-associated variation, which is appropriate for subsequent linear association testing.

Association testing is then performed by assessing the correlation between each adjusted SNP vector and the adjusted phenotype. For SNP $j$, let $\mathbf{G}_j^*$ denote the $j$-th column of $\mathbf{G}^*$. EIGENSTRAT employs the test statistic

$$\chi_j^2 = \frac{(n-k-1)(\mathbf{G}_j^* \cdot \mathbf{y}^*)^2}{\|\mathbf{G}_j^*\|^2\, \|\mathbf{y}^*\|^2},$$

which follows a $\chi^2$ distribution with one degree of freedom under the null hypothesis. Equivalently, letting

$$r_j = \frac{\mathbf{G}_j^* \cdot \mathbf{y}^*}{\|\mathbf{G}_j^*\|\, \|\mathbf{y}^*\|}$$

denote the sample correlation between the ancestry-adjusted genotype and phenotype, the statistic can be written as $\chi_j^2 = (n-k-1)r_j^2$. The factor $n-k-1$ reflects the number of remaining degrees of freedom after removing $k$ principal components and the intercept. Under the null hypothesis

$$H_0 : \mathbf{G}_j^* \text{ and } \mathbf{y}^* \text{ are uncorrelated,}$$

this statistic provides a valid test of association that is robust to confounding due to population stratification.

### 4.4.3 Linear Mixed Models

Linear mixed models (LMMs) (Yu et al., 2006; Kang et al., 2010) provide a principled framework for association testing in GWAS by explicitly modeling genome-wide genetic similarity, thereby addressing both population stratification and sample relatedness. Unlike standard regression and PCA-adjusted methods such as EIGENSTRAT, which retain marginal SNP-wise inference after fixed-effect correction and assume independence across individuals, LMMs incorporate the aggregate contribution of unobserved variants through a polygenic random effect. This modeling choice allows phenotypes of genetically similar individuals to be correlated and yields calibrated association tests even in the presence of complex population structure and fine-scale relatedness.

Let $\mathbf{G} \in \{0, 1, 2\}^{n \times m}$ denote the genotype matrix of $n$ individuals at $m$ SNPs, and let $\mathbf{y} \in \mathbb{R}^n$ be a quantitative phenotype. For a given SNP $j$, association testing under an LMM is based on

$$\mathbf{y} = \mathbf{1}\beta_0 + \mathbf{g}_j\beta_j + \mathbf{C}\boldsymbol{\gamma} + \mathbf{u} + \boldsymbol{\varepsilon},$$

where $\mathbf{1} \in \mathbb{R}^n$ denotes the vector of ones, $\mathbf{g}_j \in \mathbb{R}^n$ is the genotype vector for SNP $j$, $\mathbf{C} \in \mathbb{R}^{n \times K}$ is a matrix of fixed covariates (e.g., age, sex, or principal components), and $\beta_0$, $\beta_j$, and $\boldsymbol{\gamma}$ are the corresponding fixed-effect coefficients. The vectors $\mathbf{u}$ and $\boldsymbol{\varepsilon}$ represent random genetic effects and residual noise, respectively.

The defining feature of LMMs is the distributional assumption placed on the random effect:

$$\mathbf{u} \sim \mathcal{N}(\mathbf{0}, \sigma_g^2\boldsymbol{\Psi}), \qquad \boldsymbol{\varepsilon} \sim \mathcal{N}(\mathbf{0}, \sigma_e^2\mathbf{I}),$$

where $\boldsymbol{\Psi} \in \mathbb{R}^{n \times n}$ denotes the GRM, typically computed from standardized genome-wide SNP data analogously to Section 4.4.2. Unlike EIGENSTRAT, however, LMMs do not use $\boldsymbol{\Psi}$ merely to extract a small number of principal components. Instead, the full matrix directly parameterizes the covariance structure of the phenotype through the random effect. The term $\sigma_g^2\boldsymbol{\Psi}$ captures genome-wide polygenic similarity between individuals, including population structure and fine-scale relatedness, while $\sigma_e^2\mathbf{I}$ models residual variation not explained by genetic relatedness. In practice, $\boldsymbol{\Psi}$ is estimated from a large set of SNPs, often *excluding* the tested locus or chromosome to avoid proximal contamination, which can occur when the association signal under test is partially absorbed into the polygenic background term (Listgarten et al., 2012; Tucker et al., 2014).

Under this model, phenotypes of genetically similar individuals are no longer assumed independent. Their covariance is given by

$$\mathrm{Var}(\mathbf{y}) = \sigma_g^2\boldsymbol{\Psi} + \sigma_e^2\mathbf{I},$$

where $\sigma_g^2$ and $\sigma_e^2$ quantify the contributions of polygenic background contributions and residual noise, respectively.

The random effect $\mathbf{u}$ captures the aggregate contribution of many background genetic variants, while the GRM $\boldsymbol{\Psi}$ models how these genetic contributions are shared across genetically similar individuals. This formulation enables calibrated association testing in structured and related samples.

Association testing for SNP $j$ proceeds by testing the null hypothesis

$$H_0 : \beta_j = 0 \qquad \text{vs.} \qquad H_1 : \beta_j \neq 0,$$

The variance components $(\sigma_g^2, \sigma_e^2)$, which quantify genetic and residual contributions to phenotype covariance, are typically estimated once under the null model using likelihood-based approaches applied to genome-wide genotype data and covariates. Association testing is then performed separately for each SNP using Wald, likelihood-ratio, or score tests. Modern implementations exploit this structure to achieve computational scalability in large-scale GWAS (e.g., Kang et al., 2008; 2010; Zhou & Stephens, 2012).

From a conceptual perspective, LMMs generalize both standard regression and PCA-based correction methods by explicitly modeling genetic correlations between individuals. Standard linear or logistic regression assumes $\mathrm{Var}(\mathbf{y}) = \sigma^2\mathbf{I}$ and therefore ignores population structure and relatedness, which can inflate false-positive rates in structured cohorts. EIGENSTRAT mitigates this issue through fixed-effect adjustment using

a small number of ancestry-informative principal components. In contrast, LMMs model covariance using the full GRM through a polygenic random effect, allowing them to account simultaneously for global population structure and fine-scale relatedness. This distinction becomes particularly important in large cohorts, where confounding structure is often distributed across many weak components that are not adequately captured by a limited set of principal components (Yang et al., 2014).

## 4.5 Genome-Wide Significance and Multiple Testing

Although the association methods discussed above differ substantially in their statistical formulation, they share a common inference paradigm: association is evaluated separately for each SNP across the genome. Consequently, a GWAS involving $m$ variants produces a large collection of simultaneous hypothesis tests, creating a substantial multiple-testing problem.

To control false positives arising from repeated testing, genome-wide association analyses require multiple-testing correction procedures. A widely used approach is the Bonferroni correction (Bonferroni, 1936), which adjusts the significance threshold according to the number of SNPs tested by evaluating each SNP at significance level $\alpha/m$, where $m$ denotes the number of tests. Although conservative, particularly because LD induces dependence among nearby SNPs, it provides a simple and interpretable genome-wide significance threshold. In human GWAS, a commonly used significance threshold is $5 \times 10^{-8}$ (Chen et al., 2021; Pe'er et al., 2008).

Alternative approaches, including false discovery rate control (Benjamini & Yekutieli, 2005) and permutation-based procedures (Che et al., 2014), are also used in practice. Regardless of the specific correction strategy, multiple-testing adjustment constitutes a general component of GWAS inference and applies broadly across table-based, family-based, regression-based, and mixed-model association frameworks.

While the methods discussed in Sections 4.3 and 4.4 primarily follow a SNP-wise testing paradigm, more complex joint and multivariate formulations have also been explored, particularly for model-based GWAS analyses involving multiple loci or multiple traits (Hayes, 2013; Yang et al., 2013; Zhou & Stephens, 2014).

## 4.6 Evaluation Metrics

Evaluating the utility of DP mechanisms for GWAS requires assessing how well downstream association analyses are preserved after the introduction of DP noise. Accordingly, evaluation can be framed in terms of the fidelity of association testing outcomes. In this section, we summarize commonly used evaluation metrics for GWAS downstream tasks.

**Ranking and Selection Fidelity.** Many works evaluate DP mechanisms based on how well the relative ordering of SNPs by significance or effect size is preserved (e.g., Simmons & Berger, 2016; Wang et al., 2017a; Yu et al., 2014a; Yamamoto & Shibuya, 2023b). This includes the stability of SNP rankings under noise and the recovery accuracy of highly ranked variants. Such metrics reflect whether privatization alters the prioritization of candidate variants, even if exact test statistics differ.

**Statistic-Level Fidelity.** A complementary class of metrics examines the fidelity of association statistics or regression outputs themselves. This includes deviations in test statistics (e.g., Wald or likelihood ratio statistics) as well as perturbations to estimated regression coefficients and their standard errors. By comparing the distributions or numerical differences of these quantities before and after privatization, one can assess how DP noise propagates through the association pipeline and whether calibration and effect size estimation remain reliable (e.g., Uhlerop et al., 2013; Yamamoto & Shibuya, 2021c; Johnson & Shmatikov, 2013).

**Hypothesis-Level Fidelity.** Finally, evaluation may focus directly on hypothesis testing behavior. This involves assessing whether the statistical power to detect genuinely associated SNPs is preserved while controlling the Type I error (false positive) rate at nominal significance thresholds. From this perspective, utility is measured by the agreement in rejection decisions between private and non-private analyses (e.g., Sei & Ohsuga, 2021; Roozgard et al., 2016; Halimi et al., 2022).

Overall, these evaluation metrics provide complementary views of utility, ranging from local accuracy at individual SNPs to global preservation of statistical behavior. A comprehensive assessment of DP methods for GWAS typically reports multiple metrics to capture trade-offs between privacy, statistical power, and inferential reliability.

## 5   Beyond GWAS Outputs: Structural Notions of Utility

In the previous section, we reviewed the main downstream tasks associated with GWAS, including classical hypothesis testing approaches as well as model-based methods. While performance on these tasks constitutes an important notion of utility, it does not fully characterize the usefulness of a genomic dataset, and in particular of SNP datasets. Genomic data exhibit rich population-level structure that extends beyond association signals and plays a central role in many analyses outside the GWAS setting.

When a DP mechanism releases data in the same representation space as the original dataset, either through synthetic data generation or through LDP applied at the individual level, it is important that these underlying population-specific structures are preserved. In this section, we focus on utility notions related to population structure beyond GWAS. We provide an overview of key structural properties that a privatized SNP dataset should retain, along with standard methods of quantifying their agreement with a non-private reference dataset. We organize these properties in order of increasing structural complexity.

### 5.1   Allele Frequency Structure

A natural first property to examine when comparing a synthetic or DP-sanitized dataset to the original data is whether the marginal distributions are preserved. In SNP datasets, these marginals correspond to per-locus allele frequencies, typically summarized by the MAF as defined in Equation 2. A variety of complementary measures can be used to assess how well allele frequency structure is preserved.

**Per-locus error.** Let $f_j$ denote the MAF of SNP $j$ in the real dataset and $f'_j$ the corresponding frequency in the private dataset, for a total of $L$ loci. Common summary statistics include:

$$\text{Mean Absolute Error:} \quad \frac{1}{L} \sum_{j=1}^{L} |f_j - f'_j|$$

$$\text{Root Mean Squared Error:} \quad \sqrt{\frac{1}{L} \sum_{j=1}^{L} (f_j - f'_j)^2}$$

$$\text{Maximum Error:} \quad \max_j |f_j - f'_j|$$

Since rare variants play an important role in many genomic analyses (Saint Pierre & Génin, 2014) and are disproportionately affected by privacy-preserving perturbations, as a fixed amount of DP noise can induce substantially larger relative errors at low frequencies, it is advisable to stratify these errors by MAF bins, for example $[0, 0.001)$, $[0.001, 0.01)$, $[0.01, 0.1)$, and higher-frequency ranges.

**Allele frequency spectrum.** Beyond per-locus errors, population-level marginal structure can be assessed through the *allele frequency spectrum* (AFS) (Evans et al., 2007; Wright, 1938). Let $\mathbf{x} = (x_1, x_2, x_3, \ldots)$ denote the histogram of allele frequencies or allele counts, where $x_i$ represents the number of loci with frequency $i$ (or falling into a predefined frequency bin). The similarity between the AFS of the real and DP-sanitized datasets can then be quantified using standard distributional distances such as total variation distance, Wasserstein distance, or Jensen-Shannon divergence.

**Genotype frequency marginals.** As a slightly richer alternative to allele frequency comparisons, one may directly compare genotype frequency marginals. For each locus, the proportions of genotypes $g \in \{0, 1, 2\}$ can be computed in the real and private datasets and summarized using average $\ell_1$ or $\ell_2$ distances across loci.

While still marginal in nature, this view captures deviations that may not be visible at the allele-frequency level alone.

## 5.2 LD Patterns

Moving beyond marginal distributions, we now consider pairwise and local dependency structures in SNP datasets. As described in Section 3.2.1, SNPs exhibit non-random correlation patterns known as LD. It is important that these patterns are not severely distorted in a synthetic or DP-sanitized dataset, as they underpin a range of downstream analyses.

A simple qualitative check of LD preservation is provided by visual inspection of LD matrices, which summarize pairwise correlations along genomic regions using measures such as $r^2$ (Equation 3). While such visualizations assess second-order (pairwise) dependency patterns, they do not capture the full multivariate structure of genetic variation within a region.

In practice, many downstream tasks rely on higher-order and structured dependencies beyond isolated pairwise correlations. These include extended LD patterns across genomic regions and haplotype-level dependencies spanning multiple loci. Therefore, evaluating the utility of a DP-sanitized dataset requires task-based assessments that probe this richer dependency structure. We describe the most relevant ones below.

### 5.2.1 Genotype Imputation

Genotyping remains a costly and error-prone process, and large-scale SNP datasets typically contain missing or unobserved genotypes. The statistical inference of these missing genotypes is known as *imputation* and is commonly performed using high-quality reference panels of densely genotyped individuals.

Imputation methods operate on *haplotypes*, defined as the ordered sequence of alleles along a single chromosome. Each diploid individual therefore carries two haplotypes for any genomic region. State-of-the-art imputation methods are based on the Li-Stephens model (Li & Stephens, 2003; Li et al., 2009), which assumes that a target haplotype can be represented as an imperfect mosaic of reference haplotypes drawn from a population. In practice, this model is implemented using hidden Markov models (HMMs) trained on reference haplotype panels, as in methods such as MaCH (Li et al., 2010), Minimac (Das et al., 2016), Beagle (Browning et al., 2021), and SHAPEIT (Delaneau & Marchini, 2014).

The effectiveness of HMM-based imputation relies directly on LD, since correlations among nearby SNPs constrain the space of plausible haplotypes and enable accurate inference of missing genotypes from observed neighboring loci. Consequently, disruption of LD patterns severely degrades imputation performance.

**Evaluation.** Imputation quality is typically assessed on held-out genotypes after training of the imputation model. Common metrics include the *concordance rate*, defined as the proportion of correctly imputed genotypes, and the *dosage $R^2$*, which measures the squared Pearson correlation between the imputed allele dosage and the true genotype. The allele dosage is defined as

$$\text{Dosage} = \Pr(\text{het} \mid \text{data}) + 2 \Pr(\text{alt} \mid \text{data}),$$

where $\Pr(\text{het} \mid \text{data})$ denotes the probability of a heterozygous genotype and $\Pr(\text{alt} \mid \text{data})$ the probability of being homozygous for the alternate allele. As with allele frequency evaluation, it is advisable to stratify these metrics by MAF bins. A detailed discussion of imputation metrics can be found in Ramnarine et al. (2015).

### 5.2.2 Haplotype Phasing

Standard genotyping technologies do not resolve which alleles originate from which homologous chromosome, resulting in unphased genotypes. The task of reconstructing haplotypes from unphased genotype data is known as *phasing*. Like imputation, phasing relies heavily on LD patterns and is commonly performed using HMM-based models trained on reference data, as in PHASE (Stephens et al., 2001), fastPHASE (Scheet & Stephens, 2006), and IMPUTE2 (Howie et al., 2009). These methods exploit correlations across neighboring loci to infer the most likely haplotype configuration consistent with the observed genotypes.

**Evaluation.** Phasing accuracy is evaluated by comparing inferred haplotypes to ground-truth phase information, typically obtained from held-out data with known haplotypes. Common metrics include the *individual error rate*, defined as the proportion of individuals with at least one phasing error, and the *switch error rate*, which measures the proportion of heterozygous loci at which the inferred phase is inconsistent with the true phase relative to the preceding heterozygous locus.

### 5.2.3 Polygenic Risk Score Construction

Many complex traits and diseases are influenced by a large number of genetic variants, each contributing a small effect. A *polygenic risk score* (PRS) (International Schizophrenia Consortium, 2009) provides an individual-level summary of this aggregate genetic contribution. The PRS represents a weighted sum of risk alleles carried by an individual, where the weights reflect the estimated effect sizes from association tests.

Given GWAS-derived effect size estimates (weights) $\hat{\beta}_j$ for a selected set of $m$ SNPs, the PRS for an individual is computed as

$$\widehat{\text{PRS}} = \sum_{j=1}^{m} \hat{\beta}_j g_j,$$

where $g_j \in \{0, 1, 2\}$ denotes the genotype at locus $j$. The resulting score can be interpreted as a linear predictor of genetic liability, such that individuals with higher PRS values are predicted to have a higher genetic predisposition to the trait under consideration.

Although PRS aggregates marginal SNP effects, LD plays an important role in its construction. SNPs in strong LD convey redundant information, and failure to account for local correlations can lead to inflated variance and reduced predictive performance. As a result, most PRS pipelines incorporate LD-aware steps such as pruning or shrinkage, making PRS construction sensitive to both under- and over-estimation of LD structure (Choi et al., 2020; Vilhjálmsson et al., 2015).

**Evaluation.** Utility can be assessed at different stages of the PRS pipeline. If SNP weights $\hat{\beta}_j$ are treated as fixed (e.g., derived from an external GWAS), privacy may be applied only to individual-level genotypes used for scoring. In this case, evaluation compares PRS values computed from privatized genotypes to those obtained from the original data, measuring distortion at the risk score level.

Alternatively, privacy may be applied during model estimation. Effect sizes and, where applicable, LD-adjusted models are learned from privatized genotypes or summary statistics. Evaluation then focuses on the predictive performance of the resulting PRS model.

## 5.3 Genetic Distances

While LD captures local correlations among nearby loci, global properties of a genomic dataset are more naturally characterized through genetic distances between individuals and populations.

Genetic distances summarize how individuals or populations relate to one another in genotype space and provide a distribution-level notion of similarity between datasets. Such distances have strong biological foundations and have long been used to study population differentiation, evolutionary divergence, and shared ancestry. In the context of DP, genetic distances offer a natural way to assess how closely a DP-sanitized dataset resembles its non-private counterpart beyond marginal distributions and local correlation structures.

Genetic distances can be considered at two distinct but related levels: distances between populations and distances between individuals. These serve complementary roles when evaluating both the utility and the privacy implications of synthetic or DP-protected genomic data.

### 5.3.1 Population-Level Distances

Population-level genetic distances characterize aggregate divergence between groups and are commonly used to study population structure, differentiation, and evolutionary relationships. In the DP setting, these measures are particularly useful for assessing whether a sanitized dataset preserves the overall structure of

populations present in the original data. Preservation of population-to-population distances indicates that a DP mechanism maintains global population relationships and relative genetic divergence, even if fine-grained individual-level structure is perturbed. A wide range of such measures has been proposed; here, we focus on the most commonly used and informative ones.

**Fixation index ($\mathbf{F_{st}}$).** One of the most fundamental measures of population differentiation is the fixation index (Wright, 1933; 1949). Conceptually, $F_{st}$ quantifies the extent to which allele frequency variation is attributable to differences between populations rather than variation within populations. Its values range from 0 to 1, with 0 indicating no genetic differentiation.

While Wright's original formulations are biologically detailed and context-dependent, several practical estimators have been proposed. Here, we introduce Nei's heterozygosity-based estimator (Nei, 1973; Nei & Chakravarti, 1977), noting that the estimator of Weir & Cockerham (1984) is also widely used, particularly for small sample sizes. For a detailed comparison, see Bhatia et al. (2013).

For $K$ subpopulations and $L$ SNP loci, $F_{st}$ can be estimated as

$$\widehat{F_{st}} = \frac{\sum_{l=1}^{L} (H_{T,l} - H_{S,l})}{\sum_{l=1}^{L} H_{T,l}},$$

where $H_{T,l}$ denotes the expected heterozygosity in the total population at locus $l$, and $H_{S,l}$ is the average expected heterozygosity within subpopulations at that locus.

For a bi-allelic SNP with minor and major allele frequencies $f(a)$ and $f(A)$, respectively, the expected heterozygosity under HWE (see Section 3.2.1) is $H = 2f(a)f(A) = 2f(a)\big(1 - f(a)\big)$. Accordingly,

$$H_{S,l} = \frac{1}{K} \sum_{k=1}^{K} 2f_l^k(a)\big(1 - f_l^k(a)\big), \qquad H_{T,l} = 2\bar{f}_l(a)\big(1 - \bar{f}_l(a)\big),$$

where $f_l^k(a)$ denotes the allele frequency in subpopulation $k$ at locus $l$, and $\bar{f}_l(a)$ is the pooled allele frequency across all subpopulations.

**Nei's standard genetic distance.** Nei's standard genetic distance (Nei, 1972; Katada et al., 2004) is a probability-based measure of divergence between populations that reflects the likelihood that two alleles drawn from different populations are identical in state. Unlike $F_{st}$, which measures relative differentiation, Nei's distance provides an absolute measure of genetic divergence and increases approximately linearly with time under the assumptions of genetic drift and mutation.

Let $f_j^P(a)$ and $f_j^P(A)$ denote the minor and major allele frequencies of population $P$ at locus $j$. The probability that two alleles drawn from population $P$ are identical at locus $j$ is

$$p_j^P = (f_j^P(a))^2 + (f_j^P(A))^2,$$

and similarly for population $Q$. The probability that one allele drawn from $P$ and one from $Q$ are identical is

$$p_j^{PQ} = f_j^P(a)f_j^Q(a) + f_j^P(A)f_j^Q(A).$$

The normalized identity of genes at locus $j$ is

$$I_j = \frac{p_j^{PQ}}{\sqrt{p_j^P p_j^Q}}.$$

Aggregating across loci, Nei's genetic distance is defined as

$$D_{\text{Nei}}(P, Q) = -\ln\left(\frac{\sum_{j=1}^{L} p_j^{PQ}}{\sqrt{\left(\sum_{j=1}^{L} p_j^P\right)\left(\sum_{j=1}^{L} p_j^Q\right)}}\right).$$

When allele frequencies are identical across populations, $D_{\text{Nei}} = -\ln(1) = 0$, and the value approaches infinity as the populations diverge. For further discussion of the relationship between $F_{\text{st}}$ and Nei's distance, see Kalinowski (2002).

**Czekanowski (Manhattan) and Euclidean distances.** Geometric distances such as Manhattan and Euclidean distances provide simple, model-free measures of population dissimilarity. Although they lack explicit biological interpretation, they are computationally efficient and commonly used in practice.

The normalized Euclidean distance between populations $P$ and $Q$ is

$$D_{\text{Eu}}(P, Q) = \sqrt{\frac{1}{L} \sum_{j=1}^{L} \left( f_j^P(a) - f_j^Q(a) \right)^2},$$

and the normalized Manhattan distance is

$$D_{\text{Cz}}(P, Q) = \frac{1}{L} \sum_{j=1}^{L} \left| f_j^P(a) - f_j^Q(a) \right|.$$

Here, $f_j(a)$ denotes the MAF at locus $j$, and normalization ensures that distances lie in $[0, 1]$.

### 5.3.2 Individual-Level Distances

Individual-level genetic distances are primarily used in GWAS as quality control tools (Anderson et al., 2010) rather than as analysis objectives. They enable detection of duplicates and close relatives, whose presence violates the independence assumptions underlying standard association tests and can inflate false positive rates. Individual-level genetic distances represent a particularly sensitive aspect of DP-sanitized genomic data. While preserving them supports essential quality control tasks, such as identifying close relatives, overly accurate retention of fine-scale relatedness risks enabling familial inference. Here, we present some of the most commonly-used individual-level measures.

**Identity-by-descent (IBD).** IBD quantifies genetic relatedness between two individuals through the amount of genome they inherit from a common ancestor. While all individuals share common ancestry at sufficiently long time scales, close relatives exhibit substantially higher IBD sharing.

At a fixed locus, the alleles carried by two diploid individuals can be compared in terms of how many alleles are identical by descent. A standard formalization uses the probabilities

$$k_0 = \Pr(0 \text{ alleles IBD}), k_1 = \Pr(1 \text{ allele IBD}), k_2 = \Pr(2 \text{ alleles IBD}),$$

with $k_0 + k_1 + k_2 = 1$. These quantities summarize genome-wide IBD sharing and are commonly estimated from dense SNP data using HMM-based approaches (e.g., Browning & Browning, 2011). Model-based inference is necessary to distinguish true shared ancestry from chance allele sharing, which is inflated for common variants. Closely related pairs have characteristic $(k_0, k_1, k_2)$ patterns; for example, parent-child pairs satisfy $(k_0, k_1, k_2) = (0, 1, 0)$ and full siblings satisfy $(k_0, k_1, k_2) = (\frac{1}{4}, \frac{1}{2}, \frac{1}{4})$ under idealized assumptions (see Section 3.2.2 for intuition).

**KING coefficient.** The KING coefficient (Manichaikul et al., 2010) provides a fast, robust estimator of pairwise kinship derived from IBD principles. It estimates the probability $\phi_{ij}$ that two alleles sampled at random from individuals $i$ and $j$ are identical by descent:

$$\phi_{ij} = \frac{k_{1,ij}}{4} + \frac{k_{2,ij}}{2}.$$

An empirical estimator based on genotype counts is given by

$$\hat{\phi}_{ij} = \frac{M_{Aa,Aa} - 2M_{AA,aa}}{2M_{Aa}^i} + \frac{1}{2} - \frac{M_{Aa}^i + M_{Aa}^j}{4M_{Aa}^i}, \tag{5}$$

where $M_{Aa,Aa}$ denotes number of loci where both individuals are heterozygous, $M_{AA,aa}$ number of loci with opposite homozygotes, and $M_{Aa}^i$ and $M_{Aa}^j$ the number of heterozygous loci for individuals $i$ and $j$, respectively. A value of $\phi_{ij} = 0.5$ corresponds to duplicates or identical twins, with smaller values indicating more distant relationships.

**Genetic relationship matrix.** The GRM (Yang et al., 2011) provides a continuous, genome-wide measure of pairwise genetic similarity and is central to methods such as EIGENSTRAT and linear mixed models (see Section 4.4). Let $\mathbf{G} \in \mathbb{R}^{n \times m}$ denote the genotype matrix for $n$ individuals and $m$ SNPs, where $g_{ij} \in \{0, 1, 2\}$ represents the genotype of individual $i$ at locus $j$. Standardizing each SNP column-wise yields a normalized genotype matrix $\mathbf{G}^s \in \mathbb{R}^{n \times m}$ with entries

$$g_{ij}^s = \frac{g_{ij} - 2f_j(a)}{\sqrt{2f_j(a)\big(1 - f_j(a)\big)}}, \tag{6}$$

where $f_j(a)$ denotes the MAF at locus $j$. The GRM is then defined as

$$\mathbf{\Psi} = \frac{1}{m}\mathbf{G}^s\mathbf{G}^{s\top}.$$

Unlike IBD or KING coefficients, which target discrete relationship classes, the GRM captures overall genetic similarity on a continuous scale. By aggregating information across all loci, it smooths local variation and provides a robust measure of non-independence among individuals.

**Evaluation.** Evaluation of individual-level genetic distances in DP-sanitized datasets should focus on whether fine-scale relatedness structure is preserved or intentionally attenuated. For IBD-based measures, this can be assessed by comparing the distributions of the estimated sharing probabilities $\hat{k}_0, \hat{k}_1, \hat{k}_2$ (or derived quantities) across all individual pairs, as well as by evaluating the accuracy of relationship classification (e.g., unrelated, second-degree, first-degree) obtained from thresholds on estimated kinship coefficients (Manichaikul et al., 2010). Similar analyses can be performed using the KING coefficient, which provides a computationally efficient proxy for pairwise kinship (e.g., Dervishi et al., 2023b). In contrast, the GRM enables the evaluation of continuous genome-wide similarity, for example, by comparing the empirical distributions of GRM entries or their spectral properties between private and non-private datasets.

## 5.4 Population Stratification

Population stratification refers to the presence of subgroups with distinct ancestries within a genomic dataset, which can induce spurious allele–phenotype associations if not properly accounted for. It reflects latent, genome-wide structure arising from ancestry differences and constitutes a major source of confounding in downstream analyses. Reliable GWAS analyses therefore require population structure being both detectable and correctable, for example through PCA, EIGENSTRAT, or linear mixed models (see Section 4.4). In the context of DP, it is crucial that a DP-sanitized dataset preserves this large-scale ancestry structure sufficiently for standard stratification analyses to remain meaningful.

**Detection.** A classical approach to detecting confounding due to population stratification is the genomic control factor $\lambda_{\mathrm{GC}}$ (Devlin & Roeder, 1999; Pritchard & Rosenberg, 1999; Reich & Goldstein, 2001). It is defined as the ratio between the median of the observed $\chi^2$ association statistics across SNPs and the theoretical median under the null hypothesis of no association. Values of $\lambda_{\mathrm{GC}} \approx 1$ indicate little or no inflation, while $\lambda_{\mathrm{GC}} > 1$ suggests the presence of uncorrected population structure or other confounders.

**Inferring genetic ancestry.** PCA has been used for decades to infer genetic ancestry and population structure from genomic data (Cavalli-Sforza et al., 1994; Patterson et al., 2006; Novembre & Stephens, 2008). In GWAS, the leading principal components are commonly included as covariates to correct for stratification.

Let $\mathbf{G}^s \in \mathbb{R}^{n \times m}$ denote the standardized genotype matrix for $n$ individuals and $m$ SNPs, with entries defined as in Equation 6. PCA is applied by performing eigendecomposition of the GRM $\mathbf{\Psi}$, yielding a set of

orthogonal ancestry axes. In practice, a small number of leading principal components (often on the order of $5 - 10$) is sufficient to reveal population clusters corresponding to shared ancestry (Zhao et al., 2018; Novembre et al., 2008).

It is important to note that leading principal components do not exclusively capture population structure. They may also reflect family relatedness, long-range LD, or technical artifacts (Tian et al., 2008; Patterson et al., 2006; Price et al., 2010). These effects can be mitigated through standard quality control steps, including removal of closely related individuals, filtering of problematic variants, and LD pruning prior to PCA. LD pruning typically proceeds by sliding a window along the genome, computing pairwise LD within each window, and removing one SNP from each highly correlated pair; this process may be iterated multiple times (Miles, 2015; Novembre et al., 2008). Since modern SNP datasets often already include extensive quality control, LD pruning is usually performed explicitly by the researcher before PCA.

### 5.4.1 Evaluation

In the context of DP-sanitized datasets, the central question is whether applying PCA to the sanitized data yields ancestry structure that is comparable to that obtained from the original, non-private dataset (e.g., Ghasemian et al., 2025).

**Geometric consistency.** A natural first evaluation criterion is the geometric consistency of inferred sub-populations in the PCA space. This can be assessed by comparing the relative arrangement of individuals or population clusters between the DP-sanitized and original datasets. Standard metrics include within- versus between-population distance ratios in the latent space, as well as clustering quality measures such as the silhouette score (Rousseeuw, 1987), which quantifies the separability of groups in the embedding.

**Recovery of ancestry information.** Ancestry-related signals should remain recoverable at the individual level. In particular, individuals belonging to the same subpopulation in the original data should remain close in the PCA embedding after DP sanitization, including under local DP mechanisms. Since PCA is performed in the space of individuals, each eigenvector is $n$-dimensional. Denoting by $\mathbf{U}_k = [\mathbf{v}_1, \ldots, \mathbf{v}_k]$ the matrix of the top-$k$ eigenvectors for the original dataset and by $\mathbf{U}'_k$ the corresponding matrix for the DP-sanitized data, alignment between the two embeddings can be quantified using matrix norms such as the Frobenius norm, or by correlating corresponding principal components.

**Effectiveness of stratification correction.** Ultimately, the practical utility of preserved population structure lies in its effectiveness for confounding correction in association testing. After principal component-based correction, the genomic control factor $\lambda_{\mathrm{GC}}$ computed on the DP-sanitized dataset should should remain comparable to that obtained from the original dataset and, ideally, close to 1, indicating adequate control of stratification. Quantile-quantile (Q-Q) plots (or probability-probability (P-P) plots) provide a complementary diagnostic by comparing observed test statistics to their expected null distribution: strong deviations from the diagonal before correction should be substantially reduced after principal component adjustment. Performance can be evaluated by comparing these diagnostics between the original and DP-sanitized datasets. Finally, downstream PCA-based association methods such as EIGENSTRAT or regression with principal component covariates should yield results comparable to those obtained on the non-private data, with agreement assessed using the evaluation criteria described in Section 4.6.

## 6 Privacy Attacks on SNP Datasets

We dedicate this section to illustrating the various types of privacy attacks that can be conducted on SNP datasets, highlighting the pressing need for privacy-preserving methodologies. We provide a high-level categorization of the most prominent classes of attacks. It is important to note that this section does not aim to provide an exhaustive review of all existing attacks and literature in this domain; rather, its purpose is to acquaint the reader with the principal directions of research and types of threats. For a more detailed analysis of these attacks, real-world examples of individual privacy breaches, and an extensive collection of related studies, see, for example, Erlich & Narayanan (2014); Mohammed Yakubu & Chen (2020); Ayday & Humbert (2017); Naveed et al. (2015).

## 6.1 Membership Inference Attacks

One of the earliest and most extensively studied forms of privacy attacks on genomic data is the *membership inference (MI) attack*, also known as *identification attack* or *attribution disclosure attack*. In an MI attack, the adversary is assumed to know the genome, or parts of the genome, of a target individual and seeks to determine whether that individual participated in a particular dataset or study.

In the simplest scenario, the attacker is assumed to have access to a dataset containing the genotypes of individuals whose identities have been anonymized. Under this assumption, Lin et al. (2004) demonstrated that as few as 30 to 80 statistically independent SNPs are sufficient to re-identify individuals. While such an attack is theoretically powerful, it relies on an unrealistic premise, namely that the adversary has direct access to all data points within the dataset.

A more realistic scenario arises when only aggregate or summary statistics are publicly released following a study. In this context, Homer et al. (2008) introduced an attack motivated by a forensic setting, where the adversary aims to determine whether a target individual's DNA contributes to a complex genomic mixture, such as DNA recovered from a forensic scene, contaminated biological sample, or highly mixed trace sample. By comparing an individual's genotype against the allele frequencies of both the target mixture and a reference population, they showed that it is possible to infer the individual's presence using only 10,000 to 25,000 SNPs, even when that individual contributes less than 1% of the total mixture population. This result prompted policy changes in genomic data sharing. Notably, NIH moved GWAS aggregate data to controlled-access mechanisms, and organizations such as the Wellcome Trust also restricted public access to aggregate genomic datasets (Zerhouni & Nabel, 2008; Naveed et al., 2015).

Building on this line of work, Wang et al. (2009) considered a more direct biomedical privacy threat model. Many genomic datasets, particularly GWAS datasets, are collected around a specific disease phenotype and are organized into case and control cohorts. Consequently, successfully inferring that an individual belongs to the case cohort may immediately reveal highly sensitive medical information, namely that the target is associated with the disease under study. Wang et al. (2009) demonstrated that even highly compact released statistics, such as the correlation coefficient $r^2$ or the $p-$value of an association test, can still enable successful inference of an individual's presence within a dataset.

In the following, we describe several fundamental approaches by which MI attacks can be practically executed on SNP datasets.

### 6.1.1 Distance-based Attacks

In distance-based MI attacks, the adversary is assumed to have direct access to the anonymized and (possibly) privatized records themselves. This threat model is particularly relevant when LDP mechanisms are applied to publish a privacy-preserving dataset, or when a trusted curator releases a privacy-preserving synthetic dataset.

Let the released sensitive dataset be denoted by $D^r$. The attacker first samples $|A|$ individuals from a population $A$ known *not* to belong to $D^r$. For each record $i \in A$ the attacker computes the distance (e.g., Hamming or Euclidean) from $i$ to every record in $D^r$ and retains the minimum distance:

$$d_{\min}^A(i) = \min_{x \in D^r} d(i, x).$$

A decision threshold $\gamma$ is selected such that $\alpha\%$ of the records in $A$ are correctly classified as non-members (i.e., $d_{\min}^A(i) > \gamma$ for $\alpha\%$ of $i \in A$), corresponding to a $(100 - \alpha)\%$ false positive rate.

Next, the attacker samples $|B|$ records from a population $B$ that *are* members of $D^r$. For each $j \in B$ the minimum distance to $D^r$ is computed similarly:

$$d_{\min}^B(j) = \min_{x \in D^r} d(j, x).$$

The resulting decision rule is that a query record is classified as a *member* if its minimum distance is below $\gamma$, and as a *non-member* otherwise. The **power** of the attack is then defined as the fraction of member

records in $B$ that are correctly classified as members:

$$\text{power} \; = \; \frac{1}{|B|} \sum_{j \in B} \mathbf{1}\{d_{\min}^{B}(j) < \gamma\},$$

where $\mathbf{1}\{\cdot\}$ is the indicator function. A larger power indicates greater success of the attacker at correctly inferring membership. The same attack can also be carried out in a reduced representation space (e.g., PCA space) by letting $d(\cdot, \cdot)$ operate on the corresponding embeddings rather than on the original records.

### 6.1.2 Likelihood Ratio Test

Hypothesis testing has been employed as a strategy for MI attacks against genomic data since at least 2008 (Homer et al., 2008). In this setting, the null hypothesis states that the target individual is not included in the studied or released dataset. Such tests are typically performed using aggregate statistics, such as MAFs, implying that the attacker does not require access to individual-level genotypes. Here, we focus on the likelihood ratio test (LRT), originally proposed by Sankararaman et al. (2009), as it remains one of the most widely adopted and empirically strongest MI attacks in the genomic privacy literature.

Assume that a data release provides MAFs for $L$ loci computed from a sensitive dataset $D^r$, and consider a target individual $i$ whose membership in $D^r$ is to be assessed. The attacker additionally requires access to a reference population that does not include individual $i$. Let $g_{i,j} \in \{0, 1, 2\}$ denote the genotype of individual $i$ at locus $j$, and let $f_j^{D^r}(a)$ and $f_j^{\text{ref}}(a)$ denote the MAFs at locus $j$ in $D^r$ and in the reference population, respectively. Assuming HWE (see Section 3.2.1), the genotype distribution at locus $j$ in a population $P$ with MAF $f_j^P(a)$ is given by

$$P(g \mid f_j^P(a)) = \begin{cases} (1 - f_j^P(a))^2, & g = 0, \\ 2 f_j^P(a)(1 - f_j^P(a)), & g = 1, \\ (f_j^P(a))^2, & g = 2. \end{cases}$$

The LRT evaluates, at each locus $j$, the likelihood of the *observed* genotype $g_{i,j}$ under the allele frequencies from $D^r$ versus the reference population, and sums over loci:

$$L_i \; = \; \sum_{j=1}^{L} \log \frac{P\big(g_{i,j} \mid f_j^{D^r}(a)\big)}{P\big(g_{i,j} \mid f_j^{\text{ref}}(a)\big)}.$$

Intuitively, this statistic measures whether the genotype of individual $i$ is more likely under the allele frequency estimates derived from $D^r$ or under those of the reference population. The same HWE assumption is used for both populations; and despite being an idealization, this model has been shown to work well in practice for MI in genomic data. The null distribution of $L_i$ can be empirically estimated by computing the same score for individuals known not to belong to $D^r$, enabling statistical inference on the membership status of the target individual.

## 6.2 Reconstruction Attacks

In reconstruction attacks, the adversary is assumed to have access to records in their original dimensionality, but with some SNP loci either masked, missing, or released in perturbed form. The attacker's objective is to recover these unknown genotypes for specific target individuals by exploiting LD structure, which can be estimated from public reference panels. A key privacy concern is that masking sensitive disease- or trait-associated loci may not protect the hidden information if correlated variants remain available. For example, the *APOE* locus in James Watson's publicly released genome was withheld because certain *APOE* variants are strongly associated with elevated risk of late-onset Alzheimer's disease. Nevertheless, Nyholt et al. (2009) showed that the hidden *APOE* status could be inferred from LD patterns among nearby variants, with roughly 50 surrounding SNPs being sufficient for recovery, illustrating how reconstruction attacks can undermine attempts to conceal sensitive genetic information.

A similar reconstruction strategy is employed in Yilmaz et al. (2022) against a LDP mechanism based on randomized response. Let $g_i \in \{0, 1, 2\}$ denote the true genotype at locus $i$ (the value the attacker aims to

infer), and let $y_k$ denote the *perturbed* genotype observed at locus $k$ in the released sequence (for $k \neq i$). The attacker is assumed to have access to *publicly available* pairwise genotype statistics derived from LD. For each candidate value $a \in \{0, 1, 2\}$, the attacker counts the number of loci whose conditional probability renders $g_i = a$ implausible given the observed $y_k$:

$$c_{i,a} = \sum_{k \neq i} \mathbf{1}\big\{ \Pr\big(g_i = a \mid g_k = y_k\big) < \tau \big\},$$

where $\mathbf{1}\{\cdot\}$ is the indicator function and $\tau$ is a plausibility threshold. Given a hyperparameter $\gamma \in [0, 1]$, any candidate $a$ for which $c_{i,a} \geq \gamma \cdot L$ is deemed improbable and eliminated, where $L$ is the number of loci used in the test. The attacker then selects among the remaining candidates using additional evidence or a ranking rule derived from the conditional probabilities.

This procedure illustrates how LD information can substantially reduce uncertainty about masked or perturbed genotypes, especially when many correlated loci are available. While the attack is described using pairwise correlations, higher-order correlations among multiple loci can be exploited in the same manner and would generally further strengthen the attacker's inference power, as demonstrated, for example, by Samani et al. (2015).

### 6.3 Kinship Attacks

Kinship attacks can be seen as a powerful class of reconstruction attacks that explicitly exploit *familial inheritance laws* in addition to statistical correlations in the genome. They were first proposed by Humbert et al. (2013) and later significantly strengthened by Deznabi et al. (2017) by incorporating higher-order genomic correlations and phenotype information. To date, this line of work represents one of the most sophisticated and effective inference frameworks for reconstructing hidden SNPs in the presence of family structure. The original work was motivated in part by concerns surrounding the publication of Henrietta Lacks' genome without her family's consent, which raised the question of whether the genome of one individual could reveal sensitive information about their genetically related relatives (Khan, 2011). More broadly, these attacks highlight that adversaries may infer health- or identity-sensitive genomic information about individuals who never disclosed their own genomes by exploiting relatives' released genomic data and publicly available family relationships.

In this attack model, the adversary is assumed to have access to some combination of the following information: (i) a partially observed, masked, or perturbed genotype of a target individual, (ii) partial genotype information of some relatives, (iii) phenotype information (e.g., physical traits or diseases) of some family members or the target individual, (iv) the family tree structure, and (v) public statistical knowledge, including population-level SNP frequencies, SNP-SNP correlation patterns, and SNP-trait association probabilities. The attacker's goal is to infer the missing genotypes of one or more family members as accurately as possible.

Let $X$ denote the collection of all SNP variables for all individuals in the family, and let $X_K$ and $X_U$ denote the observed and unknown subsets, respectively. The attack is formulated as a probabilistic inference problem and the adversary aims to compute the posterior marginals $p(x \mid X_K)$ for $x \in X_U$, and to output the most likely values of the hidden SNPs.

The key idea is to factorize the global joint distribution into a product of *local factors* that capture the different dependency sources:

$$p(X \mid X_K) \propto \prod_{\text{families}} f_{\text{fam}}(\text{parents}, \text{children}) \cdot \prod_{\text{correlations}} g_{\text{corr}}(\text{SNP block}) \cdot \prod_{\text{phenotypes}} t_{\text{pheno}}(\text{SNPs}, \text{traits}).$$

Here, *familial factors* $f_{\text{fam}}$ encode Mendelian inheritance constraints between parents and children, *correlation factors* $g_{\text{corr}}$ encode statistical dependencies between SNPs and *phenotype factors* $t_{\text{pheno}}$ encode probabilistic associations between SNPs and observable traits or diseases.

These factors define a factor graph whose variable nodes correspond to SNPs of individuals and whose factor nodes represent the three dependency types above. We include a simple illustrative example of

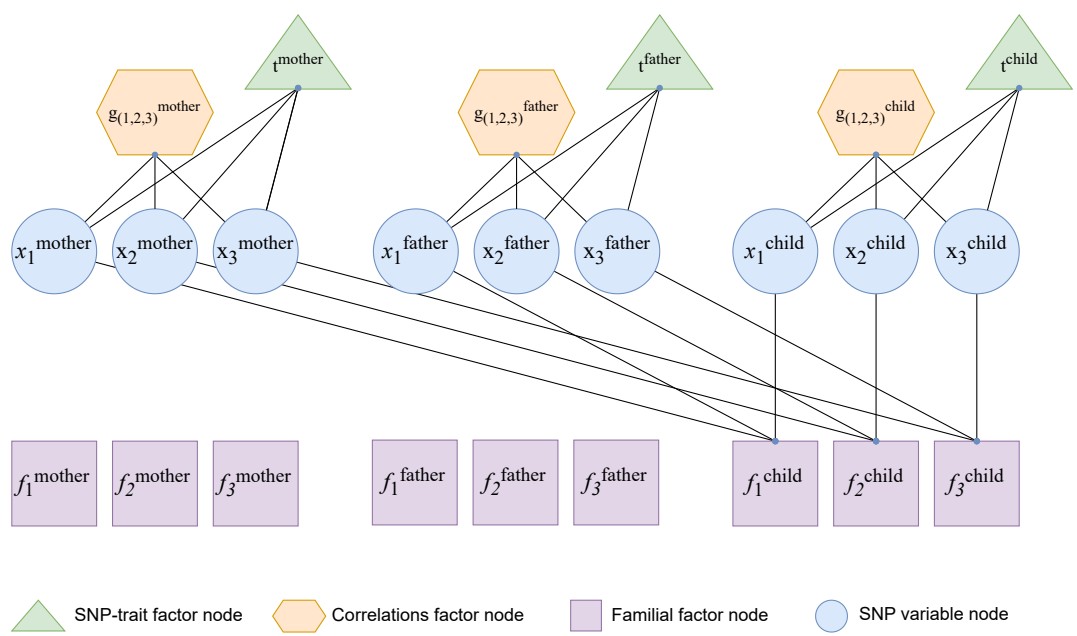

Figure 4: The factor graph representation for a trio family of father-mother-child and three SNP nodes.

such a factor graph for a trio family and three SNPs in Figure 4. Inference is then performed using belief propagation through message passing (Pearl, 2014), which efficiently approximates the marginal distributions of all unknown SNP variables by iteratively exchanging local messages between variables and factors.

After convergence, the attacker obtains approximate posteriors $p(x \mid X_K)$ for all $x \in X_U$ and reconstructs the genome by selecting the most likely value at each position. The strength of this attack comes from the *joint exploitation* of inheritance constraints, long-range statistical structure in the genome, and phenotype-genotype relationships, allowing highly accurate recovery even when only limited direct genomic information is available.

In a less sophisticated but still effective line of work, Almadhoun et al. (2020) and the GenShare framework (Alserr et al., 2021) build simplified kinship-based inference attacks directly on Mendelian inheritance constraints and aggregate (DP) query answers. Unlike the factor-graph-based attacks of Humbert et al. (2013); Deznabi et al. (2017), these methods do not perform global probabilistic inference, nor do they exploit SNP–SNP or SNP–trait correlations. Instead, the adversary uses knowledge of which relatives are included in a query result, together with inheritance probabilities, to statistically constrain and infer the target's SNP values from noisy summary statistics (e.g., counts or MAF queries). Despite their simplicity, these attacks demonstrate that even basic inheritance laws alone are sufficient to cause substantial privacy leakage from aggregate genomic releases when family members are present in the dataset.

# 7 A Release-Oriented Taxonomy of Differentially Private GWAS Methods

In this section, we present a systematic categorization of DP solutions for GWAS. In contrast to conventional surveys in the DP literature, which typically organize methods according to the privacy model, trust setting, or underlying mechanism, we adopt a *release-oriented* perspective centered on the information ultimately disclosed beyond the privacy barrier.

Focusing on released outputs allows us to disentangle the intended analytical objective of a method from the specific algorithmic tools used to achieve privacy, and to place approaches designed for association testing alongside methods that support broader downstream genomic analysis. This perspective further makes

explicit the auxiliary information assumed or required by different approaches, as well as the classes of techniques employed to enable release under privacy constraints.

Across the surveyed literature, we identify four primary categories of released information:

1. Methods that release aggregate association statistics for loci,

2. Methods that select and release the most significant loci based on statistics,

3. Methods that generate and release synthetic datasets in the SNP feature space, and

4. Methods that release sample-level information used during quality control and preprocessing.

In addition, we include a dedicated subsection addressing methods designed to handle datasets containing related individuals. These approaches target privacy risks arising from genetic dependencies and familial correlations and are, therefore, largely orthogonal to the type of released output. As such, they are treated separately from the main release-based categorization.

This taxonomy provides a unified view of how privacy mechanisms interact with GWAS workflows, clarifies implicit modeling assumptions across methods, and enables a more transparent comparison of privacy–utility trade-offs across otherwise heterogeneous lines of work. We provide summary tables for the surveyed papers in Appendix A.1.

### 7.1 Methods for Releasing Association Statistics

**High-level categorization**

Existing approaches to releasing DP GWAS association statistics can be broadly grouped according to the form of the association analysis being privatized and the privacy and utility considerations shaping the release setting:

1. **Classical contingency-table methods** derive central-DP global sensitivities for contingency-table-based association statistics, usually at the level of individual SNPs, and release privatized test statistics or their corresponding $p$-values. Genome-wide analyses therefore require privacy accounting across loci.

2. **Model-based association methods** privatize parameters or score statistics from multivariate or stratification-aware models, such as logistic regression, or EIGENSTRAT, operating on coefficients, latent scores, or model outputs rather than contingency-table statistics.

3. **Utility-oriented refinements** retain the underlying association statistics but improve utility through tighter sensitivity notions (e.g., smooth sensitivity), alternative representations (e.g., transform-domain perturbation), or refined noise calibration.

4. **Alternative privacy and query models** modify the privacy assumptions or querying setting rather than the underlying statistic, for example, LDP without a trusted curator or online settings with adaptive composition and budget management.

We first review methods that release DP versions of statistical quantities used in GWAS. Unless otherwise specified, these works operate in the central CDP setting, where a trusted data curator holds the dataset and releases privatized statistics.

We organize this section primarily by the type of association statistic being released, distinguishing between classical contingency-table tests and model-based approaches. We then discuss methodological refinements that aim to improve utility, as well as variations in the privacy or query model, such as local or interactive settings.

### 7.1.1 Classical Association Tests

**Case-control table-based tests (Section 4.3.1).** The first DP study tailored to GWAS (Uhlerop et al., 2013) proposed releasing MAFs, Pearson's $\chi^2$ statistics, and corresponding $p$-values under $\epsilon$-DP. Their approach relied on explicit global sensitivity derivations for the $3 \times 2$ case–control contingency table with equal numbers of cases and controls, and introduced a projection technique that truncates large $p$-values at a threshold $p^*$ to reduce sensitivity and improve utility. As in the subsequent contingency-table-based methods surveyed below, these sensitivity analyses are derived *per SNP (per contingency table)*, requiring privacy-loss composition when statistics are released across multiple loci.

Extending this line of work, Yu et al. (2014a) generalized the sensitivity analysis to settings with arbitrary case–control imbalance and derived refined bounds for both Pearson's $\chi^2$ and the allelic test statistic, thereby removing the equal-sample assumption common in earlier analyses. Moving beyond the specific GWAS setting, Sei & Ohsuga (2021) considered the release of $\chi^2$ statistics for general $I \times J$ contingency tables, with particular emphasis on small-sample regimes where classical asymptotic approximations degrade, and proposed a noise calibration strategy that better controls Type I errors.

Yamamoto & Shibuya (2021c) further broadened the scope of classical case–control tests by developing DP mechanisms not only for Pearson's $\chi^2$, but also for Fisher's exact test and the CATT. In contrast to earlier direct perturbation approaches, they advocated releasing $\log(p)$ rather than raw $p$-values to improve numerical stability and conducted empirical analyses to determine appropriate decision thresholds under DP. Collectively, these works progressively refine sensitivity analysis, relax structural assumptions, and improve the practical interpretability of privatized classical association tests.

**Family-based table-based tests (Section 4.3.2).** For family-based association studies, Wang et al. (2017a) provided the first systematic sensitivity analysis of the classical TDT statistic defined in Equation 4 under a family-level neighboring model. They derived explicit global sensitivity bounds for the TDT $\chi^2$ statistic and its corresponding $p$-value, and, following the projection strategy of Uhlerop et al. (2013), introduced truncated (projected) $p$-values to mitigate the large sensitivity inherent to raw $p$-values.

Extending this framework, Yamamoto & Shibuya (2021a) considered a more complex TDT setting with two affected children per family and introduced three linkage statistics, namely $\chi^2_{\mathrm{td}}$, a haplotype-based statistic, and their combined statistic. They derived separate sensitivity bounds for each statistic under the Laplace mechanism.

### 7.1.2 Model-Based Association Tests

**Regression models (Section 4.4.1).** Yu et al. (2014b) proposed a DP procedure for penalized logistic regression with elastic-net regularization and applied it to multi-SNP association analysis. Building on objective-function perturbation techniques for convex empirical risk minimization (Kifer et al., 2012; Chaudhuri & Vinterbo, 2013), they release a DP estimate of the regression coefficients, which serve as model-based association statistics. The privacy guarantee applies to the training step of the regression model; however, any prior SNP screening performed before fitting the model is not covered by their mechanism and would require separate privacy protection. Thus, their contribution lies in enabling the private release of multivariate regression coefficients rather than per-SNP test statistics.

**EIGENSTRAT and LMMs (Sections 4.4.2 and 4.4.3.)** Simmons et al. (2016) addressed model-based association testing under population stratification by privatizing EIGENSTRAT and LMM score statistics. They adopt a *phenotypic differential privacy* framework, in which neighboring datasets differ in one individual's phenotype while the genotype matrix is treated as fixed. Exploiting the linear dependence of the resulting association scores on the phenotype vector, they derive sensitivity bounds and apply Laplace perturbation to release privatized statistics for individual SNPs. This approach demonstrates that stratification-aware, model-adjusted association statistics can be released under DP, although the protection guarantee is restricted to phenotypic information; extending the mechanism to jointly protect genotype data would require a revised sensitivity analysis and potentially different mechanisms.

### 7.1.3 Performance-Oriented Refinements

The methods above primarily rely on global sensitivity under CDP with Laplace perturbation. While conceptually simple, global sensitivity is a worst-case measure and may introduce substantial noise in GWAS settings. Several works improve utility by adopting tighter sensitivity notions or perturbing alternative representations of the data.

**Smooth sensitivity.** Yamamoto & Shibuya (2023d) proposed smooth-sensitivity-based mechanisms for releasing classical GWAS statistics, including $\chi^2$ statistics (from both $3 \times 2$ genotype and $2 \times 2$ allele contingency tables) and TDT statistics, under CDP. Subsequently, Yamamoto & Shibuya (2025; 2026) introduced a direction-oriented smooth sensitivity framework, which refines the sensitivity analysis by accounting for the direction of perturbations and allows for tighter noise calibration. Empirical evaluations in these works demonstrate improved accuracy compared to global-sensitivity-based mechanisms. Together, these studies establish advanced sensitivity analysis as a promising direction for improving the utility of GWAS statistics under CDP.

**Transform-domain mechanisms.** Rather than perturbing statistics directly in the original data space, Roozgard et al. (2016) applied compressed sensing techniques and injected noise in a sparse transform domain before reconstructing summary statistics. This representation-level perturbation suggests that alternative compression or sampling schemes may provide complementary avenues for improving utility under DP, although further theoretical and empirical evaluation remains necessary.

### 7.1.4 Alternative Privacy and Query Models

**Local differential privacy.** Most prior work assumes a trusted curator under CDP. In contrast, Yamamoto & Shibuya (2023c) extended GWAS statistic release to the LDP setting using the GRR, enabling the computation of contingency-table tests, TDT, and EIGENSTRAT without a trusted data curator. Unlike central DP approaches that perturb aggregated statistics, their method privatizes individual-level categorical data, such as allele type, and reconstructs the required statistics via matrix-based estimation, with optimized distortion matrices for stronger table-level privacy guarantees. As with contingency-table-based CDP methods, privacy loss composes across queried loci and analyses; thus, genome-wide deployment requires careful budget management and may involve utility degradation under fixed privacy constraints.

**Interactive and online release.** Finally, Aziz et al. (2021) considered the online release of GWAS statistics under adaptive querying, where subsequent analyses depend on previously observed outputs. They proposed a framework that perturbs intermediate count statistics while dynamically managing privacy budgets through adaptive composition. In particular, their method addresses the privacy-accounting challenges that arise when multiple SNP statistics are queried sequentially, introducing online budget allocation and composition management strategies to control cumulative privacy loss. Their contribution lies primarily in privacy accounting and utility optimization under sequential querying, rather than in new sensitivity analyses for specific tests. This direction highlights the importance of supporting interactive GWAS workflows under DP.

## 7.2 Methods for Extracting Top Significant SNPs

> **High-level categorization**
> Existing approaches to private top-$k$ SNP extraction can be broadly grouped according to how statistical significance is privatized and used for locus selection and how candidate loci are scored or prioritized:
>
> 1. **Naïve perturbation-based selection methods** enforce DP by perturbing association statistics (see Section 7.1) and selecting loci by ranking the privatized scores.
>
> 2. **Utility-based selection methods** formulate SNP identification itself as a DP selection problem, using mechanisms such as the exponential mechanism to sample loci according to their statistical significance.
>
> 3. **Distance-based utility methods** retain the exponential-mechanism framework but redesign the utility function to capture the stability or robustness of significance decisions rather than raw statistic magnitude.
>
> 4. **Performance-oriented refinements** improve the privacy–utility trade-off through tighter sensitivity notions, alternative representations of the statistics vector, or related techniques that reduce effective noise requirements.
>
> 5. **Alternative problem formulations** depart from standard top-$k$ SNP extraction by modifying either the optimization objective or the underlying privacy model.

While privatizing association statistics provides a natural starting point, practical GWAS analysis ultimately requires the identities of the most significant loci rather than noisy versions of all test scores. In high-dimensional settings, perturbing and releasing statistics for all SNPs often incurs substantial utility loss, particularly when small differences in scores determine statistical significance. This motivates approaches that treat top-$k$ SNP identification itself as the primary DP task.

All methods considered in this section operate in the CDP setting, where a trusted data holder has access to the full dataset and releases either the identities of selected loci or privatized statistics associated with those loci. Because the identities of returned SNPs reveal ranking information, the selection procedure itself must satisfy DP. In general, a private mechanism first identifies the top-$k$ loci, after which their association statistics may optionally be released under a suitable allocation of the privacy budget.

Existing approaches primarily differ in how privacy is enforced during this selection process. As discussed below, methods vary in whether privacy is achieved through perturbation of statistics, randomized utility-based selection, stability-aware utility design, or sensitivity-reduction strategies that limit the amount of noise required for accurate locus extraction.

### 7.2.1 Perturbation-Based Selection Methods

**Overview.** The most direct strategy for releasing the top-$k$ significant SNPs under $\varepsilon$-DP follows a perturbation-and-selection philosophy. Each test statistic is first privatized via the Laplace mechanism according to its global sensitivity, and the top-$k$ SNPs are then determined from the perturbed values. Conceptually, selection is performed through ranking after noise injection, relying on post-processing invariance for privacy of the ordering. In its simplest form, a single round of Laplace noise is added to all statistics and the top-$k$ noisy values are released. Several works instead adopt a two-stage procedure in which noise is first added to privatize the selection step, followed by additional noise applied only to the statistics of the selected SNPs before release. This refinement, adapted from private frequent pattern mining (Bhaskar et al., 2010), allows the noise scale to depend on $k$ rather than the total number of SNPs.

**Examples.** Uhlerop et al. (2013) first formalized this paradigm for case-control GWAS by analyzing the sensitivity of the $\chi^2$-statistic and proposing Laplace perturbation followed by the selection of the most

significant SNPs. Yu et al. (2014a) further clarified the privacy implications of releasing the top-$k$ SNPs, explicitly addressing the information revealed by ranking and adopting the two-stage Laplace mechanism for private selection and release. The same perturb-then-rank template was subsequently extended to family-based association tests by Wang et al. (2017a) and Yamamoto & Shibuya (2021a), where they derived sensitivity bounds for TDT statistics and employed analogous Laplace-based top-$k$ release procedures.

**Remarks.** While conceptually simple, these approaches require perturbing all statistics, which can substantially reduce selection accuracy in high-dimensional settings. This motivates mechanisms that enforce privacy directly at the selection stage.

### 7.2.2 Utility-Based Selection Methods

**Overview.** While perturbation-based selection methods privatize test statistics prior to selection and rely on deterministic ranking, an alternative perspective formulates SNP identification itself as the primary private task. Instead of perturbing all statistics before ranking, the test statistic can be interpreted as a utility (score) function within the exponential mechanism (see Definition 7), enabling direct DP sampling of SNPs according to their statistical significance. In this framework, statistics such as the $\chi^2$ or TDT statistic define the utility function, and SNPs are sampled, typically sequentially without replacement, with probability proportional to $\exp\left(\frac{\epsilon \cdot u_i}{2\Delta}\right)$, where $u_i$ denotes the statistic and $\Delta$ its sensitivity. Following selection, the association statistics of the top-$k$ SNPs may be privatized and released under an appropriate privacy budget allocation, analogous to perturbation-based selection approaches.

**Examples.** Yu et al. (2014a) adopted this strategy for case-control GWAS by using the $\chi^2$-statistic as the score in the exponential mechanism to release the top-$k$ SNPs. Similarly, Wang et al. (2017a) applied the exponential mechanism to TDT statistics in family-based association studies.

**Remarks.** Because selection is performed through probabilistic sampling rather than deterministic ranking, the accuracy of the exponential mechanism depends critically on the separation between utility values and on the sensitivity of the chosen score function. In GWAS settings, small score gaps near the significance boundary and the inherently high sensitivity of classical test statistics may therefore lead to degraded selection accuracy. These limitations have motivated subsequent work that redesigns the utility function itself to capture the stability of significance decisions rather than raw statistic magnitude, giving rise to distance-based utility methods discussed next.

### 7.2.3 Distance-Based Utility Methods

**Overview.** Utility-based selection methods described above employ classical association statistics directly as utility scores. In practice, however, their accuracy may be limited by the high sensitivity of these statistics and by small score gaps near the significance boundary. This observation motivated a redesign of the utility function itself, leading to distance-based approaches that measure the stability of significance decisions rather than raw statistic magnitude.

First introduced by Johnson & Shmatikov (2013), the distance-score approach instantiates the exponential mechanism using the *shortest Hamming distance* (SHD) to a significance threshold as its score function. For a given SNP $i$ with test statistic $T_i(D)$ and threshold $\tau$, the score is defined as

$$\text{SHD}_i(D) = \min_{D'} \left\{ \|D - D'\|_H : \ T_i(D') \text{ crosses } \tau \right\},$$

that is, the minimum number of individual records that must be modified so that the SNP flips its significance status, either from significant to non-significant or vise-versa. Intuitively, this score measures how *stable* the significance decision is under perturbations of the dataset. The exponential mechanism then selects SNPs with probability proportional to

$$\exp\left(\frac{\varepsilon}{2} \text{SHD}_i(D)\right),$$

favoring loci whose significance is robust to small changes in the data. A central advantage of this formulation is that SHD has global sensitivity 1, since altering a single individual can change the distance to the decision

boundary by at most one. By replacing high-sensitivity statistics with a stability-based score, this approach substantially reduces the injected randomness and improves utility in top-$k$ SNP selection.

As with other top-$k$ mechanisms, once loci are selected, their association statistics may be released separately under DP. In Johnson & Shmatikov (2013), this is achieved by adding Laplace noise to the relevant contingency table counts of the selected SNPs and computing the desired test statistics from these noisy counts.

**Examples.** Originally introduced by Johnson & Shmatikov (2013) for selecting the top-$k$ associated SNPs in case–control contingency-table $\chi^2$-type statistics, this SHD-based exponential mechanism was formulated as a general distance-to-threshold selection principle. Yu & Ji (2014) subsequently specialized this framework to the allelic $\chi^2$ test, providing an explicit geometric interpretation of the score and efficient algorithms for computing the SHD with a formal sensitivity analysis. The mechanism has also been extended to TDT statistics in family-based studies (Wang et al., 2017a; Yamamoto & Shibuya, 2021a) and to population-stratification-corrected statistics such as EIGENSTRAT (Simmons et al., 2016). Empirically, these studies show that, by appropriately choosing the threshold $\tau$, the SHD-based exponential mechanism can achieve higher accuracy in identifying top significant loci than straightforward Laplace perturbation of the statistics or the direct application of the exponential mechanism to raw test statistics.

**Computational improvements.** A primary limitation of SHD-based methods is that computing the SHD score can be computationally intensive, as it requires determining the minimum number of record modifications needed to cross a significance threshold for each SNP. In high-dimensional GWAS settings, this per-SNP optimization quickly becomes a bottleneck. Yu & Ji (2014) were the first to explicitly address this issue for contingency-table $\chi^2$ statistics, providing a geometric reformulation that enables efficient SHD computation with a formal sensitivity guarantee. Simmons & Berger (2016) further refined the neighbor-distance approach by reformulating SHD evaluation as a structured optimization problem, significantly improving practical efficiency.

For family-based TDT statistics, the computational burden is even greater. Yamamoto & Shibuya (2021a;b) introduced new exact and approximation algorithms that substantially accelerate SHD computation while preserving sensitivity 1, making large-scale analyses feasible.

More recently, rather than accelerating exact SHD itself, Yamamoto & Shibuya (2023b) proposed a pseudo-SHD score that can be computed via a simple closed-form expression and combined it with a joint permute-and-flip mechanism (McKenna & Sheldon, 2020). This approach reduces runtime significantly while maintaining competitive accuracy, offering a practical alternative to threshold-dependent and computationally demanding exact SHD-based methods.

### 7.2.4 Performance-Oriented Refinements

Beyond perturbation-based and utility-based approaches, another line of work improves top-$k$ SNP identification by reducing the effective sensitivity involved in private selection. These methods exploit structural properties of GWAS statistics or data-dependent sensitivity bounds to limit the amount of noise required for accurate locus extraction.

**Transform-domain methods.** In GWAS, significant SNPs typically exhibit larger statistics than non-significant loci, suggesting that the structural separation between these groups may remain stable across neighboring datasets. Exploiting this observation, compressive mechanisms based on compressed sensing (Li et al., 2011) have been applied to sparse representations of the statistics vector. In particular, Yamamoto & Shibuya (2023a) combine the compressive mechanism with the Haar wavelet transform, applying compressed-domain perturbation to significant components while using the Laplace mechanism for the remaining entries. This hybrid design aims to leverage sparsity to reduce effective noise.

A related approach applies the discrete Fourier transform to the statistics vector before perturbation (Yamamoto & Shibuya, 2022). By truncating high-frequency components and adding noise in the spectral domain, the method reduces effective sensitivity and reconstructs a privatized statistics vector prior to top-$k$

selection. Both approaches rely on structural assumptions, such as sparsity or spectral concentration, whose validity and stability under neighboring datasets warrant further theoretical examination.

**Smooth-sensitivity-based methods.** Rather than modifying representation, smooth-sensitivity-based methods reduce noise by replacing global sensitivity with a data-dependent upper bound. The smooth private selection framework (Yamamoto & Shibuya, 2024) calibrates perturbation according to a smooth upper bound on local sensitivity of the statistics function, enabling more refined noise injection while maintaining pure $\varepsilon$-DP. In contrast to transform-based approaches, this strategy does not rely on structural assumptions about sparsity or frequency decay, but instead leverages dataset-specific sensitivity bounds to improve utility.

### 7.2.5 Alternative Problem Formulations

Beyond mechanism and sensitivity design, some works improve utility by modifying the problem formulation itself rather than the private selection mechanism. These approaches depart from the standard top-$k$ locus extraction setting by redefining either the optimization objective or the underlying privacy model.

Intelligent privacy-preserving scheme (IPP) method (Wang & Wu, 2022), for example, formulates the discovery of interacting groups of SNPs that jointly influence a phenotype as a DP combinatorial search problem, perturbing a multi-objective fitness function within an ant colony optimization framework to identify higher-order genetic interactions.

In contrast, bounded-prior membership privacy (Tramèr et al., 2015) relaxes the classical DP adversarial model by restricting prior beliefs, enabling improved utility guarantees under a weaker but explicitly characterized threat model for the task of top loci selection.

Rather than proposing new mechanisms for private top-$k$ SNP selection, these works alter the analytical or privacy framework under which genomic association discovery is performed.

### 7.3 Methods for Synthetic (Pseudo-)Dataset Generation

> **High-level categorization** Existing approaches to releasing privacy-preserving genomic datasets can be broadly grouped according to how the high-dimensional structure of SNP data is handled and where the privacy mechanism is applied:
>
> 1. **Block-based methods** reduce the effective dimensionality of SNP data by grouping strongly correlated loci (e.g., haplotype blocks) and release noisy block-level counts or synthetic block segments.
>
> 2. **Matrix-based / correlated-noise methods** treat the dataset as a matrix and inject structured noise whose correlation pattern reflects SNP–SNP (and optionally individual–individual) dependencies estimated from public data.
>
> 3. **Model-based methods** learn an explicit generative model of genomic variation (e.g., factor graphs, HMMs, or deep generative models) and enforce DP during training or at the level of the learned representations before sampling synthetic data.
>
> 4. **SNP subset selection methods** avoid modeling the full joint distribution and instead release carefully chosen subsets of loci or genotype states, possibly under relaxed or attacker-model-based privacy notions, yielding partial but structured views of the data.

While the works reviewed in Sections 7.1 and 7.2 focus on DP mechanisms that directly support association discovery in GWAS, an alternative paradigm is to release DP sanitized synthetic datasets (Dwork et al., 2009; Fung et al., 2010; Chen et al., 2024) that approximate the original, non-private data distribution. This approach is particularly appealing in the genomic setting, where domain knowledge is still rapidly evolving and analytical practices remain highly exploratory. As new statistical tests, downstream tasks, or analysis

pipelines continue to emerge, a sanitized dataset offers substantially greater flexibility, enabling reuse across a wide range of analyses, including tasks such as genotype imputation (see Section 5.2.1).

### 7.3.1 Block-Based Methods

**Overview.** Despite their apparent flexibility, DP synthetic data generation methods for SNP data were largely driven by necessity rather than convenience. Imagine a downstream task where, for example, the MAFs are required. Then, the extreme dimensionality of genomic sequences leads to an immediate explosion in global sensitivity, rendering naïve mechanisms unusable. Early approaches, therefore, rely on aggregating loci into tightly correlated *blocks* (e.g., haplotype blocks), thereby reducing the effective dimensionality. Instead of modeling the $\sim 2^L$ possible configurations of a sequence of length $L$, these methods operate on $B \ll L$ blocks, each admitting $M$ possible configurations, resulting in a state space of size $M^B$.

**Examples.** An early instance of this strategy is the top-down specialization method of Wang et al. (2014). The genome is partitioned into SNP blocks organized by simple taxonomy trees. Starting from a fully generalized representation, blocks are iteratively specialized to refine the partition, and the noisy counts of the resulting leaf groups are released using the Laplace mechanism (sensitivity 1). Because each individual appears in exactly one leaf, privacy follows by parallel composition. Zhao et al. (2015) extend the haplotype-block paradigm by perturbing haplotype counts at the block level and deriving noisy SNP allele counts while introducing an unequal budget allocation across blocks. Blocks with more complex haplotype structure receive larger privacy budgets and thus less noise, whereas simpler blocks are perturbed more aggressively, which significantly improves downstream utility. In contrast, Ahmed & Shimizu (2021) move from releasing noisy counts to directly synthesizing haplotypes. The genome is segmented into blocks, unique haplotype combinations are extracted within each block, and the exponential mechanism is used to sample segments in a DP manner, which are then concatenated to form a synthetic haplotype reference panel for genotype imputation.

**Remarks.** A common limitation of all haplotype-based approaches is their reliance, either explicitly or implicitly, on *a-priori* knowledge of a meaningful block or haplotype structure. If blocks are chosen arbitrarily, the correlation structure of the genome is not exploited, and little utility is gained over SNP-wise perturbation. At the same time, the dataset to be released cannot itself be used freely to infer this structure without violating privacy. In practice, these methods assume access to public reference panels or externally provided haplotype block definitions.

### 7.3.2 Matrix-Based / Correlated-Noise Methods

**Overview.** A distinct line of work treats a genotype dataset directly as a matrix and applies *matrix-structured* perturbations, whose scale or correlation pattern is designed to reflect dependencies in the data. Since both LD across SNPs and, in some settings, correlations across individuals (see Section 3.2) are highly structured, injecting i.i.d. noise at the level of individual entries can be unnecessarily destructive. Instead, these methods attempt to incorporate structural information into the noise model to better preserve utility for a given privacy budget.

**Examples.** A first and largely heuristic approach in this direction is proposed by Liu et al. (2017). They measure inter-individual correlation and high-order LD and perturb the genotype dataset at the matrix level by adding random noise, followed by a modulo operation to map values back into the valid genotype domain. The noise magnitude is scaled using summary measures of correlation and LD, reflecting a view in which correlation primarily modulates perturbation strength rather than being explicitly preserved in the noise model.

A more formal and structurally grounded solution is proposed by Jiang et al. (2025) in the PROVGEN framework for the privacy-preserving release of SNP datasets. The method first encodes each SNP into two binary bits and applies an XOR-based mechanism (Ji et al., 2021) that adds matrix-valued Bernoulli noise with an explicit dependence structure. In particular, column-wise correlations of the noise are calibrated

from a public reference panel in order to reflect SNP-SNP dependencies. Optionally, row-wise correlations can be introduced to model kinship relations if such information is available.

**Remarks.** The effectiveness of these methods relies on auxiliary knowledge about the correlation structure (e.g., LD estimated from public reference panels or pedigree information). In principle, since the noise model can include both column-wise and row-wise dependencies, this paradigm can be extended to settings with dependent individuals. We discuss this connection further in Section 7.5.

### 7.3.3 Model-Based Methods

**Overview.** A more complex class of approaches to generating synthetic datasets relies on expressive probabilistic or machine learning models to learn the structure of SNP data in an end-to-end fashion, and to enforce DP either during model training or at the level of lower-dimensional representations induced by the model.

**Examples.** He et al. (2017) propose an early model-based data release mechanism based on a factor graph similar to that in Figure 4 and belief propagation to generate synthetic genomic data. Their method factorizes the high-dimensional joint distribution using Mendelian inheritance and SNP-trait association statistics obtained from *public* resources, injects DP noise into the resulting low-dimensional factors, and samples synthetic genomes from the perturbed model. While conceptually appealing in that it shifts perturbation from raw data to a structured probabilistic model, the approach relies on external public summary statistics and would require a more rigorous sensitivity and privacy accounting to fully solidify its theoretical foundations.

A more advanced approach is taken by Hashimoto & Shimizu (2024), who build on the work of Yelmen et al. (2021) on GAN-based generation of synthetic haplotypes for imputation. In this line of work, a generative adversarial network (GAN) is trained to directly model the distribution of haplotypes, and DP stochastic gradient descent (DP-SGD) Abadi et al. (2016) is applied during training to control information leakage from the training data. DP-SGD enforces gradient clipping and injects noise into the optimization process, thereby limiting the influence of individual samples. This results in a pipeline that produces realistic yet sanitized synthetic haplotypes, with privacy protection enforced through the learning dynamics of the generative model itself.

Directly inspired by the Li-Stephens model (see Section 5.2), Rahimian & Fritz (2025) propose an end-to-end generative approach based on HMMs to learn SNP-SNP dependencies along genomic sequences. They extend the classical HMM by introducing locus-dependent (time-inhomogeneous) transition probabilities, which substantially increase the model's flexibility in capturing local LD patterns. The model is trained using DP-SGD to enforce DP at the level of full SNP sequences, and synthetic datasets are subsequently generated by sampling from the trained private model.

**Remarks.** Owing to their data-driven and end-to-end training paradigm, these models can learn complex correlation structures without requiring manually specified or predefined dependency rules. This makes this class of methods particularly appealing in that they have the potential to eliminate the need for any auxiliary public dataset or public summary statistics altogether. On the other hand, they naturally support a separation between a non-private pretraining phase and a private fine-tuning phase, as DP fine-tuning of deep models after non-private pretraining is by now a well-established technique (e.g., Abadi et al., 2016; Bu et al., 2023; Li et al., 2021), often yielding substantially improved utility for the same privacy budget on the sensitive data.

### 7.3.4 SNP Subset Selection Methods

**Overview.** This category comprises methods that relax or modify classical privacy notions in order to release *subsets* of SNPs, rather than the full dataset in its original feature space. The resulting privacy-preserving partial datasets can be used, for instance, to validate GWAS results (Turner et al., 2011; Zuvich et al., 2011) or to support downstream tasks such as imputation, linkage analysis, or population genetics while limiting privacy risks.

**Examples.** Halimi et al. (2022) consider a setting in which, in addition to releasing the summary statistics of the top-$k$ associated SNPs, the data owner also releases a *sanitized pseudo-dataset* consisting of individual-level genotypes restricted to these $k$ loci. Each genotype is perturbed independently using an LDP mechanism based on randomized response, yielding a noisy and partial view of the original data. This pseudo-dataset is intended for downstream consistency and verification checks.

Yilmaz et al. (2022) propose a correlation-aware local privacy notion called $(\varepsilon, T)$-dependent LDP for sanitizing individual-level genomic data. Their mechanism processes SNPs sequentially and, for each SNP, eliminates *genotype states* that are statistically implausible given the previously released (up to $T$) correlated SNPs under a public correlation model. The remaining states are then perturbed using a modified randomized response mechanism that preserves $\varepsilon$-indistinguishability among the non-eliminated states. Although this approach releases the full SNP sequence, this can be viewed as selective sharing at the *state level* rather than at the locus level.

Yilmaz et al. (2020) propose a selective disclosure framework for individual-level genomic data based on a new privacy notion called $\varepsilon$-indirect privacy. The genome is partitioned into predefined *sensitive* and *non-sensitive* SNPs, where sensitive SNPs (e.g., disease-associated variants) are never released. Non-sensitive SNPs are processed sequentially and are released in their true form only if doing so does not increase the attacker's ability to distinguish between any two possible states of any sensitive SNP by more than a factor $e^{\varepsilon}$, under an explicit attacker model based on public allele frequencies and LD. This method does not add noise and does not generate synthetic data; instead, it releases a filtered subset of the true SNPs. Since phenotype-associated variants are preferentially hidden, the resulting data is not suitable for association testing, but remains useful for tasks such as population genetics, or imputation.

**Remarks.** In the latter two works, the privacy and utility behavior of the algorithm is fundamentally determined by the assumed *auxiliary information* available to the adversary. In both cases, the sharing or sanitization decision at each step is driven by an explicit attacker model that specifies which correlations or background statistics the adversary can exploit. This auxiliary information model directly determines what increases or decreases the adversary's posterior confidence and therefore translates into which SNPs (or, in the state-level setting, which genotype values) must be suppressed or eliminated in order to satisfy the chosen privacy constraint.

## 7.4 Methods for Collaborative Preprocessing

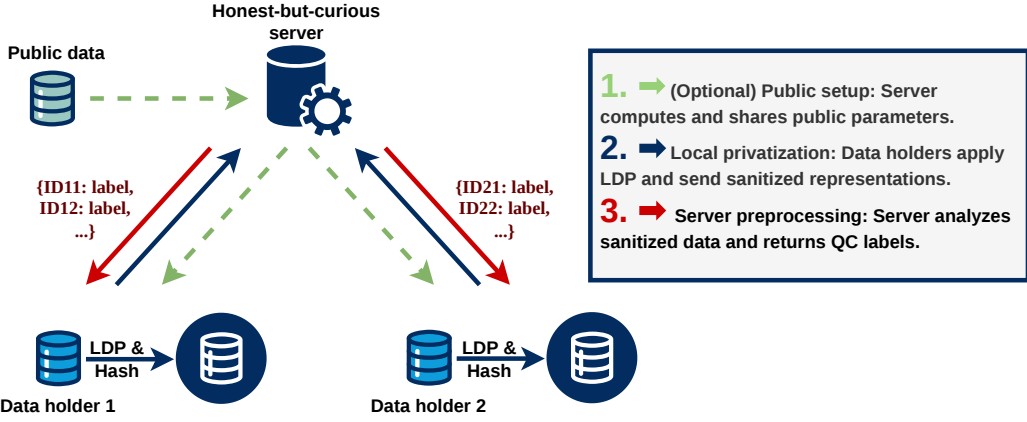

Figure 5: General pipeline of collaborative and privacy-preserving preprocessing.

So far, we have reviewed works that propose DP mechanisms for SNP datasets and for downstream GWAS analysis tasks. However, in realistic genomic analysis pipelines, an essential and unavoidable preliminary

stage is *quality control*, which consists of a series of data cleaning and preprocessing steps applied before any association testing is performed. For instance, as discussed in Section 5.4, detecting and correcting for population stratification is a critical step to ensure that GWAS results are not driven by confounding ancestry effects. Similarly, the presence of closely related individuals in a dataset can introduce bias into association results, as discussed in Section 5.3.2.

Performing such preprocessing steps is far from trivial in modern collaborative settings. Genomic data are typically fragmented across multiple data holders, and while federated learning or federated analytics approaches (Kairouz & McMahan, 2021; Google AI Blog, 2020) (see Section 2.3) enable joint model training or joint statistical analysis without centralizing the data, they implicitly assume that the underlying datasets have already been cleaned and harmonized. In practice, however, this cleaning and quality control must itself be performed *across* data holders, and *prior* to any downstream federated analysis.

**Setup and threat model.** In this section, we review an emerging line of work on DP-based *collaborative genomic preprocessing* in multi-institutional settings. Multiple data holders collaborate with a coordinating server to perform preprocessing tasks jointly, in a manner analogous to federated computation, without sharing raw genomic data. The server is assumed to be honest-but-curious. The pipeline is shown in Figure 5

**Population stratification.** The problem of collaborative population stratification and ancestry inference is addressed by Dervishi et al. (2023a). The server has access to a rich public reference genomic dataset, which is used to learn a global PCA model whose principal axes capture the major directions of population variation. The server distributes the top principal components to the participating data holders. Each data holder then projects its high-dimensional genotype matrix onto this shared low-dimensional subspace and applies an $\varepsilon$-LDP mechanism based on the Laplace distribution to the resulting real-valued coordinates. The sensitivity of the Laplace mechanism is *estimated* by the server from the public reference dataset and communicated to the participants. The locally privatized low-dimensional representations are sent back to the server, which aggregates them and applies $k$-means clustering (Hartigan & Wong, 1979) to infer population structure and assign each individual to an ancestry cluster. The resulting cluster labels are then returned to the respective data holders Ghasemian et al. (2025) generalize this framework by allowing the representation, noise-injection order, and clustering configuration to be selected in a data-driven yet privacy-preserving manner. They consider three LDP pipelines: *PCA→Noise*, *Noise→PCA*, and a *Noise-only* baseline. They let the server select both the clustering algorithm and the number of clusters using internal validation metrics computed on privatized data, incurring no additional privacy loss. Empirically, they show that the PCA-based pipelines consistently dominate the noise-only approach in terms of clustering utility, underscoring the role of public-model-based dimensionality reduction as a key enabler for private collaborative preprocessing.

**Kinship inference.** In a complementary direction, Dervishi et al. (2023b) address the problem of identifying related individuals, where multiple data holders wish to detect kinship relationships across the union of their datasets without sharing raw genomic data. The data holders first coordinate on a set of $m$ SNPs with similar MAFs. These SNPs are then shuffled using a shared permutation, perturbed using a custom variant of LDP, and augmented with synthetic individuals, with all steps designed to heuristically reduce re-identification risk. The chosen LDP variant is specifically constructed to preserve the structure relevant for kinship estimation. Using these obfuscated representations, the server computes pairwise relatedness via the KING coefficient (Equation 5) and reports related pairs, which can then be removed during quality control or, alternatively, grouped for collaborative family-based analyses (see Section 4.2). Notably, the invariance of the KING coefficient to SNP ordering ensures that the shuffling step does not degrade utility.

**Remarks.** An important observation is that, for these preprocessing tasks, the use of LDP rather than CDP is a necessity rather than a choice. Even though the setting involves multiple data holders, the operations considered here are inherently record-level, not aggregate. As a result, the only viable privacy-preserving strategy is to sanitize each data point individually in the chosen representation before any information is shared.

## 7.5 Dependent-Genomes Solutions

> **High-level categorization.** Methods for privacy-preserving genomic data release under familial dependence can be grouped by where the effect of inter-individual correlations is handled:
>
> 1. **Dependence-aware sensitivity methods** keep the DP mechanism unchanged but calibrate noise to a correlation-aware worst-case change, accounting for how modifications to one individual can propagate to relatives.
>
> 2. **Correlated noise methods** assume sensitivity is correctly calibrated and introduce structured correlations between noise terms to improve utility in high-dimensional or structured releases, while preserving the same DP guarantee.
>
> 3. **Familial constraint weakening methods** deliberately weaken, relax, or partially invalidate genetic transmission constraints, instead of preserving Mendelian structure and compensating for it in the privacy accounting. Therefore, limiting how information can propagate across related individuals, at the cost of biological or genealogical fidelity.

The original definition of DP (Definition 3) is designed for neighboring datasets that differ in only one record, but this assumption breaks down when multiple records are correlated. Liu et al. (2016) show that the existence of dependent records weakens classical DP: modifying one individual can induce changes in several correlated individuals. This issue is particularly prevalent in genomic datasets, where the genomes of relatives are highly correlated (see Section 3.2.2), especially when large datasets are assembled or synchronized without perfectly consistent cleaning and quality control. In this section, we review works that explicitly address this challenge.

### 7.5.1 Dependence-Aware Sensitivity

**Overview.** Perhaps the most straightforward solution to dependence is to deploy *group DP* (Section 2.2.1), which protects a group of up to $R$ individuals simultaneously by scaling the global sensitivity $\Delta f$ to $R\Delta f$. This, however, corresponds to a worst-case scenario of perfectly correlated or duplicate records, which, while possible, is typically overly conservative for genomic data, as relatives are highly and structurally correlated but not identical. Liu et al. (2016) were the first to introduce *dependent differential privacy*, which uses public or assumed probabilistic dependency structures and an upper bound on the number of correlated individuals to reduce the crude $R\Delta f$ bound of group DP to $\sigma\Delta f < R\Delta f$, where $\sigma$ is a correlation-aware factor, thereby aiming to reduce noise while preserving privacy.

**Examples.** Following this strategy, Almadhoun et al. (2020) show experimentally that an attacker can exploit, for example, Mendelian correlations (Section 3.2.2) to improve genotype inference when relatives are present in the dataset. They therefore propose to scale the global sensitivity of a *sum query $Q$* as $\Delta_{\text{dependent}}Q = \sigma\Delta Q$. Here, $\sigma$ depends on the number of related and unrelated individuals in the query result and is derived empirically from the attacker's success. Through their experiments they show that more relatives increase the attack success, while adding unrelated individuals decreases it.

GenShare (Alserr et al., 2021) can be viewed as a refinement of this approach. Instead of a purely multiplicative correction, it augments the classical global sensitivity by an additive dependency term, $\Delta_{\text{dependent}}Q = \Delta Q + \sigma$, which captures how changes to one individual's genotype probabilistically propagate to relatives via joint genetic distributions. This allows *Laplace-based mechanisms* to be tailored on a per-query basis, depending on which family members and unrelated individuals are included. Empirically, GenShare achieves privacy-utility trade-offs close to those obtained under the idealized independence setting. However, despite the more principled modeling, it still relies on heuristic calibration to resolve a circular dependence between the noise scale and the dependency term, and the sensitivity must be recomputed for each query and participant subset. Thus, both works ultimately provide carefully engineered, application-specific corrections to DP under dependence, rather than a fully general and compositionally robust framework.

### 7.5.2  Correlated Noise

**Overview.**  In most classical DP mechanisms, randomness is injected independently across queries, coordinates, iterations, or participants. By contrast, *correlated noise* mechanisms (Pillutla et al., 2025) deliberately introduce statistical dependence between noise terms added at different releases, while still ensuring that the *joint* output distribution satisfies the required DP guarantee. The key observation is that DP constrains only the likelihood ratios between the distributions of full outputs under neighboring datasets, and does not require the injected noise to be independent. By carefully correlating noise across time, dimensions, or agents, one can often preserve privacy while improving utility. Typical benefits include partial cancellation of noise across iterations, concentrating variance in less harmful subspaces, or avoiding repeatedly paying independent noise in directions that are queried many times. Such ideas arise naturally in continual and streaming release, iterative optimization and learning (e.g., DP-SGD (Abadi et al., 2016) and its variants), and distributed or federated learning, where many related outputs are produced.

Conceptually, correlated-noise mechanisms address a different issue from the dependence-aware sensitivity approaches discussed above. In the latter, the goal is to correct the *magnitude* of noise by recognizing that, under dependence, changing one individual can propagate to multiple records, thereby increasing the true worst-case change of a query. Correlated-noise mechanisms, by contrast, assume that sensitivity has already been calibrated correctly (e.g., via group DP or dependence-aware sensitivity), and use correlation only to *reshape* the randomness in order to recover utility. The two ideas are therefore complementary; sensitivity calibration fixes the privacy threat model, while correlated noise improves accuracy under that corrected model.

**Examples.**  In genomics, correlated noise is most naturally relevant in settings where the output is high-dimensional or structured, such as synthetic data generation or the release of many related statistics. For example, PROVGEN (Jiang et al., 2025) generates a synthetic genotype matrix using a matrix-valued noise distribution with explicitly modeled correlations. In their implementation, correlations are used primarily across SNPs to preserve LD structure, but the framework also allows, in principle, correlations across rows based on a kinship model. Such row-wise correlated noise would aim to improve fidelity and statistical realism by perturbing related individuals in a coordinated manner, thereby better preserving within-family similarity, IBD sharing, and pedigree-level structure in the synthetic data. While this possibility is mentioned in the paper, the experiments do not exploit it and the technical treatment of kinship-based row correlations is not fully developed.

**Remark.**  Overall, correlated-noise mechanisms represent a promising and still under-explored direction for genomic data release. They do not weaken the formal privacy guarantee, but instead trade independence of randomness for better preservation of population-level structure and downstream utility, once the correct sensitivity under dependence has been taken into account.

### 7.5.3  Familial Genetic Constraints Weakening

**Overview.**  A third class of methods addresses kinship-induced privacy leakage by deliberately weakening the Mendelian structure of the data itself. Instead of compensating for familial dependencies at the level of sensitivity calibration or noise correlation, these approaches directly degrade, relax, or partially invalidate genetic transmission constraints, either by masking loci, restricting disclosures, or perturbing inheritance factors. The underlying idea is to reduce the amount of information that can propagate across related individuals through Mendelian laws and linkage correlations, even if this comes at the cost of biological realism or genealogical consistency.

**Examples.**  A first line of work in this category operates under a strong and explicit threat model in which the adversary is assumed to know dataset membership, family relationships, and population-level genetic statistics. The $\varepsilon$-indirect privacy framework of Yilmaz et al. (2020) (see Section 7.3.4) can be extended to account for kinship and familial correlations by explicitly simulating Mendelian and linkage-based inference attacks. The release decision for each non-sensitive SNP is constrained not only by the privacy of the individual's own sensitive variants, but also by the potential information leakage about the sensitive SNPs

of relatives. As a result, SNPs that would enable effective propagation of information along family lines are withheld, thereby limiting the practical impact of Mendelian constraints on inference.

The core idea in Alserr et al. (2023) is conceptually similar but is cast within a standard DP framework. Instead of controlling disclosures directly, the method strategically hides a minimal number of SNPs from related individuals to reduce pairwise kinship coefficients (e.g., the KING metric in Equation 5) below a predefined threshold. This effectively weakens or destroys the statistical signatures of familial relationships in the dataset. The SNP selection problem is formulated as an integer optimization task that minimizes the number of masked loci subject to kinship constraints, after which standard Laplace-based DP mechanisms, as in the independent setting, are applied to answer aggregate queries. Empirical results show that this preprocessing-based weakening of familial structure yields substantially better privacy-utility trade-offs than DP mechanisms that instead increase noise to account for dependence. The method, however, relies on accurate kinship estimation and solving combinatorial optimization problems, which may limit scalability in large or complex pedigrees.

Finally, a more explicitly model-based approach is proposed by He et al. (2017), who address kinship by injecting DP not at the level of records or queries, but directly into the probabilistic structure that encodes familial and genetic dependencies. In their framework, Mendelian inheritance is represented as local factors in a factor graph. Privacy noise is added to these factors before sampling a fully synthetic dataset, which weakens the dependency patterns themselves and produces synthetic families that no longer strictly obey Mendelian statistics. Here, privacy is achieved by perturbing the generative mechanism of familial structure, rather than by protecting it through more conservative noise calibration. While this does not fundamentally redefine the privacy semantics under correlated records, it points to an interesting and still largely unexplored direction in which privacy mechanisms operate directly on structured generative models of genetic inheritance, suggesting that tighter integrations between dependency modeling and privacy definitions remain an open research problem.

**Remarks.** Taken together, these methods share the common philosophy of mitigating dependency-driven leakage by deliberately compromising the genetic coupling between relatives. In contrast to dependence-aware sensitivity or correlated-noise mechanisms, which preserve Mendelian structure and merely adjust the privacy accounting, this class of approaches trades biological and genealogical fidelity for privacy by weakening the very constraints that enable information to propagate across family members.

# 8 Future Directions and Open Challenges

Building on the insights from the surveyed works, this section highlights key open challenges that arise from fundamental mismatches between differential privacy assumptions and the scientific, statistical, and operational characteristics of genomic data. We structure future research directions across multiple levels, ranging from genomic data modeling and dependency-aware mechanisms to methodological and deployment-level considerations.

## 8.1 Genomics-Aware Privacy Modeling

A recurring challenge in privacy-preserving genomics arises from a mismatch between the assumptions underlying classical differential privacy mechanisms and the realities of modern genomic data collection and analysis. While many DP formulations implicitly assume static datasets and well-defined analytical objectives, genomic data is continuously reused, expanded, and reinterpreted as biological understanding evolves. These characteristics motivate privacy models that more closely reflect how genomic data is generated and utilized in practice.

**Reusable and exploratory genomic data resources.** Genomic data is inherently information-rich, simultaneously relating to multiple phenotypic traits, disease risks, and biological processes. Large-scale genomic datasets are therefore routinely reused to study diverse phenotypes over time, often for analytical tasks that are not fully specified at the time of data collection (Visscher et al., 2017; Bycroft et al., 2018). Such exploratory and repeated access patterns challenge privacy mechanisms designed for single-release

statistical outputs, whose guarantees degrade under adaptive reuse (Dwork et al., 2015). These characteristics motivate privacy-preserving data release paradigms that support sustained reuse, such as locally differentially private data collection or differentially private generative models capable of producing sanitized synthetic datasets (Ponomareva et al., 2025). Developing reusable genomic resources while maintaining meaningful privacy guarantees remains an important open problem.

**Beyond SNP-centric privacy models.** Existing differentially private approaches in genomics largely focus on single nucleotide polymorphisms, reflecting their historical dominance in genetic association studies. However, modern genomic resources increasingly capture a broader spectrum of genetic variation (as explained in Section 3.1), including insertions and deletions, short tandem repeats, and structural variants (Consortium et al., 2015). These variation types contribute substantially to gene regulation and phenotypic diversity (e.g. Fotsing et al., 2019; Weischenfeldt et al., 2013). Privacy mechanisms restricted to SNP-based representations may therefore overlook important sources of sensitive biological information, suggesting the need for privacy models capable of accommodating heterogeneous genomic variation.

**Whole-genome sequencing (WGS) and foundation models.** Advances in sequencing technology and computational infrastructure have accelerated the transition from genotyping arrays toward WGS (Park & Kim, 2016; Ng & Kirkness, 2010). The increasing availability of WGS datasets has enabled the development of large-scale genomic foundation models that learn biological structure directly from raw sequence data without strong prior assumptions (Feng et al., 2025). These models are beginning to reshape genomic analysis pipelines and motivate privacy mechanisms that operate beyond traditional summary statistics, as initial efforts such as (Huang et al., 2024) demonstrate.

Importantly, the shift to WGS substantially increases data dimensionality, introducing new challenges for reusable privacy-preserving data release. Mechanisms such as locally differentially private perturbation or synthetic data generation, which may remain feasible for lower-dimensional genotype representations, face significant scalability and utility limitations when applied to full genome sequences. Understanding how privacy mechanisms scale under the high-dimensional and strongly correlated structure of WGS data therefore represents a key direction for future research in genomics-aware differential privacy.

## 8.2 Modeling Genetic Dependencies under DP

Classical DP mechanisms are typically derived under assumptions of weak or manageable dependence between records. In contrast, genomic data exhibits structured multi-scale dependencies across loci, individuals, and populations, as discussed in Section 3.2. While the surveyed literature demonstrates that such dependencies can be exploited to improve utility or support specific analytical tasks, their broader implications for privacy accounting and mechanism design remain comparatively underexplored. This gap motivates future work on dependency-aware privacy mechanisms that more fully reflect the biological structure of genomic data.

**Extreme and inheritance-induced correlations.** Certain genomic components, such as mitochondrial DNA and the Y chromosome, exhibit near-deterministic inheritance patterns (Giles et al., 1980; Lippold et al., 2014; Jobling & Tyler-Smith, 2003). These regions are transmitted almost unchanged across generations, creating biologically coupled records among related individuals. Such inheritance-induced dependencies challenge the implicit assumption that privacy loss can be analyzed independently at the individual level (Gymrek et al., 2013; Samani et al., 2017; Ayday & Humbert, 2017). Existing DP formulations rarely account for privacy risks propagating along inheritance lines, suggesting the need for both theoretical analyses and empirical evaluation frameworks capable of quantifying privacy loss under extreme familial correlations.

**Correlation-aware noise and mechanism design.** Rather than treating dependence as an obstacle, structured correlations offer opportunities for improving the privacy–utility tradeoff. When dependency structure is properly incorporated, coordinated perturbation mechanisms can reduce effective privacy budget consumption while maintaining analytical utility. Prior work has demonstrated the benefits of correlated noise across a range of settings, including DP-SGD training (Choquette-Choo et al., 2023), group privacy (Jiang et al., 2024), graph data (Chen et al., 2014), matrix mechanisms (Li et al., 2015), LDP with correlated attributes (Du et al., 2021), and collaborative learning (Imtiaz et al., 2021). Genomic datasets

simultaneously exhibit locus–locus dependencies and individual-level correlations governed by inheritance constraints, suggesting that independently injected noise may be fundamentally suboptimal. Initial steps toward incorporating correlated perturbations in genomic settings have recently been explored in (Jiang et al., 2025).

A complementary line of work in computational genomics focuses on learning factorized representations that separate distinct sources of variation, such as familial relationships, population structure, or biological effects (He et al., 2017; Rakowski et al., 2024; Meisner & Albrechtsen, 2022). The ability of these models to disentangle structured dependencies suggests a promising direction for future research. Privacy mechanisms may be applied directly to such latent factors rather than raw genomic records. Developing DP mechanisms compatible with these factorized representations therefore represents an important open challenge for dependency-aware privacy in genomics.

**Learning dependencies end-to-end.** Many existing approaches surveyed in Sections 7.3 and 7.5 explicitly encode genomic dependencies through predefined assumptions, such as haplotype blocks or Mendelian inheritance rules. However, recent advances in representation learning increasingly shift this paradigm toward models that learn dependency structure directly from large-scale genomic data. Neural architectures have been employed to predict phenotypic traits (Liu et al., 2019), infer population structure (Siekiera et al., 2025), and train large genomic foundation models (Brixi et al., 2026), enabling complex biological effects to be captured within learned latent representations. In this setting, dependency modeling emerges implicitly rather than through manual specification, suggesting that future privacy mechanisms may operate on learned representations instead of explicitly released genomic statistics, as initial efforts such as (Chen et al., 2020; Huang et al., 2024) begin to explore.

## 8.3 Methodological Gaps in Modern GWAS Pipelines

Most differentially private GWAS methods have been developed under simplified statistical assumptions, whereas contemporary GWAS analyses operate through multi-stage, model-based pipelines involving population correction, feature screening, and complex regression models. This discrepancy introduces fundamental methodological barriers, as privacy mechanisms designed for isolated statistical queries do not naturally transfer to modern genomic analysis workflows.

**Modern threat models and privacy definitions.** Privacy guarantees are meaningful only with respect to the objects exposed by an analysis and the assumptions under which they are interpreted. Increasingly, genomic analyses incorporate expressive models such as neural architectures and genomic foundation models rather than relying solely on traditional statistical outputs. As a result, the effective release surface extends beyond association statistics to include intermediate representations, trained model parameters, and auxiliary artifacts. These additional outputs introduce new avenues for information leakage, even when individual components are analyzed in isolation (Fredrikson et al., 2015; Zhu et al., 2019; Carlini et al., 2021).

Consequently, limitations of existing differentially private GWAS approaches often arise not from the definition of DP itself, but from incomplete accounting of what is released by modern model-based genomic analyses. Developing stronger and systematically evaluated privacy attacks therefore plays a central role in identifying such vulnerabilities and guiding mechanism design (Abadi et al., 2016; Geiping et al., 2020; Nasr et al., 2018).

At the same time, the worst-case guarantees provided by $\varepsilon$-DP may be unnecessarily restrictive for certain genomic data-sharing settings. Relaxations such as $(\varepsilon, \delta)$-DP (Dwork et al., 2006a), concentrated DP (Dwork & Rothblum, 2016), Rényi DP (Mironov, 2017), and Gaussian DP (Dong et al., 2022) offer alternative tradeoffs between privacy and utility through refined characterizations of privacy loss. In parallel, prior-dependent frameworks such as Bayesian DP (Triastcyn & Faltings, 2020) and bounded-prior membership privacy (Tramèr et al., 2015) explicitly tailor guarantees to assumptions about data distributions or adversarial prior knowledge. Designing privacy notions that balance formal guarantees with realistic deployment assumptions in multi-stage genomic analyses remains an important open challenge.

**Revisiting foundational DP definitions for genomics.** Adapting DP to genomic data requires revisiting its foundational definitions in light of the structural properties of GWAS. In particular, the standard notion of neighboring datasets—differing in a single individual record—does not naturally capture dependencies induced by kinship, population structure, or shared genetic variation. This motivates alternative formulations, such as family-based neighboring relations in transmission disequilibrium tests (Wang et al., 2017a) or phenotype-level privacy guarantees for association models (Simmons et al., 2016).

Redefining neighboring relations in turn changes the relevant notion of sensitivity, making its computation a central methodological issue. In this context, smooth sensitivity is particularly promising, as it can provide tighter, instance-dependent bounds and thereby enable more realistic noise calibration. However, its practical use remains limited by the difficulty of deriving general computation methods and suitable noise distributions with strong theoretical guarantees. Recent work has begun to explore these challenges (Yamamoto & Shibuya, 2023d; 2025), but developing tractable and theoretically grounded approaches remains an open problem.

**DP mechanisms for GWAS pipeline components.** A central methodological gap lies in extending DP to the core components of widely-used GWAS analyses. Contemporary association studies routinely rely on model-based approaches such as EIGENSTRAT and LMMs, yet little work has addressed their sensitivity properties or developed general DP mechanisms for genotype–phenotype association under these models. The complexity of these estimators makes sensitivity analysis non-trivia and remains a major obstacle to principled noise calibration. Developing general approaches for sensitivity estimation and privacy-preserving inference in these widely used association models therefore represents an important direction for future research.

A closely related challenge concerns feature selection within GWAS analyses. While genome-wide association studies typically evaluate variants across the entire genome, such exhaustive testing becomes difficult under DP due to privacy-budget consumption and utility degradation. As a result, many approaches require a preliminary selection stage that identifies a smaller set of candidate loci prior to downstream association modeling. Designing effective privacy-preserving mechanisms for this selection step is therefore a key methodological challenge, distinct from the privacy modeling and deployment considerations discussed elsewhere in this section.

Existing approaches highlight the importance of score function design for top loci selection. In particular, distance-to-threshold metrics such as SHD have been shown to outperform direct use of statistical test scores (Wang et al., 2017a; Johnson & Shmatikov, 2013), although they often incur significant computational cost. Similarly, prior work has explored transform-domain representations for genomic data (Roozgard et al., 2016; Li et al., 2011), demonstrating that compressing signals and focusing on essential features can reduce sensitivity and improve utility. These approaches provide promising directions, but have primarily been studied in isolated settings and are not yet integrated into full GWAS analysis workflows.

Further improvements may be obtained through instance-adaptive mechanisms. For example, smooth sensitivity has been applied to top loci selection to introduce tailored noise at the level of individual candidates (Yamamoto & Shibuya, 2024). However, practical challenges remain in deriving tight bounds and selecting appropriate noise distributions. In particular, the role of noise directionality, including the potential benefits of one-sided perturbations observed empirically (McKenna & Sheldon, 2020; Yamamoto & Shibuya, 2024), is not yet well understood from a theoretical perspective. Addressing these limitations is essential for developing efficient and statistically meaningful DP mechanisms that integrate seamlessly with GWAS pipelines.

## 8.4 From Privacy Mechanisms to Real-World Deployment

Despite theoretical progress, many differentially private methods for genomic data remain difficult to translate into real-world deployment. Most approaches surveyed in this work assume controlled analytical environments, whereas practical genomic data collection and sharing operate within complex institutional pipelines involving multiple stakeholders, distributed infrastructures, and regulatory constraints (see e.g., Weinstein

et al., 2013; Freeberg et al., 2022). Consequently, challenges arising during data governance, preprocessing, and access management often emerge independently of mechanism design itself.

**Pretraining on public data, privacy on sensitive data.**  One promising direction for improving the deployability of differentially private genomic analysis lies in architectural redesign rather than purely mechanism-level refinement. Recent genomic foundation models demonstrate that expressive sequence representations can be pretrained on large publicly available or cross-species genomic resources, where privacy concerns are minimal (Nguyen et al., 2023; 2024; Brixi et al., 2026). Although these works do not explicitly incorporate DP, such pretraining regimes naturally decouple large-scale representation learning from downstream analyses performed on sensitive cohort data. This separation suggests a privacy-aware workflow in which only the final task-specific stage requires formal privacy protection. Restricting the application of DP to a limited portion of the training process may reduce utility loss while preserving meaningful guarantees. More broadly, this perspective underscores that improving the deployability of genomic privacy mechanisms may depend not only on new theoretical advances, but also on principled system-level design choices.

**Verifiability, data availability, and benchmarking.**  Publicly available genomic resources, such as the 1000 Genomes Project (Consortium et al., 2015), have provided important reference datasets for evaluating privacy-preserving methods. However, standardized genomic datasets with rich phenotype annotations remain scarce, largely due to privacy and governance constraints surrounding participant data (Perez-Pozuelo et al., 2021). Consequently, many studies either rely on a small number of commonly accessible datasets or evaluate methods on simulated or private institutional data, limiting independent verification and systematic comparison across works (Ayday et al., 2023). While existing public datasets enable partial evaluation, assessing DP mechanisms under realistic data complexity and deployment conditions remains challenging. Progress toward practically deployable solutions therefore depends not only on methodological advances but also on mechanisms that enable safe sharing of more representative genomic datasets, for example, through LDP data collection or the release of differentially private synthetic datasets.

The lack of standardized benchmarking frameworks further limits systematic comparison across proposed methods. Attack models, evaluation metrics, and utility pipelines vary substantially across studies, making it difficult to assess progress at the field level. Establishing community benchmarks for privacy-preserving genomics, analogous to evaluation platforms in machine learning (Pineau et al., 2021), therefore represents an important direction for enabling rigorous and comparable evaluation of future approaches.

**Privacy throughout the genomic data lifecycle.**  Privacy risks in genomics arise well before downstream statistical analysis. Genomic data must undergo sequencing, preprocessing, phasing, imputation, and population stratification, often across multiple laboratories and data custodians (Kamm et al., 2013; Zeevi, 2019). These distributed preprocessing stages introduce additional attack surfaces and governance challenges that are typically outside the scope of analysis-time privacy mechanisms. As discussed, for example, in Section 7.4, protecting genomic data therefore requires privacy-aware design across the entire analytical pipeline rather than solely at the final analysis stage. DP applied only during downstream computation may consequently fall short of deployment requirements faced by real institutions and policy makers.

**Interactive and personalized genomic data access.**  Practical genomic data sharing increasingly occurs through collaborative and interactive access models. In cross-silo environments, participating institutions or data contributors may require heterogeneous privacy guarantees depending on data sensitivity, regulatory constraints, or consent agreements. This motivates personalized DP frameworks capable of accommodating varying privacy requirements across participants (Niu et al., 2021).

Furthermore, genomic datasets are frequently accessed through iterative querying workflows in which researchers adapt subsequent analyses based on prior outputs (Cummings et al., 2018). Under naive implementations, such adaptive interaction leads to cumulative privacy loss through standard composition effects, limiting long-term usability of shared datasets. Developing mechanisms that support sustained interactive analysis while maintaining meaningful privacy guarantees therefore remains an important challenge for deployable genomic privacy systems.

# 9    Conclusion

In this work, we presented a structured and domain-aware survey of DP methods for GWAS, positioned intentionally at the intersection of privacy research and statistical genomics. The paper combines a structured conceptual foundation for genomic data analysis with a systematic compilation of existing literature, enabling privacy researchers to meaningfully engage with this interdisciplinary area.

Beyond compiling prior work, we adopt a release-oriented perspective that organizes methods according to *what is released across the privacy barrier* rather than solely by algorithmic technique or privacy definition. This viewpoint allows us to jointly analyze approaches that would otherwise appear disconnected, including methods operating under different trust models, release mechanisms, and variations of classical differential privacy. By directly covering a broad body of work, including many contributions that are often overlooked in conventional surveys, we emphasize emerging design patterns, implicit modeling assumptions, and underexplored methodological directions rather than pursuing narrow performance comparisons between individual methods.

A central contribution of this survey lies in bridging disciplinary gaps. A substantial portion of the paper is devoted to introducing the structural properties of SNP data and the operational realities of GWAS pipelines using terminology and abstractions accessible to the privacy community. While it is neither possible nor desirable to exhaustively cover all aspects of modern genomics within a single work, our objective is that readers completing this survey acquire sufficient conceptual grounding to reason about genomic data, interpret existing DP mechanisms correctly, and confidently navigate the broader interdisciplinary landscape.

We therefore view this work as an enabling foundation rather than a terminal summary. By unifying background knowledge, systematizing existing methods, and articulating open challenges arising from genomic structure, dependency, and repeated data use, we aim to support the development of the next generation of DP methods that are both theoretically principled and scientifically meaningful. Establishing a shared conceptual framework between genomics and privacy research is a necessary step toward realistic, trustworthy, and practically usable privacy-preserving GWAS, and we hope this survey contributes toward that goal.

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

# A    Appendix

## A.1    Summary Tables of Taxonomy

Tables 6 and 7 provide a structured overview of the methods discussed throughout the Taxonomy section. Specifically, Table 6 summarizes the works covered in Sections 7.1 and 7.2, while Table 7 covers the methods discussed in Sections 7.3, 7.5, and 7.4.

For each method, we report several key attributes to enable a consistent comparison across the literature. The **Output** specifies the final quantity or object released by the method. The **Task** describes the underlying problem formulation, distinguishing between classical statistical tests, model-based approaches (cf. Sections 4.3 and 4.4), and broader tasks as outlined in Section 5. For methods based on classical tests, we further indicate the supported study design using the labels (C) and (F), denoting *case-control* and *family-based* settings, respectively; methods that support both settings are marked as (C,F).

We further characterize the **Privacy barrier** as *central*, *local*, or *federated* differential privacy, depending on the trust and data access assumptions. The corresponding **Privacy notion** is categorized as *add-remove* (unbounded) or *replace* (bounded) in the central setting, or follows alternative definitions when specified by the original work. In the local setting, we use the term *per-attribute local* to indicate that the privacy budget is applied independently to each SNP, without explicit scaling with the total number of attributes.

The **Sensitivity type** indicates how sensitivity is defined within each method. We distinguish between *global* sensitivity (worst-case and dataset-independent), *local* sensitivity (data-dependent), and *smooth* sensitivity (data-dependent with a smooth upper bound). This terminology is applied consistently across privacy settings; for example, if the sensitivity of a locally private mechanism depends on the specific dataset, it is labeled as *local*.

In addition, we report whether the method relies on **Public information**, and at which level such auxiliary data is incorporated. Finally, the **Attack type** column indicates whether the work includes empirical privacy attacks, and if so, which type—*membership inference* (MI), *kinship*, or *reconstruction*—as defined in Section 6. Finally, beneath each table, we provide a link to the corresponding implementation repository whenever such information is publicly available.

| Approach | Output | Task | Privacy barrier | Privacy notion | Sensitivity type | Public Information | Attack type |
|---|---|---|---|---|---|---|---|
| Uhlerop et al. (2013) | Statistics, Top SNPs, MAFs | Classical (C) | Central | Replace | Global | None | None |
| Yu et al. (2014a) | Statistics, Top SNPs | Classical (C) | Central | Replace | Global | None | None |
| Sei & Ohsuga (2021) | Statistics | Classical (C) | Central | Replace | Global | None | None |
| Yamamoto & Shibuya (2021c)[a] | Statistics | Classical (C) | Central | Replace | Global | None | None |
| Wang et al. (2017a)[b] | Statistics, Top SNPs | Classical (F) | Central | Replace | Global | None | None |
| Yamamoto & Shibuya (2021a)[c] | Statistics, Top SNPs | Classical (F) | Central | Replace | Global | None | None |
| Yu et al. (2014b) | Logistic regression parameter | Model-based | Central | Replace | Global | None | None |
| Simmons et al. (2016)[d] | Statistics, Top SNPs | Model-based | Central | Replace | Global | None | None |
| Yamamoto & Shibuya (2023d)[e] | Statistics | Classical (C, F) | Central | Replace | Smooth | None | None |
| Yamamoto & Shibuya (2025)[f] | Statistics | Classical (C, F) | Central | Replace | Smooth | None | None |
| Yamamoto & Shibuya (2026)[g] | Statistics | Classical (C, F) | Central | Replace | Smooth | None | None |
| Roozgard et al. (2016) | MAFs | Classical (C) | Central | Replace | Global | None | None |
| Yamamoto & Shibuya (2023c)[h] | Statistics | Classical (C, F), Model-based | Local | Local | Global | None | None |
| [+]Aziz et al. (2021) | Statistics | Classical (C, F), Model-based | Central & Local | Replace & Local | Global | None | None |
| Johnson & Shmatikov (2013) | Statistics, Top SNPs, Others | Classical (C), Query release, LD pattern release | Central | Replace | Global | None | None |
| Yu & Ji (2014) | Top SNPs | Classical (C) | Central | Replace | Global | None | None |
| Simmons & Berger (2016)[i] | Top SNPs | Classical (C) | Central | Replace | Global | None | None |
| Yamamoto & Shibuya (2021b)[j] | Top SNPs | Classical (F) | Central | Replace | Global | None | None |
| Yamamoto & Shibuya (2023b)[k] | Top SNPs | Classical (C, F) | Central | Replace | Global | None | None |
| Yamamoto & Shibuya (2023a)[l] | Top SNPs | Classical (C) | Central | Replace | Global | None | None |
| Yamamoto & Shibuya (2022)[m] | Top SNPs | Classical (C, F) | Central | Replace | Global | None | None |
| Yamamoto & Shibuya (2024)[n] | Top SNPs | Classical (F) | Central | Replace | Smooth | None | None |
| Wang & Wu (2022) | SNPs interactions | Classical (C) | Central | Replace | Global | None | None |
| Tramèr et al. (2015) | Top SNPs | Classical (C) | Central | PMP | Global | None | None |

Table 6: Summary of methods covered in Sections 7.1 and 7.2. Rows corresponding to family-based methods are highlighted in  blue . All methods in this category operate without relying on public or auxiliary information, and none of the works evaluate empirical privacy attacks.

+ Online setting.
a https://github.com/ay0408/DP-statistics-GWAS
b https://github.com/mwgrassgreen/dpTDT
c https://github.com/ay0408/DP-linkage-analysis-TDT
d https://cb.csail.mit.edu/PrivGWAS
e https://github.com/ay0408/SS-based-Stats
f https://github.com/ay0408/DOSS
g https://github.com/ay0408/EnhancedDOSS
h https://github.com/ay0408/LDP-genome-statistics
i https://cb.csail.mit.edu/DiffPriv
j https://github.com/ay0408/DP-trio-TDT
k https://github.com/ay0408/Joint-PnF
l https://github.com/ay0408/CompLaplace
m https://github.com/ay0408/DP-DFT
n https://github.com/ay0408/Smooth-Private-Selection

| Approach | Output | Task | Privacy barrier | Privacy notion | Sensitivity type | Public information | Attack type |
|---|---|---|---|---|---|---|---|
| Wang et al. (2014) | Dataset | Classical (C) | Central | Add-remove | Global | Data owner | MI |
| Zhao et al. (2015) | Dataset | Classical (C) | Central | Add-remove | Global | Data owner | MI |
| Ahmed & Shimizu (2021) | Dataset | Imputation | Central | Replace | Global | Data owner | None |
| Liu et al. (2017) | Dataset | LD pattern release | Central | Replace | Local | None | None |
| Jiang et al. (2025) | Dataset | Classical (C) | Central | Replace | Global | Data owner | MI |
| He et al. (2017) | Dataset | None | Central | Add-remove | Global | Data Owner | None |
| Hashimoto & Shimizu (2024)$^a$ | Dataset | Imputation, Stratification, LD pattern release | Central | Add-remove | Global | None | MI |
| Rahimian & Fritz (2025) | Dataset | Classical (C), Population matching, LD pattern release | Central | Add-remove | Global | None | None |
| Halimi et al. (2022)$^b$ | Pseudo-dataset | Classical (C) | Local | Per-attribute local | Global | Verifier | MI |
| Yilmaz et al. (2020) | Pseudo-dataset | Other | Local | $\varepsilon$-indirect | Local | Data Owner | Kinship, Reconstruction |
| Yilmaz et al. (2022) | Pseudo-dataset | Query release$^q$ | Local | $(\varepsilon, T)$-dependent | Local | Data Owner | Reconstruction |
| $^+$Dervishi et al. (2023a) | Reduced-dimension dataset | Stratification | Local | Local | Local | Server | MI |
| $^+$Ghasemian et al. (2025) | Reduced-dimension dataset, dataset | Stratification | Local | Local | Local, Global | Server | MI |
| $^+$Dervishi et al. (2023b) | Shuffled pseudo-dataset | Kinship inference | Local | Per-attribute local | Global | None | MI |
| Almadhoun et al. (2020)$^c$ | Sum query | Query release$^q$ | Central | Replace | Local group | None | Kinship |
| Alserr et al. (2021) | Sum query, statistical query | Query release$^q$ | Central | Replace | Local group | None | Kinship |
| Alser et al. (2023)$^d$ | Count query, statistical query | Query release$^q$ | Central | Add-remove | Global | None | Kinship |

Table 7: Summary of methods covered in Sections 7.3, 7.5, and 7.4. The  shaded  rows correspond to methods that explicitly address the presence of dependent individuals.

---

$^+$ Methods for pre-processing.

$^q$ Queries are simple per-SNP questions such as *"what is MAF/$\chi^2$-statistic at locus i?"*. For CDP solutions, the global sensitivity is defined per query, and the privacy budget is allocated on a per-SNP (per-query) basis.

a https://github.com/cBioLab/PrivacyProtectedArtificialGenomes

b https://github.com/SpidLab/GWAS-Verification

c https://github.com/nourmadhoun/Differential-privacy-genomic-inference-attack

d https://github.com/CMU-SAFARI/SNP-Selective-Hiding

