# OpenReview forum: "What Survives Privatization? A Guide to Structure and Utility in Differentially Private Genome-Wide Association Studies"
_TMLR — Decision pending for TMLR_

### Review · Reviewer_pCHr · 2026-05-11

**Summary Of Contributions:**

The authors survey the literature on DP protection in GWAS. They begin by introducing several key concepts, including targeted DP regimes, SNPs, target GWAS models, and metrics for genome distances. Then, they describe the attack modes against genome databases and proceed to an overview of defense methods. They finish with an overview of the current problems in the field.

**Strength**
1. The survey structure is reasonable.
2. The survey looks comprehensive.

**Weaknesses**
1. Sections 3 and 4 are rather dense.

**Audience:**

Yes

**Audience Explanation:**

I think that medical or biological applications of ML constitute a vast field. However, I struggle to comment on the relevance of the problem considered in the paper.

**Broader Impact Concerns:**

I do not see a need for a Broader Impact Statement.

**Claims And Evidence:**

Yes

**Claims Explanation:**

The survey does not directly involve my research interests. Thus, it is hard for me to judge its comprehensiveness and rigor. However, I want to point out several issues as someone familiar with DP literature but not GWAS.

1. Section 3 is rather dense. While I acknowledge that it is impossible to rigorously describe all genetic concepts, I still think this section would benefit from more discussion. For instance, the concept of locus was quite confusing to me since the paper also describes mutations that could change the length of a DNA.
2. I think Section 4.4.2 would benefit from more discussion of ancestry structure. Currently, it is unclear why removing this structure from the data is desirable or important. Also, it is unclear how to pick hyperparameters for PCA.
3. Similarly, for Section 4.4.3, it can be beneficial to discuss what exactly changes in the $\Psi$ matrix and, in general, what is included in the estimation of covariance.
4. The paper would benefit from establishing a clearer link between Sections 7 and 8. For example, while Section 8.2 claims that genetic dependencies are rarely modeled for DP releases, it does not obviously follow from Section 7.

**Requested Changes:**

Please consider incorporating some feedback from the Claims and Evidence section.

---

> ### Author Response · Authors · 2026-06-05
>
> We thank the reviewer for the constructive feedback and positive assessment of the survey’s structure and scope. We especially appreciate the comments from the perspective of a DP researcher outside GWAS, as accessibility to such readers is a central goal of this work. We address the specific points below.
>
> > 1) concept of locus is vague
>
> We thank the reviewer for this suggestion. We revised Section 3 to improve clarity and flow, including restructuring parts of the exposition and expanding the discussion of genomic loci. In particular, we clarified that loci are defined relative to a reference genome coordinate framework and remain well defined across different forms of genetic variation, including variants that differ in scale or alter sequence length. We hope these changes address the reviewer’s concern regarding the notion of locus. If there are additional concepts in Section 3 that the reviewer feels would benefit from further clarification or discussion, we would be grateful for more specific guidance and would be happy to expand the section accordingly.
>
>
> > 2) Section 4.4.2. motivation unclear / how to pick PCA hyperparamters
>
> We thank the reviewer for this helpful suggestion. We have expanded Section 4.4.2 to clarify the role of ancestry structure in GWAS and the motivation for correcting it. In particular, we now explicitly discuss how population structure can induce spurious associations by creating ancestry-driven correlations between SNPs and phenotypes that do not reflect true causal effects. We have also clarified the role of the PCA hyperparameter $k$, noting that it determines the number of retained ancestry components and providing a brief discussion of practical component selection. To avoid duplication, we additionally point the reader to the later subsection on Population Stratification (Section 5.4), where PCA-based ancestry inference, practical considerations regarding principal component selection, and downstream diagnostics for assessing correction quality are discussed in greater detail.
>
>
>
> > 3) Section 4.4.3. what changes in $\Psi$.
>
> Thank you for this helpful suggestion. We have expanded Section 4.4.3 to clarify the role of the GRM $\boldsymbol{\Psi}$ in LMMs and to contrast it more explicitly with EIGENSTRAT. In particular, we now explain that, unlike PCA-based approaches that use $\boldsymbol{\Psi}$ to extract a limited number of principal components, LMMs incorporate the full GRM directly into the phenotype covariance model through the polygenic random effect. We also clarify what is included in covariance estimation, namely the genetic covariance component ($\sigma_g^2 \boldsymbol{\Psi}$) and the residual component ($\sigma_e^2 \mathbf{I}$), and briefly note the likelihood-based estimation of variance components from genome-wide genotype data and covariates under the null model.
>
>
> > 4) connection between section 7 and 8.
>
> We thank the reviewer for this observation. Our intention was not to suggest that genomic dependencies are absent from the literature. Indeed, Section 7 discusses several methods that explicitly incorporate dependency structures, including haplotype blocks, linkage disequilibrium, and Mendelian inheritance.
>
> The point we intended to emphasize in Section 8.2 is that these dependencies are typically incorporated through task-specific biological assumptions, whereas broader uses of dependency structure for privacy accounting, sensitivity analysis, correlated perturbation mechanisms, or learned representations remain comparatively underexplored. To make this connection clearer, we have revised the introduction of Section 8.2 to explicitly clarify the distinction between existing dependency-aware models and the broader future directions discussed in this section.

---

### Review · Reviewer_UWp7 · 2026-05-21

**Summary Of Contributions:**

This survey paper covers the topic of differential privacy (DP) within the context of genome-wide association studies (GWAS), particularly for data on single-nucleotide polymorphisms (SNPs). The main thrust of the paper is to bridge a gap between the existing DP literature (which is largely agnostic to domain-specific structure) and the reality of GWAS applications. Specifically, they point to domain-specific structure in GWAS data (including correlations between features and correlations between individuals) and application-dependent notions of utility, both of which should be incorporated into the development of DP methods for GWAS data.

The first main contribution is an extensive background on differential privacy; biological foundations; GWAS study designs, inference/estimation targets; other downstream tasks; and privacy attacks on SNP datasets. As I understand, this background is primarily targeted towards privacy researchers who may not have familiarity with GWAS, and offers an entry point to that audience.

The second main contribution introduce a taxonomy of DP GWAS methods, classified by the information they release, with four main categories of released information: aggregate statistics for loci (specific positions where SNPs are measured), most significant loci, synthetic datasets, and sample-level information (particularly in collaborative/federated settings focused on quality control and preprocessing).

The final contribution is a summary of future directions and open challenges, using their taxonomy to highlight unexplored areas of the design space for methods, including DP mechanisms that account for more of the dependence structure specific to GWAS data, threat models and privacy definitions that are more aligned with modern GWAS analysis pipelines, and DP mechanisms geared toward logistical aspects of real-world deployment.
### Strengths
1. **Bridging fields:** The paper serves a valuable role as a bridge between the DP literature and modern GWAS workflows: it provides enough detail on both DP and GWAS for readers from either community to step foot into the other, and it provides a well-curated set of background references and references to prior works at the intersection of these fields.
2. **Helpful taxonomy:** Rather than following the traditional taxonomy for DP methods (which focus more on DP-relevant ideas like privacy models or the underlying DP mechanisms), the taxonomy is geared towards GWAS applications, providing a helpful alternative angle on this literature. Further, this taxonomy emphasizes the richness of the application setting: genomic data is used for many downstream tasks of varying flavors, rather than a single type of downstream task, posing a different challenge for DP research in this setting.
3. **Extensive background:** Although the background is somewhat *too* extensive for the main paper (see [Weakness 1] below), the extent is impressive and will certainly serve as a useful, frequent reference to some researchers. The interplay between the scientific background (Section 3) and the GWAS analysis background (Section 4) is especially helpful: the biological details directly relate to the statistical hypotheses and models that are introduced, and make strongly support the idea that a sufficient understanding of the science is critical for DP research in this field.

### Weaknesses
1. **Length and lack of focus:** The biggest issue with the paper is its length and lack of focus. I understand the goal of being comprehensive, but I am a strong believer that the main body of a paper should introduce the "minimal necessary background" to understand the main contributions. Currently, the background (Sections 2-5) occupies 26 pages, over half of the paper. By the time the background is finished, most first and even second-time readers will already be a bit lost by how many details they need to follow. For researchers using this paper as a close reference, they will not mind if some details are moved to the appendix. Further, the structure of Section 7 feels quite repetitive, and Section 8 seems to have a few redundancies. Overall, I can easily see many ways in which the paper could be cut down to a more reasonable length (this list is not intended to be comprehensive):
	1. *Cutting non-SNP variations:* Indels, short tandem repeats, and structural variants are not brought up again in detail after they are introduced. These details could be completely removed or moved to the appendix.
	2. *Association tests:* Many details are introduced which do not reappear. For example, much of Section 4.3.1 could be condensed into 2-3 paragraphs: in case-control, the null hypothesis is independence between phenotype class and genetic variant, Pearson's $\chi^2$ test uses a test statistic that is efficient to compute and has a known asymptotic distribution under the null, whereas Fisher's exact test computes an exact p-value, which is computationally expensive. The typical reader does not need the contingency tables, the exact form of the statistics, etc.
	3. *Other GWAS analysis tasks:* Similarly, the model-based tests give far too much detail in the main paper, and it seems that some of other downstream tasks in Section 5 (e.g. calculating genetic distances) are not revisited in any depth in the later sections.
	4. *Section 7:* We have several pages of prose, not broken up by figures/tables etc., and each subsubsection follows the same structure. I found it somewhat hard to keep track of where I was in the section and where I was going, especially after all the reading up to that point.
	5. *Section 8 redundancies:* As one example, the difference between "privacy across genomic data pipelines" in Section 8.4 and "DP mechanisms for GWAS pipeline components" in Section 8.3 is not very apparent from those titles, and these are also closely connected to "pipeline-aware threat models". Throughout this section, several of the Section 7 references are revisited (e.g. He et al. 2017) - given the density of citations, much of Section 8 feels like an extension of Section 7.
2. **Sporadic claims/sections without appropriate citations:** In general, the paper does a good job providing key references for most of the places where they are warranted, giving seminal references for the background topics, extensive references on privacy attacks and DP methods for GWAS, etc. However, there is some deviation from this otherwise admirable scholarship:
	1. *Uncited claims:* Throughout, there are sentences which make non-trivial claims, and should be supported by citations. As a few examples (though there are several others): "A fundamental property shared by DP... is closure under post-processing" (page 10), "This design is inherently robust to population stratification bias..." (page 17), "these methods are robust to confounding due to population stratification..." (page 20). I do roughly understand how these claims would be justified (closure under post-processing should be closely connected to the data processing inequality, robustness to stratification bias can be viewed as the fact that Mendelian randomization can be seen as giving an instrumental variable, etc.), but they still need references.
	2. *Sporadic citation-light sections:* A few sections do not appear to have their citations filled out as thoroughly as the others, e.g. Section 4.5 ("Evaluation Metrics"), which should point to papers that define and use these metrics.

### Minor Weaknesses
1. **Mathematical clarity:** There are some minor issues around mathematical clarity, and I was disappointed to see that the paper lost much of its mathematical flavor in Section 7 (which is where many readers may be looking to understanding the existing DP+GWAS literature). In particular:
	1. $\varepsilon$-Differential Privacy is not well-defined, since the notion of "neighboring dataset" is not uniquely defined: formally speaking, the definition should be $\varepsilon$-DP with respect to some "neighboring" relation over $\mathcal{D}$ or some set of "neighboring" pairs $\mathcal{N} \subseteq \mathcal{D} \times \mathcal{D}$.
	2. On page 13, it is stated $f_i(a) + f_i(A) = 1$: this seems to imply that *any* variant at locus $i$ is coded as the minor allele $a$, whereas in other places it seems that $a$ only corresponds to the second most frequent allele.
	3. There are a few other very minor issues, e.g. $\mathcal{D}$ is used (in Definition 4) before it is defined (Definition 6), a closer proofread is needed.
2. **Loose use of causal language:** Section 4 slips into causal language in many places where it is not justified, e.g. for case-control designs, which only establish association. For example, "effect" is used at the end of the first paragraph in Section 4.1, "genomic regions influencing the trait" in the next paragraph, and in a few other places. One could say that these methods *seek* to quantify effects / localize such regions, but should be careful not to say that they actually do so, unless you introduce assumptions in some formal framework (e.g. potential outcomes or graphical causality).
3. **Not enough description of attacker motivation:** I can use my imagination, but it would be much more engaging if the paper described *why* some entity would be motivated to perform the different privacy attacks described. Even if a privacy attack is possible, it is not worrisome unless one can imagine a situation in which some entity has something to gain from recovering that specific information. Presumably, the references at the end of the first paragraph of Section 6 have some clear examples; it would be useful to incorporate 1-2 sentences about the potential attacker incentives into each subsection.

**Audience:**

Yes

**Audience Explanation:**

Both the differential privacy and the ML for bio communities would be interested in this paper: researchers in the DP community may be interested in the unsolved practical issues regarding DP for GWAS, and researchers in the ML for bio community may have a critical need for DP methods that better align with their workflow.

**Broader Impact Concerns:**

I have no concerns about the ethical implications of the work.

**Claims And Evidence:**

Yes

**Claims Explanation:**

The main claim is that the survey bridges the gap between the DP literature and GWAS analysis (for SNP data), which it clearly does with the extensive background and the novel taxonomy. Sub-claims, such as the importance of domain-specific patterns in the data, domain-specific workflow considerations, etc., are clearly argued and supported with appropriate references. As a survey paper, they make no theoretical/empirical claims that require further evidence.

**Requested Changes:**

1. **Address the length/focus issues (see Weakness 1):** Please give a detailed description of how to sharpen the focus of the paper and cut down on the length (e.g. which sections to remove or move to the appendix). This is critical to securing my acceptance of the paper, but feel free to reject some of my suggestions (as long as there is a good reason) and come up with other ideas.
2. **Add appropriate citations (see Weakness 2):** Please provide the citations you will add for the uncited claims I mentioned (and any others you find), and provide a brief description of what will be added to the citation-light sections. These are critical to securing my acceptance.
3. **Address minor mathematical issues (see Minor Weakness 1):** Please address the details I mentioned and do a pass through the paper to check for similar issues. This would strengthen the paper.
4. **Modify/weaken causal language (see Minor Weakness 2):** Please make some change to the existing language; this is critical for securing my acceptance, but the part about adding some formalism would just strengthen the paper.
5. **Briefly describe attacker motivation (see Minor Weakness 3):** Please add a brief description of attacker incentives for each attack; this would strengthen the paper.
6. **Consider moving section on multiple testing:** I would suggest moving the paragraph on multiple testing (currently in Section 4.4.1) closer to the beginning of Section 4: in most GWAS analyses (e.g. the association tests for case-control studies), one is interested in testing multiple hypotheses, not just in model-based testing. The earlier that this is introduced, the sooner the reader can understand some of the practical challenges of GWAS. You may also consider adding other adjustment procedures like Sidak correction.

---

> ### Author Response · Authors · 2026-06-05
> **Response part 1**
>
> We thank the reviewer for the careful and rigorous reading of our manuscript and for the detailed, constructive feedback. We appreciate the reviewer’s positive assessment of the paper’s motivation, taxonomy, and interdisciplinary scope. Below, we address the concerns point by point and describe the corresponding revisions.
>
> ### *Weaknesses*
>
> #### 1) Length and focus
>
> General comment: We thank the reviewer for the thoughtful comments regarding scope and organization. Our intent is for the paper to serve not only as a survey of existing DP methods, but also as a pedagogical reference aimed at lowering the entry barrier between the DP and GWAS communities. In this setting, the goal is not a minimal front-to-back presentation containing only material immediately reused in later sections, but rather a sufficiently self-contained, high-level overview that helps readers understand which biological, statistical, and workflow considerations matter for privacy-preserving GWAS, and where deeper study may be required. Consequently, some material is included to provide conceptual context, highlight aspects currently underrepresented in the DP literature, or motivate future directions, even when not revisited extensively in the taxonomy. More generally, fragmenting this tutorial-style background across the main text and appendix can be counterproductive, as it may require readers to repeatedly navigate between sections, increasing the risk of missing context or connections relevant to the particular aspect of GWAS or privacy they are interested in. We respond to the reviewer’s individual suggestions regarding scope, organization, and restructuring point-by-point below.
>
>
> > 1.1) cutting non-SNP variation.
>
> We appreciate the suggestion. We intentionally keep the brief discussion of non-SNP variation in the main text to contextualize SNPs within the broader landscape of genomic variation for readers entering genomics from the DP community. The discussion is intentionally minimal (a short orienting paragraph), immediately followed by an explicit statement restricting the scope of the survey to SNP datasets. Moreover, the limited treatment of non-SNP variation is itself a gap in the current DP genomics literature that we highlight later in the paper as part of the future directions discussion (Section 8.1). We therefore view this brief contextualization as relevant to the paper’s broader motivation and outlook. In our view, retaining this material in the main background improves self-containment, while moving such a small amount of material to the appendix would introduce unnecessary fragmentation for limited reduction in length.
>
> >  1.2 & 1.3) Association tests
>
> We agree that Figure 3 already provides a compact workflow-level overview of the association testing landscape. However, Sections 4.3–4.4 are intentionally designed to go beyond a naming-level summary. In our experience, several distinctions that are routine in genomics remain important sources of ambiguity for readers from DP/computer science backgrounds. For example, the relationship between allelic and genotypic formulations (specifically how their contingency tables relate), differences between case–control and family-based settings, and the assumptions underlying classical versus model-based tests. A substantially shorter treatment would risk reducing these sections to a catalog of test names without providing enough intuition for readers to understand why different DP papers adopt different testing formulations. We therefore intentionally keep these sections at a high-level but sufficiently explanatory level, while avoiding heavy mathematical development and focusing instead on conceptual differences, assumptions, and practical interpretation.
>
> > 1.3) Many of the downstream tasks in Section 5 are not revisited.
>
>
> Regarding Section 5, the fact that several of these downstream or structural evaluation tasks are not extensively revisited in the surveyed literature is, in part, precisely the motivation for including them. In contrast to association testing, which is already the dominant evaluation paradigm in DP-GWAS work, properties such as LD structure, population structure, and genetic distances are often overlooked despite being highly relevant, particularly for settings involving LDP, sanitized data release, or synthetic genomic datasets. Our goal in Section 5 is therefore not only to summarize what the current literature commonly evaluates, but also to expose important genomic notions of utility that privacy researchers may otherwise miss due to the domain-specific nature of these concepts. Because the paper is intentionally structured as a tutorial-style, self-contained reference, we prefer to present these utility notions coherently within the main body rather than fragmenting them across supplementary material.

---

> ### Author Response · Authors · 2026-06-05
> **Response part 2**
>
> ### *Weaknesses*
>
> #### 1) Length and focus
>
> > 1.4) Section 7 is hard to parse.
>
> We take this concern very seriously, as clarity of our work is of utmost importance to us and Section 7 is intended to help readers understand and navigate the DP–GWAS method landscape. We agree that the density and repeated prose structure can make navigation within the taxonomy more difficult. To improve readability, we have revised Sections 7.1 and 7.2 to include opener summary boxes analogous to those already used in Sections 7.3 and 7.5. We will further review the presentation of Section 7 and, where appropriate, consider incorporating additional high-level visual summaries (e.g., figures) to break up the prose, reinforce the taxonomy structure, and make the organization of the section easier to follow and use as a reference.
>
>
> > 1.5) Section 8 redundancies
>
> We thank the reviewer for this observation. We agree that the original titles may not have sufficiently emphasized the distinction between these topics, and we have revised the paragraph titles to make their scope clearer. We would also like to clarify that these paragraphs were intended to address conceptually distinct challenges that arise at different levels of privacy-preserving genomics. To improve this distinction, we revisited Section 8.3 and refined the framing and terminology to better separate these research directions.
>
> The paragraph on threat models and privacy definitions (Section 8.3) focuses on the assumptions underlying privacy guarantees, including alternative DP formulations and adversarial models appropriate for newer model-based genomic analyses. The emphasis is therefore on what should be protected and under which threat assumptions. In contrast, the paragraph on DP mechanisms for GWAS components (Section 8.3) focuses on methodological gaps within widely used GWAS analyses themselves, including approaches such as EIGENSTRAT and LMM, sensitivity analysis, and privacy-preserving feature selection. The central question there is how DP mechanisms can be integrated into the statistical methods used for association analysis. Finally, the paragraph on privacy across the genomic data lifecycle (Section 8.4) addresses a different class of challenges that arise before and beyond GWAS analysis, including data collection, sequencing, preprocessing, imputation, governance, and multi-institutional data management. The focus is therefore on deployment and end-to-end data handling rather than statistical methodology. We intentionally separated these topics because they correspond to different research directions and help clarify which open problems remain at the levels of privacy modeling, methodological development, and real-world deployment.
>
> Regarding the overlap with literature discussed earlier in the survey, we note that only a small number of representative works are revisited in Section 8. These references are included not to further review the literature, but to illustrate specific gaps that emerge from the surveyed methods. In each case, we explicitly discuss what remains unresolved and identify directions for future research. We therefore view these references as supporting the discussion of open challenges rather than as an extension of the survey itself. Nevertheless, we will perform an additional pass over Section 8 to further reduce any language that may blur these distinctions and to ensure that the focus remains on open challenges and future directions.
>
> #### 2) Sporadic claims/sections without appropriate citations
>
> We thank the reviewer for this observation. We agree that several claims and methodological descriptions would benefit from additional citations, and we have revised the manuscript accordingly.
>
> Specifically, we have added references supporting the examples highlighted by the reviewer, including the post-processing property of differential privacy [1], the discussion of robustness to population stratification [2, 3], and related claims regarding confounding and stratification correction [2,3]. We have also revisited citation-light sections and expanded the references in places where evaluation metrics, methodological assumptions, or standard genomics concepts were introduced without adequate support.
>
> Beyond the specific instances identified by the reviewer, we performed a broader citation audit throughout the manuscript to identify additional claims that warranted references. As examples, we added references for LDP budget scaling (Section 2.1.1), the definition of neighboring datasets (Section 2.2), foundational genomics concepts (Section 3.1), GWAS study design (Section 4.2), GWAS evaluation metrics (Section 4.6), allele-frequency-based utility evaluation (Section 5.1), and several other locations throughout the paper. We appreciate the reviewer’s attention to this issue. Prior to the final revision, we will conduct one additional careful pass through the manuscript to identify and address any remaining missing citations.

---

> ### Author Response · Authors · 2026-06-05
> **Response part 3**
>
> ### *Minor weaknesses*
>
> #### 1) Mathematical clarity
>
> General comment: Thank you for these comments regarding mathematical clarity. We agree that the manuscript would benefit from a careful tightening of several mathematical and notational details, and we will perform an additional proofreading pass to improve consistency, precision, and exposition throughout.
>
> Regarding the mathematical flavor of Section 7, we appreciate this perspective and would also have liked to include a more formal mathematical treatment of the taxonomy. However, the DP-GWAS literature spans a particularly broad range of outputs, privacy settings, assumptions, and methodological directions. We found it difficult to introduce substantially more formalism without causing a large increase in manuscript length and complexity.
>
> > 1.1) neighboring relationship is not defined.
>
> Thank you for this observation. We agree that the definition can be stated more formally by making the dataset domain and neighboring relation explicit. In the revised manuscript, we define DP for a randomized algorithm $\mathcal{A}:\mathcal{D}\to\mathcal{R}$ over neighboring datasets $D,D'\in\mathcal{D}$.
>
> > 1.2) confusion about minor/major frequency
>
> Thank you for pointing this out. We agree that the original wording could create ambiguity regarding the treatment of multi-allelic loci. We have revised the manuscript to explicitly state that the survey assumes the standard biallelic SNP representation, in which loci are encoded using two allelic states (major/minor allele), while multi-allelic loci are typically transformed into a biallelic representation or excluded during preprocessing. This clarification also makes explicit why (f_i(a)+f_i(A)=1) holds under our adopted representation.
>
> > 1.3 Mathematical formulas need proofreading
>
> We thank the reviewer for pointing this out. We have corrected this issue, along with several other formula- and notation-related inconsistencies identified in Section 4 (e.g., definition of $\mathbf{I}$ and $\mathbf{1}$ in Section 4.4). We will also perform a final careful proofreading pass in the revised manuscript to inspect and correct any remaining missing definitions, notation inconsistencies, or related issues.
>
>
> #### 2) Loose use of causal language
>
> We thank the reviewer for this important observation. We agree that several formulations in Section 4 used causal language too loosely in contexts where the discussed methods establish association rather than causation. In the revised manuscript, we have carefully reviewed Section 4 and adjusted the wording throughout to distinguish more clearly between association and causal interpretation. In particular, terms such as "effect'', "influencing'', and similar formulations have been revised where appropriate to use more precise association-based language, unless explicitly referring to model parameters or to causal hypotheses under additional assumptions.
>
>
> #### 3) Not enough description of attacker's motivation
>
> Thank you for this helpful suggestion. We agree that contextualizing attacker incentives improves the interpretation and practical relevance of the attack models. In response, we revised the attack subsections to explicitly include brief descriptions of the motivations and threat settings underlying each attack class. Specifically, we incorporated examples such as forensic DNA mixture analysis and disease-cohort disclosure for membership inference attacks, concealment of sensitive disease-associated loci (e.g., Alzheimer’s risk) for reconstruction attacks, and inference of genomic information about non-consenting relatives in kinship attacks.
>
> #### 4) Multiple testing is general to all methods mentioned
>
> We thank the reviewer for this helpful suggestion. We have moved the discussion of multiple testing into a dedicated standalone Section 4.5, where it is presented as a general component of GWAS inference applicable across the association frameworks discussed in Section 4. We chose to place this discussion after the association tests rather than at the beginning of the section, as this allows the reader to first establish the common SNP-wise testing paradigm underlying these methods before introducing genome-wide multiple-testing considerations.
>
> [1] Dwork, Cynthia, and Aaron Roth. "The algorithmic foundations of differential privacy." Foundations and trends® in theoretical computer science 9.3-4 (2014): 211-487.
>
> [2] Spielman, Richard S., Ralph E. McGinnis, and Warren J. Ewens. "Transmission test for linkage disequilibrium: the insulin gene region and insulin-dependent diabetes mellitus (IDDM)." American journal of human genetics 52.3 (1993): 506.
>
> [3] Laird, Nan M., and Christoph Lange. "Family-based designs in the age of large-scale gene-association studies." Nature Reviews Genetics 7.5 (2006): 385-394.

---

> > ### Comment · Reviewer_UWp7 · 2026-06-24
> > **Reviewer response**
> >
> > The response has adequately addressed my concerns - both of the main weaknesses were fully addressed, and the minor weaknesses were almost fully addressed. I give more details on each point below, and I thank the authors for their thoughtful response.
> >
> > ### Weaknesses
> >
> > **(1) Length and focus - fully addressed**
> > Thanks for addressing this concern.
> > - I appreciate that the goal is to serve as a pedagogical reference, in which case the length and depth are more appropriate, and I understand the risk of fragmenting the background across the body and the appendix. Since the authors have made purposeful choices about the content (e.g. including non-SNP variations and association tests), supported by good reasoning, I am happy to defer to their preferences. My final recommendation here is to make these choices clear to the reader by signposting, and perhaps more emphasis in the title/abstract/introduction about the "tutorial"-style nature of some sections, potentially with guidance as to what may be skipped based on the reader's interests/background.
> > - The revisions to Sections 7 and 8 sound reasonable and helpful - removing the redundancy and breaking up the prose will enhance readability. The response makes clearer the distinctions between the different challenges in Section 8.
> >
> > **(2) Appropriate citations - fully addressed**
> > Thank you for addressing this point, for adding specific references for the claims I suggested, and for the broader citation audit. I'm very happy with the scholarly accountability taken by the authors and confident that the revisions will be more than satisfactory.
> >
> > ### Minor weaknesses
> >
> > **(1) Mathematical clarity - fully addressed**
> > Thanks for the response. The suggested plan sounds appropriate - addressing the main points in the background without adding too much mathematical treatment to Section 7 (which would introduce unwarranted complexity). The fixes to the neighboring relation, the minor/major allele frequencies, and the notational consistency are well-described and quite satisfactory.
> >
> > **(2) Causal language - fully addressed**
> > The revisions are appropriate, I'm happy with the suggested modifications.
> >
> > **(3) Attacker motivation - mostly addressed**
> > Thanks for acknowledging this point - the response indicates that the authors clearly understood my reasoning, and the examples such as forensic analysis provide a good illustration of attacker motivation. Still, I think the text leaves a little bit of work to be done by the reader. For example, when you say "adversaries may infer health- or identity-sensitive genomic information about individuals who never disclosed their own genomes", I am still left to imagine what the adversary wants to do with this information: are we imagining an insurance company raising premiums, or something even worse, e.g. a criminal using that information to hurt/kill an individual?
> >
> > **(4) Multiple testing - almost fully addressed**
> > I'm happy that the authors are moving multiple testing into a standalone section. Having this section after the others is fine, but I would suggest a brief mention/foreshadowing in the intro to Section 4.

---

### Review · Reviewer_9SnK · 2026-05-26

**Summary Of Contributions:**

The paper aims to fill a real gap in the literature which is that privacy researchers typically do not have the deep domain knowledge required for GWAS and similar genome-focused applications. It serves as a self-contained statistical primer on SNP data and GWAS analysis. It is also a taxonomy of DP methods for SNP analysis with deep perspectives. Finally, the paper ends by listing open problems and gaps in the literature where DP methods do not match the realities of GWAS data.

**Audience:**

Yes

**Audience Explanation:**

- The writing shows dual domain awareness and articulates nuances in both fields
- Coverage of topics and methods is broad, useful and up to date.
- DP researchers will benefit from having this resource as they develop methods and approaches for GWAS applications

**Broader Impact Concerns:**

Not applicable.

**Claims And Evidence:**

Yes

**Claims Explanation:**

This is a primer so the contribution is not really in the novelty of the claims but rather in the combination of resources and the awareness for nuances at the intersection of these two domains. Within that scope the claims and setups are accurate and the importance of the challenges presented is clear and convincing. Minor overclaims are listed in the suggested changes.

**Requested Changes:**

The paper is mostly correct, with only fairly minor, but somewhat important, changes required for recommendation.

Important:
- The rounding of 1/3 = $0.\overline{6}$ to 0.66 is inaccurate. The sum with 0.33 rounded then sums to 0.99. Latex supports \overline for repeating decimals.
- There is another review [1] which covers GWAS and privacy methods, including differential privacy extensively. So the claim of this being the first comprehensive overview of DP methods for GWAS might be an overstatement, or at least a grey area, given the existing GWAS and privacy method review was broader than just DP methods. In any case, discussing and citing that paper would strengthen this paper.
- The Beagle citation appears wrong. I wonder if the authors meant Browning & Browning (the imputation method). Something to check and confirm if correct or not.
- There are a few garbled references, with what looks like email addresses in the author list, e.g. "ane 3 Bentley David R. drb@ sanger. ac. uk 3 m, and Washington University in St. Louis"
- The description of the experiments in Homer et al. (2008) state that it used 10,000 SNPs to get 0.1%, From my reading, Homer et al. (2008) states "with only 10,000 to 25,000 SNPs we were able to resolve mixtures where the person of interest was less than 1% of the total mixture" (p < 10⁻⁶)" and later "At the extreme, if we use*all the available SNPs, we can easily resolve mixtures where our person of interest is less than 0.1% of the total mixture"* (p < 10⁻⁶)" The paper seems to mix the two together.

Strengthen:
- in 4.4.1 multiple testing it says Bonferroni corrects with m as the "number of independent tests". Typically, the textbook definition of Bonferroni is "number of tests", a conservative correction that is irrespective of the dependency between tests.
- Acronyms are often used before definitions, e.g. TDT and CATT, or not defined at all, IPP.
- GRM seems to be defined once for genetic relatedness matrix and in another point as genetic relationship matrix. Needs standardisation.

[1] Noura Aherrahrou, Hamid Tairi, and Zouhair Aherrahrou. "Genomic privacy preservation in genome-wide association studies: taxonomy, limitations, challenges, and vision." *Briefings in Bioinformatics*, 25(5):bbae356, 2024.

---

> ### Author Response · Authors · 2026-06-05
>
> We thank the reviewer for the careful and informed reading of our paper and for the thoughtful and encouraging feedback. We appreciate the positive assessment of the paper’s interdisciplinary positioning, coverage, and utility for DP researchers working on GWAS-related problems. We address the specific comments and suggested corrections below.
>
> ### Important changes
> > 1) rounding is inaccurate.
>
> We have corrected the text and now use overline for indicating repeating decimals.
>
> > 2) the claim of this being the first comprehensive overview of DP methods for GWAS might be an overstatement.
>
> Indeed, several prior works [e.g., 1--4] survey privacy-preserving approaches for GWAS and genomic data from broader perspectives, including attacks, cryptographic methods, and differential privacy. We agree that citing and discussing this literature strengthens the positioning of our paper, and we have added these references together with a brief related-work discussion in the Introduction.
>
> Our intended distinction is not to claim the first overview of DP methods in this area, but rather a different scope and depth of analysis. Specifically, our work focuses on differential privacy for SNP/GWAS settings and provides a systematic, release-oriented perspective that examines what each method releases, the assumptions it makes, the utility notions it targets, and the protection afforded by the privacy mechanism. We revised the Introduction to clarify this positioning and better contextualize our contribution relative to prior survey literature. We also changed the phrase "first comprehensive overview" to "a comprehensive overview" in our abstract.
>
>
> > 3) BEAGLE citation is incorrect.
>
> Thank you for catching this error. We intended the Browning et al. imputation method. The citation is now corrected accordingly [5].
>
>
> > 4) Issue with references.
>
>  Thank you for pointing this out. We corrected the garbled references and performed an additional review of the bibliography to improve the cleanliness and consistency of all entries. We will also make one final pass over the references in the final revision.
>
>
> > 5) Homer et al description is not correct.
>
> Thank you for catching this imprecision. We corrected the description to distinguish the two results reported in Homer et al. (2008).
>
>
> ### Strengtheting changes
>
>
> > 1) Bonferroni test does not need to be independent.
>
> Thank you for pointing this out. Indeed, the classical Bonferroni correction is defined in terms of total number of tests, irrespictive of dependence. Techniques such as prior LD pruning are needed to make tests independent. We corrected the text and also moved the multiple testing paragraph to its own subsection (4.5) as per request of Reviewer UWp7.
>
> > 2&3) Issue with acronyms and GRM terminology.
>
> We thank the reviewer for pointing these out. We have corrected the acronym usage by defining previously undefined or inconsistently introduced abbreviations (e.g., TDT, CATT, IPP) and standardized the terminology for GRM throughout the manuscript. We will also make a final pass over the manuscript to ensure acronym definitions and terminology are fully consistent.
>
>
>
> [1] Aherrahrou, Noura, Hamid Tairi, and Zouhair Aherrahrou. "Genomic privacy preservation in genome-wide association studies: taxonomy, limitations, challenges, and vision." Briefings in Bioinformatics 25.5 (2024).
>
> [2] Mittos, Alexandros, Bradley Malin, and Emiliano De Cristofaro. "Systematizing genomic privacy research–a critical analysis." arXiv preprint arXiv 1712 (2017).
>
> [3] Bonomi, Luca, Yingxiang Huang, and Lucila Ohno-Machado. "Privacy challenges and research opportunities for genomic data sharing." Nature genetics 52.7 (2020): 646-654.
>
> [4] Aziz, Md Momin Al, et al. "Privacy-preserving techniques of genomic data—a survey." Briefings in bioinformatics 20.3 (2019): 887-895.
>
> [5] Browning, Brian L., et al. "Fast two-stage phasing of large-scale sequence data." The American Journal of Human Genetics 108.10 (2021): 1880-1890.

---

> > ### Comment · Reviewer_9SnK · 2026-06-19
> >
> > I thank the authors for addressing my concerns and comments. I believe the paper is clearer and stronger now.

---

### Decision · Action_Editor_bwfv · 2026-06-30

**Recommendation:** Accept as is

**Audience:**

Yes

**Audience Explanation:**

The paper is in the intersection of genomic/biological data analysis and private data analysis, both highly relevant fields for the TMLR audience.

**Claims And Evidence:**

Yes

**Claims Explanation:**

The initial draft had some concerns about looseness of definitions e.g. genomic locus, genetic relatedness matrix etc. But the edits improved this and the reviewers are now satisfied with the clarity of the paper. Overall this is in the style of a survey paper and reviewers agree that despite being long, it introduces the field comprehensively and is provides clear discussion of the various places where complexities in the field of private data analysis and genomics intersect.